# Distorted learning from local metacognition supports transdiagnostic underconfidence

Sucharit Katyal [1,2] ✉, Quentin JM Huys [1,3,4], Raymond J. Dolan [1,3] & Stephen M. Fleming [1,3,5] ✉

Individuals experiencing symptoms of anxiety and depression have been shown to exhibit persistent underconfidence. The origin of such metacognitive biases presents a puzzle, given that individuals should be able to learn appropriate levels of confidence from observing their own performance. In two large general population samples ($N = 230$ and $N = 278$), we measure both 'local' confidence in individual task instances and 'global' confidence as longer-run self-performance estimates while manipulating external feedback. Global confidence is sensitive to both local confidence and feedback valence—more frequent positive (negative) feedback increases (respectively decreases) global confidence, with asymmetries in feedback also leading to shifts in affective self-beliefs. Notably, however, global confidence exhibits reduced sensitivity to instances of higher local confidence in individuals with greater subclinical anxious-depression symptomatology, despite sensitivity to feedback valence remaining intact. Our finding of blunted sensitivity to increases in local confidence offers a mechanistic basis for how persistent underconfidence is maintained in the face of intact performance.

Computational and cognitive approaches in neuroscience and psychiatry have made great strides in understanding how humans perceive and represent their environment. A significant component of human mental activity, however, involves how we think about ourselves. Metacognitive beliefs about our skills and abilities have a pervasive impact in educational and clinical settings, affecting people's decisions about whether to pursue new activities[1]. In experimental studies of metacognition, a robust link has been established between transdiagnostic symptoms of anxiety and depression and underconfidence in performance. These relationships are observed in both local confidence judgments on individual trials[2–5] and global confidence as measured through long-run self-performance estimates[6] (which we refer to as global SPEs[7–9]). Conversely, remission from depression symptoms, through either therapy or antidepressants, is found to ameliorate underconfidence[10,11]. However, previous studies examining the link between metacognition and symptoms have been descriptive, and a mechanistic understanding of why confidence distortions in anxious-depression persist despite otherwise intact performance remains elusive.

One promising route to understanding the source of these metacognitive biases is to unpack how confidence is formed, particularly at a global level. Several factors have been shown to influence confidence formation over and above objective performance. For example, induction of negative vs. positive affect or low vs. high reward expectations can decrease vs. increase local confidence, respectively[12–14]. Manipulating beliefs about performance on an upcoming task, either through feedback or expected task difficulty, affects local confidence on subsequent instances of a similar task[15]. Finally, local confidence is higher on trials immediately following positive compared to negative feedback[16]. With respect to global confidence, false performance feedback in an ambiguous task has been found to influence people's global SPEs[17].

[1]Max Planck UCL Centre for Computational Psychiatry and Ageing Research, Queen Square Institute of Neurology, University College London, London, UK. [2]Department of Psychology, University of Copenhagen, Copenhagen, Denmark. [3]Functional Imaging Laboratory, Queen Square Institute of Neurology, University College London, London, UK. [4]Mental Health Neuroscience Department, Division of Psychiatry, University College London, London, UK. [5]Department of Experimental Psychology, University College London, London, UK. ✉e-mail: ska@psy.ku.dk; stephen.fleming@ucl.ac.uk

Similarly, providing true performance feedback, compared to a no-feedback condition, can increase global SPEs, despite objective task performance remaining unaffected[8,18]. At the same time, local confidence remains a robust predictor of global SPEs in the absence of feedback[8,18,19]. Together these findings have informed a computational model in which global confidence is formed by probabilistically combining instances of local confidence with feedback to update a prior over expected performance[8] (Fig. 1a–c).

Despite this progress in understanding the computations underpinning confidence formation, whether and how these influences differ in people with anxious-depression symptoms remains unexplored. One attractive hypothesis is that underconfidence may be grounded in a tendency to incorporate more negative and less positive information

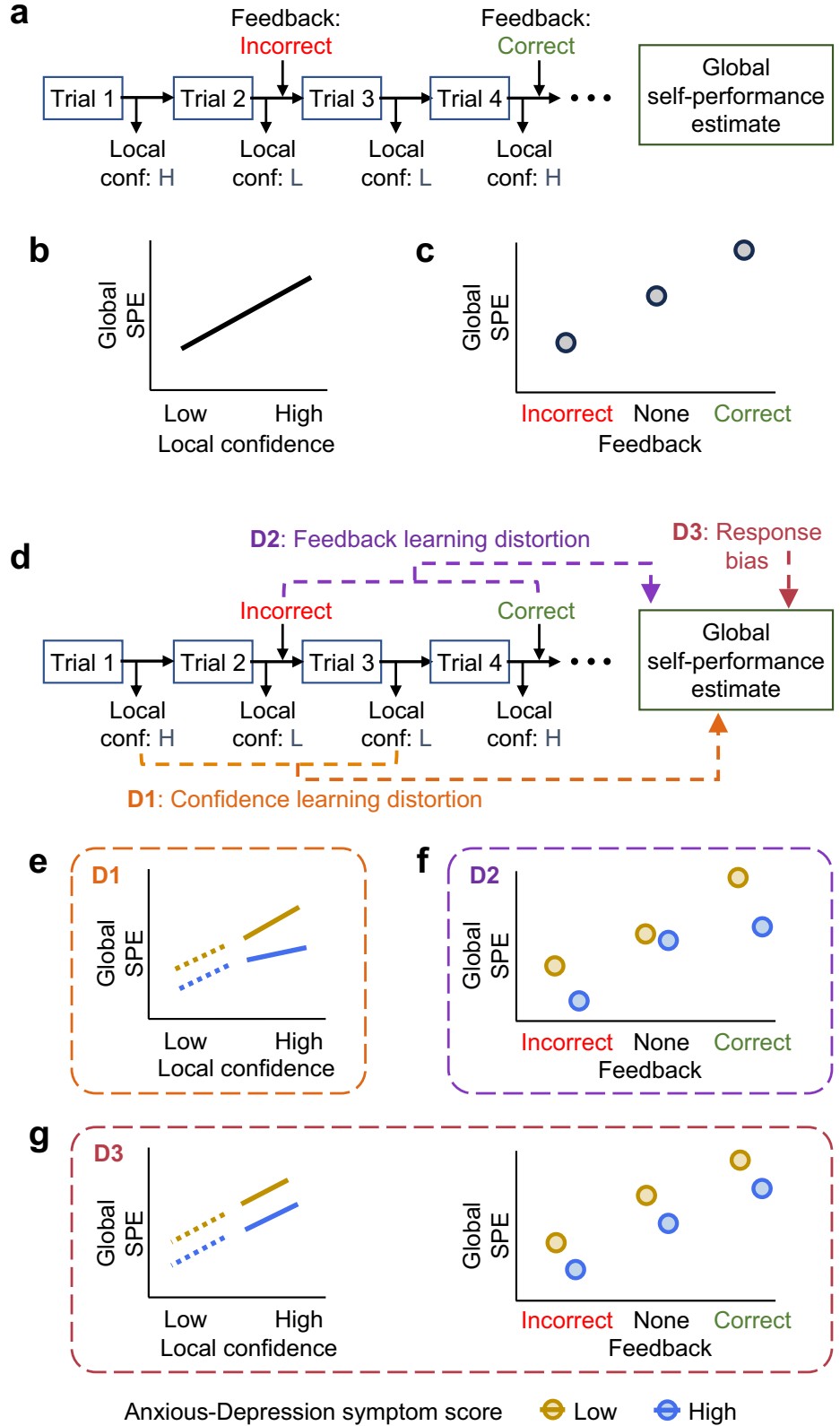

**Fig. 1 | A schematic depiction of the hypotheses. a** Outline of a canonical task in which individual trials are followed by local confidence reports and occasional (veridical) performance feedback. After completing the block, participants provide a global self-performance estimate (SPE) for that block. **b**, **c** Schematic illustration of different influences on global confidence within computational accounts of confidence formation. **b** Global SPEs are expected to increase proportionally with local confidence. **c** More frequent feedback on correct (respectively incorrect) trials is expected to increase (decrease) global SPEs. **d** Three candidate computational mechanisms for how the formation of global SPEs differs as a function of anxious-depression symptoms. **e**–**g** Visualisation of how the base patterns of global confidence formation depicted in (**b**) and (**c**) are expected to differ as a function of the different mechanisms described in (**d**). **e** Prediction for model D1, distortion in confidence learning – higher anxious-depression symptoms (blue lines) are associated with a blunted influence of higher local confidence on global SPEs compared to lower anxious-depression symptoms (gold lines). **f** Prediction for model D2, distortion in feedback learning—compared to the absence of feedback, higher anxious-depression symptoms are associated with a greater sensitivity to incorrect vs. correct feedback, and vice-versa for lower anxious-depression symptoms. **g** Prediction for D3, negative response bias – under a simple response bias model, we expect an overall reduction in global SPEs when comparing high vs. low anxious-depression symptoms in the absence of the nonlinearities depicted in (**e**) and (**f**).

when making future predictions, particularly about oneself[20–24]. Recent work has identified a self-related negativity bias in individuals with anxious-depression symptoms in relation to learning about expectations of future adverse life events[23,25] and when updating performance expectations from feedback[24]. Such biases could predispose individuals with anxious-depression symptoms to distort individual instances of feedback and/or local confidence when forming global SPEs. Specifically, when forming a global sense of confidence, individuals with anxious-depression symptoms may be more sensitive to negative (compared to positive) external feedback, low (compared to high) local confidence, or a combination of the two (Fig. 1d). In turn, local and global metacognitive evaluations not only mutually inform each other, but potentially shape abstract self-beliefs such as self-esteem[7,9,18], transcending individual tasks or cognitive domains. It is therefore also important to establish whether asymmetries in global confidence formation transfer across different tasks and/or influence more distal self-beliefs. Such transfer would be consistent with global confidence reflecting a slowly changing cognitive state with its own dynamics, akin to mood[26].

In the present study, we sought to determine how computations underpinning global confidence formation are altered in individuals with subclinical anxious-depression symptoms. We systematically manipulated performance feedback to influence global confidence. Then, using computational modelling, we distinguished between several mechanisms of how anxious-depression symptoms distort the impact of local confidence and performance feedback on global confidence. Specifically, such distortions could manifest as altered sensitivity to local confidence, altered sensitivity to feedback valence, or a general negative response bias (or a combination of these factors). Each of these three mechanisms makes qualitatively distinct predictions for the patterns of behaviour we expect to observe in our experiments (see Fig. 1e–g for schematic illustration). We also tested whether the influence of feedback on confidence generalises across two distinct cognitive domains (perception and memory) and (in Experiment 2) whether our feedback manipulation impacted broader facets of affective self-evaluation. These questions were addressed first in an exploration sample (Exp 1, N = 230), and then in a validation sample (Exp 2, N = 278) with hypotheses and analysis plans preregistered prior to data collection (preregistration document uploaded to osf.io/7xfqw). Due to small changes in the exclusion criteria, the sample size for Exp 1 deviated from the one reported in our preregistration. All deviations from the preregistered plan are detailed in Methods and Supplementary Methods.

In both experiments, our feedback manipulation robustly affected global confidence and impacted subsequent affective self-evaluations. To pre-empt our results, we found that individuals with greater anxious-depression symptoms exhibited reduced sensitivity to increases in local confidence, despite sensitivity to feedback valence remaining intact. Together our findings offer a mechanistic basis for how persistent underconfidence is maintained in the face of intact performance.

## Results

We measured performance and confidence in distinct cognitive domains using gamified perception and memory tasks (Fig. 2; for videos of sample task trials, see osf.io/2pq6y). On each trial, participants were required to choose the correct response (the higher density colour on the perceptual task; the stimulus from a previously memorised array on the memory task; see Fig. 2b for details) before rating their confidence in their response on a continuous scale. A continuous staircase procedure targeting ~71% correct performance ensured performance was equated across both participants and tasks. After completing baseline blocks of the task(s) without feedback, participants completed 'intervention' blocks in which intermittent feedback was delivered by an 'auditor' who occasionally checked participants' performance. Depending on the block type, the auditor was rigged to appear more often on correct trials (positive feedback blocks) or incorrect trials (negative feedback blocks); see 'Methods' for details. Intervention blocks were interleaved with test blocks without feedback. At the end of each block, participants provided a global self-performance estimate (SPE) on a sliding scale (Fig. 1a).

Participants were randomised to one of eight groups which differed according to the order of the intervention blocks they received (positive or negative first), whether the intervention was delivered on the perception or memory task, and whether the subsequent test blocks were in the same or different domain as the intervention blocks (Fig. 2a; see Supplementary Fig. 1 for Exp 2). Figure 2b illustrates one sample trial for each of the two tasks. Experiment 2 was designed as a replication of Experiment 1, albeit with minor differences to the design (see Methods). In what follows, we report the effects from both Experiments together, noting any inconsistencies between datasets. For statistical analyses, we report results from linear (mixed) regression models (see 'Methods' for details).

### Feedback impacts confidence without changing performance

Our asymmetric feedback manipulation led to systematic shifts in global SPEs in both tasks, despite performance remaining unchanged (Exp 1: Fig. 3a; Exp 2: Supplementary Fig. 2a; see Supplementary Fig. 3 for data separated by groups). Specifically, there was a significant main effect of Feedback type in predicting intervention block global SPEs (Exp 1: $\chi^2 = 273.77$, $p < 0.001$, β (regression estimate mean) = −0.26, 95%, CI = [−0.28 −0.23], Cohen's $d = 1.77$, 95% CI [1.55 1.99]; Exp 2: $\chi^2 = 332.08$, $p < 0.001$, β = −0.28, CI = [−0.31 −0.26], Cohen's $d = 1.77$, CI [1.57 1.96]) with positive feedback intervention blocks leading to higher global SPEs and negative feedback intervention blocks leading to lower global SPEs compared to baseline blocks (see Supplementary Information §1 for analyses of interactions between Feedback type, Task and Feedback order).

Importantly, there was no statistically significant effect of Feedback type in predicting mean accuracy (Exp 1: $\chi^2 = 0.03$, $p = 0.856$, β = −8e-4, CI = [−0.009 0.007], BF01 = 9.67 ± 1.6%; Exp 2: $\chi^2 = 0.22$, $p = 0.642$, β = 2e-3, CI = [−0.007 0.011], BF01 = 7.22 ± 28.4%; Fig. 3b) or mean difficulty (i.e. staircase) level (Exp 1: $\chi^2 = 1.73$, $p = 0.188$, β = 0.06,

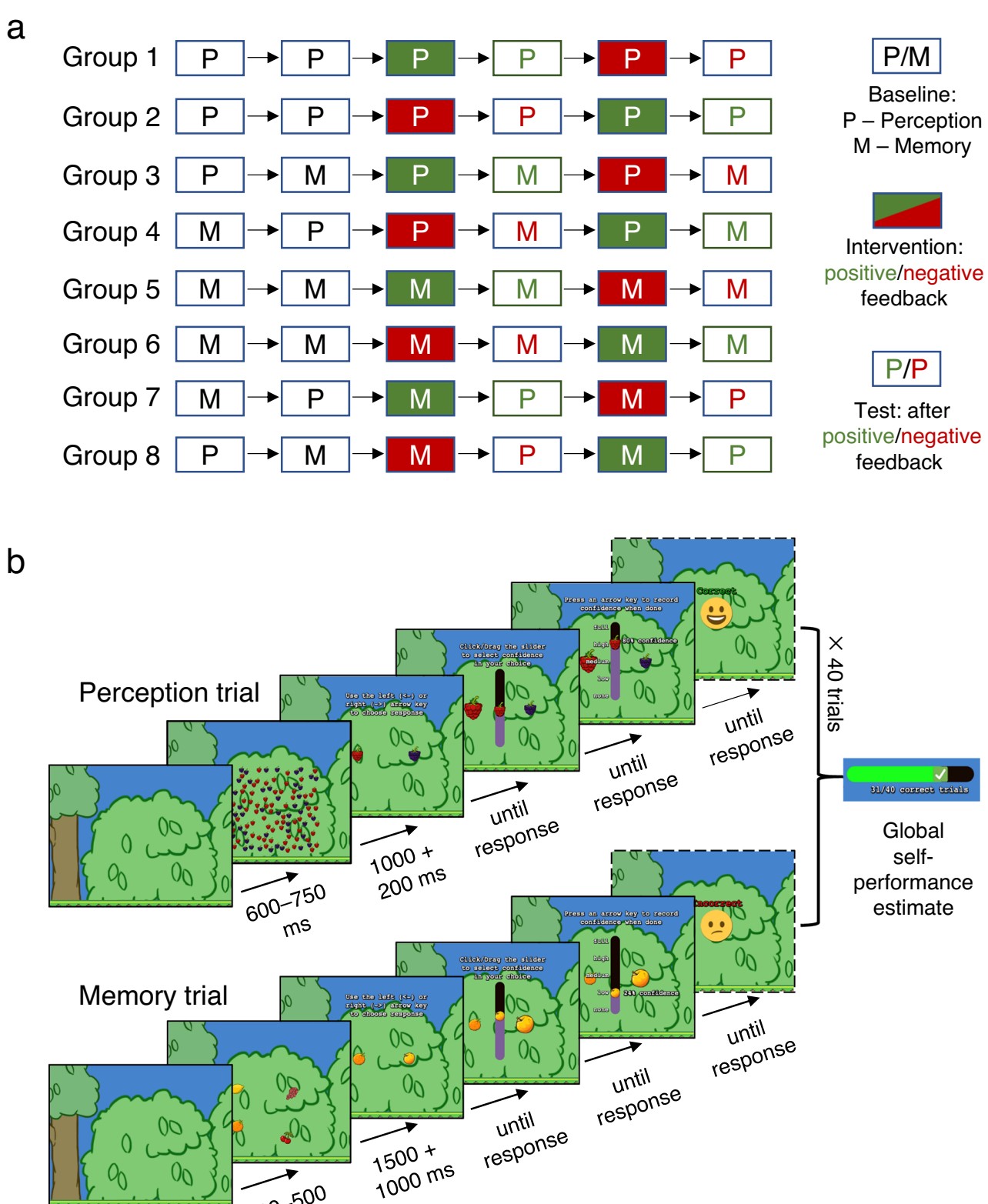

CI = [−0.03 0.14], BF01 = 4.17 ± 1.7%; Exp 2: $\chi^2$ = 1.50, $p$ = 0.221, β = −0.04, CI = [−0.11 0.03], BF01 = 4.24 ± 18.8%; Supplementary Fig. 5). Taken together, these analyses provide substantial evidence that shifts in global confidence were driven by the feedback manipulation and could not be ascribed to a secondary consequence of performance changes (Fig. 3b, Supplementary Fig. 2b, Supplementary Fig. 4). Feedback type also did not statistically significantly predict local

confidence in intervention blocks in Exp 1 ($\chi^2$ = 0.15, $p$ = 0.702, β = −2e-3, CI = [−0.03 0.14]; Fig. 3c; Supplementary Fig. 6), although in Exp 2 average local confidence was significantly higher on positive compared to negative feedback blocks ($\chi^2$ = 8.04, $p$ = 0.005, β = −0.015 CI = [−0.025 −0.005]; Supplementary Fig. 2c).

Replicating previous findings[8,18], average local confidence (meta-cognitive bias) significantly predicted global SPEs across blocks (Exp 1:

**Fig. 2 | Study design and stimuli. a** The eight groups of participants and the order in which they performed the perception (P) and memory (M) tasks in Exp 1. Participants within a group performed one or both tasks. A task sequence began with two baseline blocks followed by two sets of intervention blocks (with feedback), interleaved with test blocks (without feedback). For groups that performed both tasks, their order during baseline was randomised across participants. Unfilled boxes indicate the absence of feedback. Green-filled boxes indicate positive feedback blocks (more feedback on correct than incorrect trials) and red blocks indicate negative feedback blocks (more feedback on incorrect than correct trials). The colour of the subsequent test block indicates whether feedback delivered on the preceding (intervention) block was positive (green) or negative (red). **b** Single trials of the perception (top) and memory (bottom) tasks depicted using screenshots of stimuli shown in the experiments. For the perception task, participants attended

the green bush in the middle where 121 stimuli of two different kinds of berries (red raspberries and dark purple blackberries) appeared at non-overlapping random locations for 1000 ms. Participants were asked to respond (without time limit) as to whether there were more raspberries or blackberries, and then use a slider to report their confidence. After reporting their local confidence, participants pressed a key to proceed. At the end of each block of trials, participants were asked to report their global self-performance estimate on that block. Feedback for correct or incorrect trials appeared with a defined frequency depending on the intervention block (positive or negative). For the memory task, a set of fruits appeared on the bush for 1500 ms. Participants were tasked to memorise the fruits and then decide which of two fruits was present in the set. Confidence reporting and feedback for the memory task were similar to the perception task.

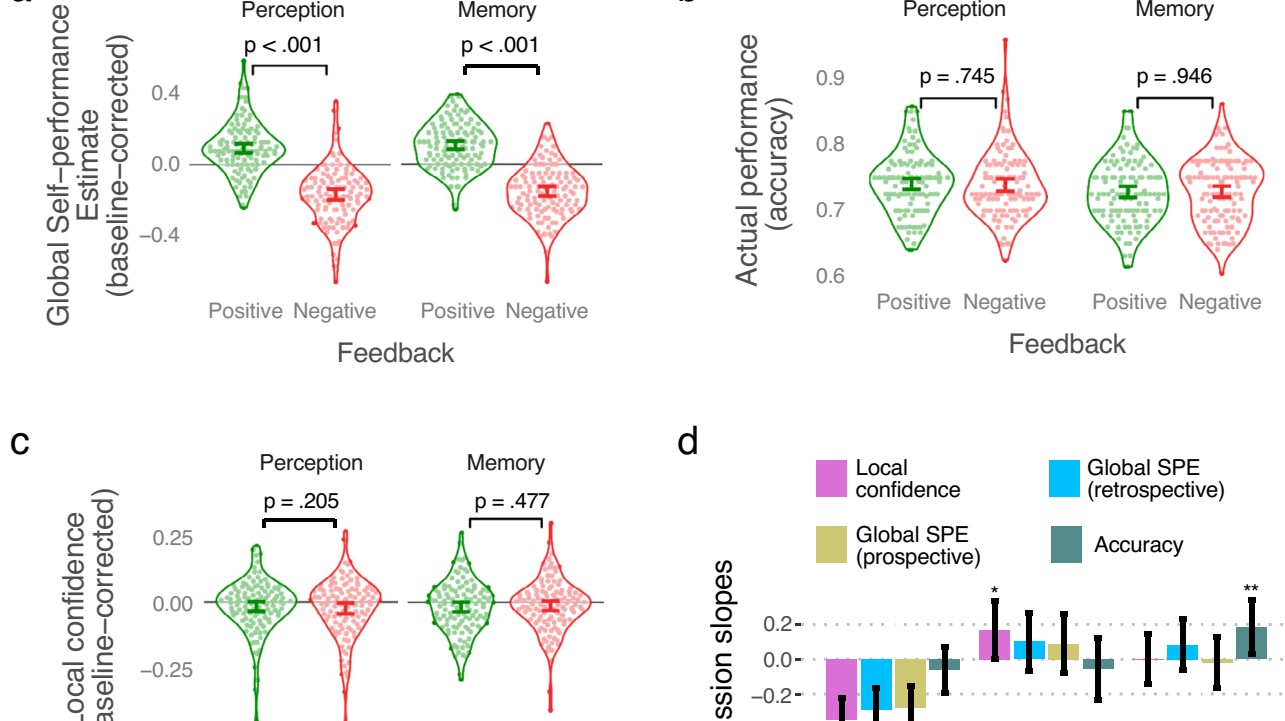

**Fig. 3 | Manipulation of global confidence by feedback and transdiagnostic symptom axes. a** Global confidence measured as self-performance estimates (global SPEs), (**b**) performance (mean accuracy), and (**c**) mean local confidence on intervention blocks, with more frequent positive (green) or negative feedback (red) plotted separately for the two tasks in Exp 1. For global SPEs and local confidence, baseline values were subtracted out for each individual. Each dot is an individual participant (N = 230). Error bars are centred on means and show bootstrapped 95% confidence intervals. **d** Regression coefficients depicting the relationship in Exp 2

(N = 287) between transdiagnostic symptom axes and mean local confidence, global SPEs measured retrospectively, global SPEs measured prospectively, and accuracy. Error bars show bootstrapped 95% confidence intervals. In all panels, statistical comparisons were performed using mixed regressions, and p-values were obtained through likelihood ratio tests with $\chi^2$ distributions. For (**a**)–(**c**), p-values were adjusted for multiple comparisons using FDR correction. ****p <0.001, **p = 0.020, *p = 0.046.

$\chi^2$ = 48.08, p <0.001, β = 0.46, CI = [0.34 0.59]; Exp 2: $\chi^2$ = 51.27, p <0.001, β = 0.46, CI = [0.34 0.58]). In contrast, after the effect of local confidence was taken into account, fluctuations in average performance across blocks no longer statistically significantly predicted global SPEs (Exp 1: $\chi^2$ = 1.03, p = 0.307, β = −0.07, CI = [−0.06 0.20]; Exp 2: $\chi^2$ = 0.24, p = 0.627, β = 0.03, CI = [−0.09 0.15]).

Finally, we examined whether our feedback intervention continued to affect local confidence on subsequent test blocks and whether these effects generalised across tasks (memory and perception). In both experiments, we found evidence that the feedback

manipulation delivered within an intervention block continued to impact local confidence estimates in the early phase of a subsequent test block, even when the test block involved a distinct task domain (perception-to-memory transfer: Exp1−t(226) = 2.19, p = 0.030, β = 0.016, CI = [−0.0005 0.032]; Exp2−t(275) = 3.12, p = 0.002, β = 0.012, CI = [0.003 0.021]; Supplementary Fig. 7; see Supplementary Information §2 for a full analysis). We also tested whether local metacognitive efficiency moderated the effect of feedback on global SPEs but did not find effects in either experiment (Supplementary Information §13).

## Distortions in the formation of global confidence in anxious-depression

We next asked how the formation of global confidence estimates related to symptoms of anxiety and depression (measured using standardised questionnaires in Exp 1, and for greater precision measured transdiagnostically in Exp 2). We replicated previous findings showing that heightened anxiety symptoms are associated with lower average local confidence (metacognitive bias; Exp 1−GAD-7: $\chi^2 = 9.76$, $p = 0.002$, $\beta = -0.03$, CI = [−0.05 −0.01]; Exp 2−Anxious-Depression axis: $\chi^2 = 29.63$, $p < 0.001$, $\beta = -0.35$, CI = [−0.47 −0.22])[2] and lower global SPEs (Exp 1: $\chi^2 = 24.11$, $p < 0.001$, $\beta = -0.05$, CI = [−0.06 −0.03]; Exp 2: $\chi^2 = 20.26$, $p < 0.001$, $\beta = -0.29$, CI = [−0.42 −0.17]; Fig. 3d; Supplementary Fig. 8; see Supplementary Information §5 for details).

We next developed distinct computational accounts of metacognitive distortions that might impact global confidence formation in individuals with anxious-depression symptoms (Fig. 1d). These accounts make qualitatively distinct predictions for how anxious-depression symptoms affect the sensitivity of global SPEs to local confidence and performance feedback (Fig. 1e−g; Supplementary Fig. 9). Specifically, we tested for the presence of three potential (non-exclusive) mechanisms: D1) Individuals with high anxious-depression symptoms have reduced sensitivity to instances of their own high (compared to low) local confidence when forming SPEs; such a mechanism blunts the positive linear relationship between local confidence and global SPEs (Fig. 1e; Supplementary Fig. 9a). D2) Individuals with high anxious-depression symptoms are more sensitive to negative compared to positive feedback when forming global SPEs, leading to asymmetries in the sensitivity to these two types of feedback (relative to no feedback trials; Fig. 1f; Supplementary Fig. 9f). D3) Individuals with high anxious-depression symptoms exhibit a (negative) response bias when rating their global SPE; such a mechanism produces an overall reduction in SPEs in the absence of distortions in learning (Fig. 1g; Supplementary Fig. 9c and g). We also constructed models characterised by combinations of these component mechanisms (D4−distortions in both confidence and feedback learning; D5−confidence learning distortion combined with response bias; D6−feedback learning distortion combined with response bias; see 'Methods' for details). Unlike models D1−D4, models D5 and D6 were not distinguishable from qualitative features of behavioural data alone but could be distinguished through model fitting (see below). We also compared models D1−D6 to a null model (D0) that did not evince differences in global SPEs with anxious-depression symptoms (Supplementary Figs. 9d and 8H; see Supplementary Information §6 for details of null model).

We next asked how these models fared in capturing the data in relation to individual variation in anxious-depression symptoms within Exps 1 and 2. Figures 4a−d show the data from Exps 1 and 2 plotted using the same conventions as Fig. 1e−g (and model simulations in Supplementary Fig. 9). In both Exps 1 and 2, we found that the slope relating higher local confidence to SPEs was blunted for individuals with higher anxious-depression scores (PHQ & GAD scores in Exp 1 and Anxious-Depression axis scores in Exp 2; Fig. 4a and b; see Supplementary Fig. 10 for similar plots for Exp 1 relative to clinical cutoffs), consistent with the qualitative pattern expected under a confidence distortion model (D1; compare Fig. 1e). This difference was confirmed statistically as a significant 2-way interaction between SPE and anxious-depression scores when predicting local confidence (specifically, within-participant z-scored local confidence values > 0; see 'Methods' and also Supplementary Fig. 9; Exp 1, GAD: $\chi^2 = 13.66$, $p < 0.001$, $\beta = -0.017$, CI = [−0.026 −0.001]; Exp 1, PHQ: $\chi^2 = 4.75$, $p = 0.029$, $\beta = -0.009$, CI = [−0.017 −0.001]; Exp 2, Anxious-Depression axis: $\chi^2 = 5.39$, $p = 0.020$, $\beta = -0.06$, CI = [−0.10 −0.01]). Next, to quantify the impact of feedback on global SPEs, we evaluated 2-way interactions between Feedback type (feedback, no feedback) and anxious-depression scores when predicting global SPEs (separately for

positive and negative feedback). When comparing positive feedback to no feedback trials, no significant interactions with symptom scores were obtained in either Exp 1 (GAD: $\chi^2 = 0.33$, p = 0.745, $\beta = 0.003$, CI = [−0.014 0.019]]; BF01 = 13.50 ± 3.4%; PHQ: $\chi^2 = 0.41$, $p = 0.523$, $\beta = -0.005$, CI = [−0.020 0.010]]; BF01 = 8.07 ± 4.5%) or Exp 2 (Anxious-Depression axis: $\chi^2 = 0.61$, $p = 0.437$, $\beta = 0.04$, CI = [−0.06 0.14]; BF01 = 11.15 ± 7.6%). Similarly, when comparing negative feedback to no feedback, no such significant interactions were obtained in either Exp 1 (GAD: $\chi^2 = 0.79$, $p = 0.374$, $\beta = 0.009$, CI = [−0.011 0.029]]; BF01 = 23.60 ± 70.9%; PHQ: $\chi^2 = 2.27$, $p = 0.132$, $\beta = 0.013$, CI = [−0.004 0.030]]; BF01 = 3.22 ± 10.5%) or Exp 2 (Anxious-Depression axis: $\chi^2 = 0.89$, $p = 0.346$, $\beta = -0.05$, CI = [−0.17 0.06]; BF01 = 7.77 ± 2.4%). Overall, this pattern of data is consistent with anxious-depression symptoms engendering a distortion in the sensitivity of global SPEs to local confidence (i.e. model D1) despite substantial-to-strong evidence for intact sensitivity to feedback.

We next pursued formal model comparisons (Fig. 4e and f). In both Exps 1 and 2, models that contained a confidence distortion either on its own (D1) or combined with either response bias (D5) or feedback distortion (D4) provided better fits to data than models without a confidence distortion. The model-estimated posteriors of the regression slope parameters relating anxious-depression symptoms to confidence distortions ($\beta_c$) and feedback distortions ($\beta_f$) from the confidence-plus-feedback distortion model (D4) are depicted in Fig. 4g and h. In both experiments, the confidence distortion parameter $\beta_c$ was significantly negative (Exp1: GAD-7, 99% HDI = [−0.037 −0.028]; Exp 1: PHQ-9, 99% HDI = [−0.048 −0.037]; Exp 2: Anxious-Depression axis, 99% HDI = [−0.101 −0.030]) while the feedback distortion parameter $\beta_f$ did not differ from 0 (GAD-7, 99% HDI = [−0.029 0.036]; Exp 1: PHQ-9, 99% HDI = [−0.034 0.035]; Exp 2: 99% HDI = [−0.030 0.370]). Fitting the confidence distortion plus response bias model (D5) or the confidence distortion only model (D1) to both experiments yielded similar values of $\beta_c$ (Supplementary Fig. 11a; see Supplementary Fig. 12 for regression parameter posteriors for all models). Finally, although DIC scores indicated that the confidence distortion plus response bias model (D5) provided a better fit to the data, the magnitudes of fitted $\beta_a$ values were extremely small in both experiments, with values two to three orders of magnitude smaller than those required to produce observed differences in global SPEs (Supplementary Fig. 11c). In addition, and contrary to our expectation, these very small values of the response bias parameter $\beta_a$ were significantly positive in both experiments (Exp 1: PHQ, 99% HDI = [1.96e-07 3.86e-05]; Exp 2: Anxious-Depression axis, 99% HDI = [4.46e-05 7.75e-04]; Supplementary Fig. 11b). Further analyses indicated that $\beta_a$ values of this small magnitude cannot be recovered in simulation and are likely to be spurious (we note that a slight bias towards preferring complex models is a known issue with DIC[27]). Taken together, these investigations indicate that the most parsimonious model of the data in both experiments was the confidence distortion-only model (D1). Such a distortion is manifest in individuals with greater anxious-depression symptoms exhibiting a blunted sensitivity to local confidence when forming global SPEs.

## Confidence, feedback, and affective self-beliefs

Finally, we asked whether our feedback intervention, which robustly modulated global SPEs independently of anxious-depression symptoms, also generalised to a more distal measure of affective self-evaluations (see 'Methods'). We first replicated earlier work showing that individuals with higher anxious-depression symptoms self-endorse more negative and fewer positive affect words compared to matched control groups[28−31] (Fig. 5a; Supplementary Information §8). We also observed a valence-specific association of self-endorsements with baseline local confidence (Exp 1: $\chi^2 = 12.91$, $p < 0.001$, $\beta = -1.04$, CI = [−1.60 −0.47], Supplementary Fig. 13d; Exp 2: $\chi^2 = 48.23$, $p < 0.001$, $\beta = -1.68$, CI = [−2.15 −1.20], Fig. 5b) and baseline global SPEs (Exp 1:

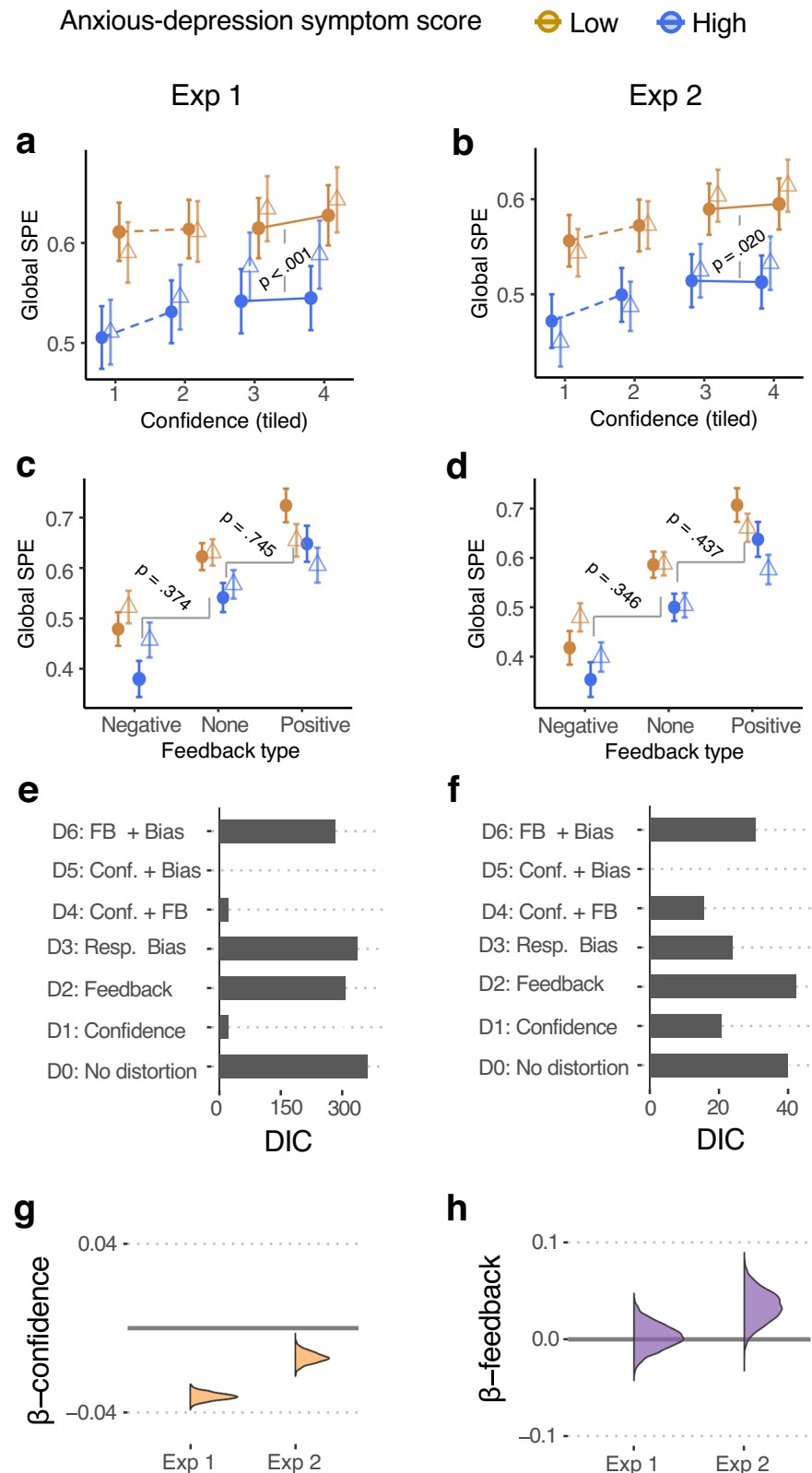

χ² = 44.79, p < 0.001, β = −2.14, CI = [−2.77 −1.52], Supplementary Fig. 13e; Exp 2: χ² = 5.63, p = 0.018, β = −0.55, CI = [−0.99 −0.10]; Supplementary Fig. 14c), with a positive relationship between confidence and self-endorsement of positive words, and a negative relationship between confidence and self-endorsement of negative words.

In Exp 2, we presented a first set of words at the beginning of the study and a second set immediately following the first intervention and

test blocks. Our key question was whether the type of performance feedback received during the intervention block would impact subsequent effective self-evaluation. Consistent with this hypothesis, we observed a significant interaction between Feedback type and Word valence (positive, negative) on the change in self-endorsements (Fig. 5c; χ² = 6.08, p = 0.014, β = 0.11, CI = [0.02 0.20]). Participants who received negative performance feedback during intervention

**Fig. 4 | Data and model fits show distorted learning from local metacognition.** Results from Exp 1 (left column (**a**, **c**, **e**)); $N = 230$) and Exp 2 (right column (**b**, **d**, **f**); $N = 287$). Mean global self-performance estimates (SPEs) and 95% confidence intervals depicted in (**a**–**d**) are obtained from linear regression models after regressing out random effects of participant, block number, and trial number from observed data (filled circles) and model fits (unfilled triangles). **a**, **b** Global SPEs plotted against local confidence (z-scored and split into 4 quantiles per individual) separately for individuals belonging to the lowest (gold) and highest (blue) tertiles of anxious-depression symptom scores (obtained using the GAD-7 questionnaire in Exp 1 and transdiagnostically in Exp 2). Depicted statistical comparisons are for slopes of higher confidence values between higher and lower anxious-depression scores. **c**, **d** Global SPEs plotted for blocks with more frequent negative feedback, no feedback, and more frequent positive feedback, separately for individuals with low and high anxious-depression scores. **e**, **f** Deviation Information Criteria (DIC) scores for the seven possible models of how different distortions in global SPE formation relate to AD symptom scores: D0−No distortion, D1−Confidence distortion only, D2−Feedback distortion only, D3−Response bias only, D4−Confidence and feedback distortions, D5−Confidence distortion and response bias, D6−Feedback distortion and response bias. Posterior distributions of regression slope parameters relating anxious-depression symptoms to (**g**) confidence and (**h**) feedback distortions for model D4 in Exp 1 and Exp 2. Regression coefficient values from Exp 2 are divided by 5 (corresponding to the difference in the range of GAD and Anxious-Depression axis scores) to allow direct comparison of coefficients.

blocks showed greater self-endorsements of negative compared to positive words ($t(593) = 4.21$, $p < 0.001$, $\beta = −0.09$, CI = [−0.14 −0.04]). There was no such difference in the change in self-endorsement following positive performance feedback ($t(593) = 0.65$, $p = 0.516$, $\beta = −0.02$, CI = [−0.06 0.04]). Overall, these results indicate that feedback interventions that alter global SPEs also impact affective self-evaluation.

## Discussion

Distortions in metacognition and self-beliefs represent a central feature of mental ill health[21,32]. Individuals high in subclinical anxious-depression symptoms tend to be chronically underconfident in their abilities both when evaluating their confidence on a particular task instance[2–5] and when providing global self-performance estimates (SPEs)[6]. Here we asked how local feedback and confidence are integrated into global SPEs, and how such integration is distorted by subclinical anxious-depression symptoms. We show that individuals with higher anxious-depression symptoms have insufficient sensitivity to instances of higher local confidence when forming their global SPEs. In other words, a subtle but pervasive source of global underconfidence may be a failure to notice one's own instances of high (local) confidence. Notably, however, this asymmetry in learning was not observed on trials in which participants received explicit feedback. Taken together, our findings indicate that global underconfidence is rooted in a failure to integrate local and global aspects of metacognition, rather than due to a generalised bias in learning.

Past work has proposed that global SPEs are formed by combining local confidence on individual trials with external feedback about performance[2,8,17,19]. Here we artificially created situations in which participants received predominantly negative feedback, or predominantly positive feedback, despite their performance being equated in both cases. Such situations mimic real-world scenarios in which teachers or supervisors may either be stern, emphasising negative feedback following mistakes, or encouraging, focusing on positive feedback following successes. In both experiments, we observed strong effects of such feedback asymmetries on global SPEs in both perception and memory task domains. By demonstrating a robust impact of feedback on global confidence, we extend previous work depicting a malleability of confidence through affective induction[12,13], expected reward[14,33] and feedback[15,16]. Our participants also self-reported that feedback induced valence-specific changes in mood (Supplementary Information §11.2), and it is possible that the impact of feedback on confidence may be mediated by shifts in mood[13].

We also extend upon a previously proposed computational model[8] of global SPEs by introducing parameters governing an asymmetry in learning from positive vs. negative feedback and high vs. low local confidence. This extension is reminiscent of similar work modelling asymmetries in learning from external reinforcement or information[34–38]. Our model and its ability to fit empirical data open a rich framework within which to understand and characterise a range of distortions in the formation of metacognitive beliefs.

Individuals scoring high in anxious-depression symptoms and patients diagnosed with clinical depression have been shown to exhibit greater sensitivity to negative compared to positive information, particularly when this information is self-relevant[20–22,29]. Recent work shows that this asymmetry may relate to a deficit in learning, where depressed individuals learn more from negative and less from positive information[23–25,39]. We tested if similar distortions existed both with regard to incorporating feedback and/or confidence when forming global confidence. We also considered an alternative account of distortions in global confidence, in which greater anxious-depression symptoms lead to (negative) metacognitive response biases rather than acting through changes in learning. In both experiments, we found evidence for a distortion in learning from local confidence—individuals with higher anxious-depression scores had blunted sensitivity to higher local confidence when forming their global confidence estimates. In contrast, no distortion was observed in relation to learning from positive compared to negative feedback—suggesting that asymmetries in learning were specific to metacognitive variables (the link between local and global confidence).

One implication of intact feedback processing in individuals with high anxious-depression symptoms is that asymmetric feedback schedules may be able to 'override' metacognitive biases that would otherwise exist in the absence of feedback. However, such an intervention would only be practically meaningful were it to be persistent and generalisable across tasks, and potentially to other more distal metrics of self-evaluation. In both experiments, we found that an impact of feedback on confidence can persist beyond intervention blocks to also affect confidence on 'test' blocks without feedback: during test blocks, local confidence was higher when the previous intervention blocks had more frequent positives compared to negative feedback. Interestingly, this effect of feedback on test block confidence was observed specifically when feedback was delivered on a perception task and confidence assayed on a memory task (and not vice-versa), and most prominently during the early phase of a subsequent test block. These results indicate that receiving feedback on some domains may have broader, domain-general, effects on people's confidence while remaining more restricted in other domains.

Our model-based approach to estimating the influence of feedback on confidence can also be used to simulate an expected change in metacognition under different feedback schedules for different individuals—highlighting a potential for developing personalised interventions to ameliorate systematic biases in over- and underconfidence[2,5,32,40,41]. As a step in this direction, we investigated if our feedback intervention impacted affective self-evaluations known to be characteristic of mental health disorders, particularly anxious-depression[28–31]. We found that asymmetries in feedback not only robustly shifted global confidence, but also led to significant changes in self-endorsement of positive vs. negative words. However, these changes in self-endorsement were predominantly observed as a reduction in self-beliefs following negative feedback blocks, and not as heightened self-beliefs following positive feedback blocks. Thus, the

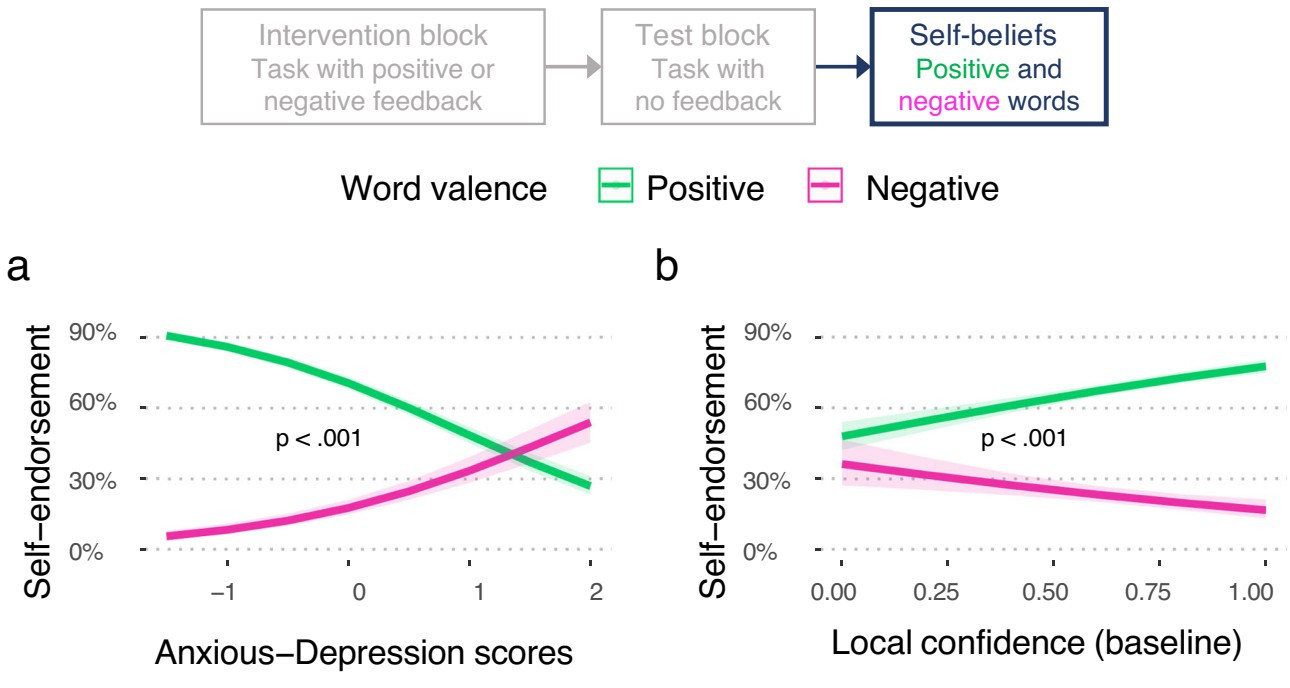

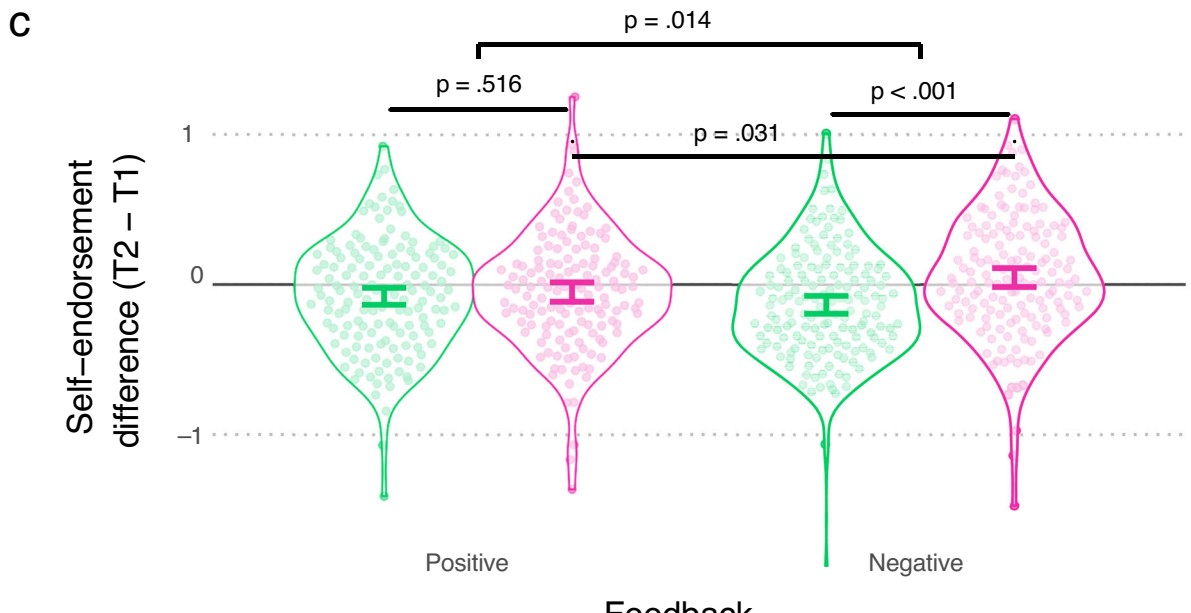

**Fig. 5 | Metacognition and its relation to affective self-evaluation.** The interaction of word valence (positive or negative) with (**a**) Anxious-Depression transdiagnostic scores ($p < 0.001$), and (**b**) mean local confidence ($p < 0.001$) during baseline runs when regressed upon self-endorsements at the first timepoint (T1) of the self-referential encoding task in Exp 2 (plotted as marginal effects from a logistic regression predicting self-endorsement). **c** Change in self-endorsement from T1 to T2 (i.e. post-feedback) separated by word valence and the type of feedback delivered during intervention blocks. While there was no difference in self-endorsements following positive feedback, negative words were endorsed more than positive words following negative feedback. For all panels, error bars show 95% bootstrapped confidence intervals, statistical comparisons were performed using mixed regressions, and $p$-values were obtained through likelihood ratio tests with $\chi^2$ distributions. For (**c**), error bars are centred at the respective means and $p$ values were adjusted using FDR.

potential of metacognitive interventions for ameliorating negative affective self-beliefs remains unclear. Future work could build on our study to more carefully calibrate feedback schedules to test for the possibility of positive changes in self-beliefs (e.g. by manipulating the frequency of positive feedback, the timescale of the intervention, and/or through the use of multiple tasks or social environments).

One limitation of the present study is that, by design, the number of intervention trials available to determine distortions in learning from feedback was substantially fewer than the trials used to estimate distortions in learning from confidence. However, several reasons lead us to believe that our observation of intact feedback learning is robust, rather than reflecting a lack of power to detect asymmetries. First, both

positive and negative feedback interventions had strong and systematic impacts on global SPEs, despite being driven by a small number of critical trials. Second, both regression analyses and computational model fits were performed hierarchically, such that data were pooled over participants when estimating learning parameters, leading to 4600 and 5560 (positive and negative) feedback trials in Exp 1 and Exp 2 respectively. Third, model-neutral Bayes Factor analysis showed at least moderate (and in some cases strong) evidence for a lack of an effect of symptom scores on feedback learning in both datasets. Fourth, and most importantly, simultaneously introduced variation in both ground-truth feedback and confidence distortion regression parameters were recoverable from our model in simulations using a similar number of trials employed in our experiment. Another limitation is that we used a general population sample from an online participant recruitment platform. While the sample contained a substantial number of individuals who surpassed clinical cutoffs on standard anxious-depression scales, further work is required to test if our findings also generalise to individuals with clinical diagnoses of anxiety and/or depression[42].

In summary, we identify a potential mechanism for why people maintain a global sense underconfidence in the face of intact performance. People with greater subclinical anxious-depression symptoms exhibited a blunted sensitivity to their own instances of high local confidence, creating a maladaptive loop in which underconfidence is maintained despite otherwise good performance. Notably, these distortions were not observed when learning from explicit feedback, pointing to the potential of calibrated feedback interventions in ameliorating metacognitive biases. More broadly, our work provides a principled computational framework within which to understand the origins of metacognitive distortions across a spectrum of psychopathology.

## Methods

### Participants

We recruited participants through the online platform Prolific (prolific.co) and included individuals between ages 18 and 55 years who reported being fluent in English. Participants were recruited in two non-overlapping samples—Exp 1 (exploration) and Exp 2 (validation; preregistered hypotheses and analyses plan uploaded to osf.io/7xfqw on 20th January 2023). After dropouts and exclusions (see Supplementary Methods for details along with sample justification), a total of 230 participants remained for Exp 1 (age mean and SD = 32 ± 9; gender: 127 females, 185 males, 2 nonbinary) and 278 participants for Exp 2 (age mean and SD = 32 ± 9; gender: 176 females, 278 males, 3 nonbinary).

### Design

Our key manipulation was the extent to which individuals received (veridical) positive or negative feedback about their decision at the end of each trial (after they provided a confidence estimate). This manipulation involved experimentally controlling the probability with which feedback was delivered across blocks, with some blocks delivering more positive than negative feedback, and others delivering more negative than positive feedback. Blocks were designated as baseline (blocks 1 and 2; no feedback), intervention (blocks 3 and 5; with feedback), and test (blocks 4 and 6; no feedback). Feedback type was manipulated as a within-participant factor. We also manipulated two between-participant factors: 1) Intervention task (perception or memory) and 2) Transfer type (whether test blocks followed an intervention block of the same or opposite task). Because Feedback type was a within-participant factor, we controlled for order effects by manipulating Feedback order between individuals (i.e. whether positive or negative feedback blocks were encountered first). This resulted in eight groups of participants for Exp 1. Figure 2a shows the order of task blocks (perception and/or memory) for each of the eight groups. Participants assigned to groups transferring to the same task repeated

that task in the two baseline blocks while others performed one block of each task whose order was randomised across participants.

Exp 2 had a similar design to Exp 1, except we no longer manipulated Transfer type. Instead, test blocks always consisted of the opposite task to the one used during the intervention block, where we sought to replicate findings in Exp 1 of cross-domain transfer in the effects of feedback on confidence. We assessed whether the effects of feedback on confidence modulated affective self-beliefs. To this end, participants self-endorsed two sets of 20 affective words (details below), once, before beginning the confidence tasks, and again after the first test block. The order of the two sets of words was counterbalanced across participants. Thus, Exp 2 also had eight groups of participants (Supplementary Fig. 1).

### Tasks

**Perception and memory tasks.** The tasks were embedded in a gamified environment. Participants were informed they were helping people of Fruitville whose livelihood depends on harvesting and packaging fruits. The perception task required participants to decide which of two types of berries—raspberries or blackberries—were more numerous, to aid the farmers in deciding which berry to harvest. The memory task required participants to decide which fruit was present in a box of fruits that was recently opened in front of them, to help the fruit packers correctly label the contents of that box. Participants were also asked to report their confidence in their choices and told that their reported confidence would help people of Fruitville adjust how much to rely on their advice. To provide a natural rationale for the intermittent feedback schedule, participants were instructed that occasionally Fruitville would hire an 'auditor' who would evaluate participants' choices and give them feedback as to whether their choice was correct or incorrect. The stimulus background was a cartoon-like nature scene and participants were instructed, in both tasks, to attend to a green bush in the centre within which all stimuli would appear.

Figure 2b illustrates the sequence for one trial of each of the perception and memory tasks. In Exp 1, each perception or memory task block consisted of 40 trials. In Exp 2, perception and memory task blocks also consisted of 40 trials except for the two test blocks, which consisted of 20 trials (we reduced the trial number in Exp 2 after observing that cross-task transfer of feedback to confidence in Exp 1 was largely restricted to the first half of each test block).

The perception task comprised a density estimation task, as used in previous psychophysical studies of perceptual metacognition[2,43]. On each trial, after initially attending to the blank central bush for a duration of 600–750 ms, 121 berry stimuli appeared at random non-overlapping locations within the area of the central bush. Some of the berries were red (raspberries) while others were dark purple (blackberries). The berries appeared on the screen for 1000 ms and were randomly replotted within the bush every 250 ms to make the stimulus dynamic and engaging and preclude explicit counting strategies. One of the two types of berries was always greater in number. 200 ms after stimulus offset, participants were shown one of each type of berry and asked to choose, by pressing the left or the right arrow keys, which they thought were presented in greater numbers. Once a choice was made, the chosen berry increased in size and a vertical slider bar appeared in the centre of the screen. Participants were then instructed to 'Click/Drag the slider to select confidence in your choice' with their mouse. After the appearance of the slider, and before selection of a confidence value, participants had the option to change their berry choice by pressing the opposite arrow key. Once the slider was clicked, 5 confidence markers (none, low, medium, high, full) appeared to the left of the slider, along with a continuous percentage between 0 and 100% to the right of the slider. After reporting their confidence, participants were asked to end the trial by pressing one of the arrow keys. On selected trials of the intervention blocks, participants then saw a

message notifying them that, 'The auditor of Fruitville is here to evaluate your response. Check how you performed.' Participants then pressed a button and were informed if they were correct or incorrect. The correct and incorrect feedback text was accompanied by a smiley or slightly frowning face respectively, together with text randomly chosen from one of three (positive or negative) feedback messages (see Supplementary Methods). To measure global SPEs, at the end of the block (i.e. retrospectively), participants were shown a slider and asked to indicate how many trials they believed they answered correctly on that block.

The memory task was a visual working memory task. Each trial began with participants attending to the central bush for a variable duration of 500–600 ms, followed by the presentation of several different fruits (between 1–12 fruits) within the area of the bush for 1500 ms. After stimulus offset, there was a delay of 1000 ms following which participants were then shown two fruit options, one of which was present in the previous stimulus set and one that was not. Participants were asked to indicate the fruit that was in the previous set. They then reported their local confidence, received feedback, and provided global SPEs in the same manner as described above for the perception task.

The difference between the number of raspberries and blackberries for the perception task, and the number of fruits displayed for the memory task (the set size), were each adjusted from trial to trial using a 1-up-2-down staircase. We calculated the probability of giving positive and negative feedback on intervention blocks based on the expected ~71% accuracy rate associated with such a staircase. Out of the 40 trials, we aimed to provide feedback on 9 correct trials and 1 incorrect trial on positive feedback blocks and 9 incorrect trials and 1 correct trial on negative feedback blocks. Feedback for each trial was thus delivered with the probabilities of .3169 (=9/(40*0.71); correct trial) and 0.0862 ( = 1/(40*(1 − 0.71)); incorrect trial) on positive feedback blocks, and 0.0352 (correct trial), and 0.7759 (incorrect trial) on negative feedback blocks.

**Self-referential encoding task.** Participants also completed a self-referential encoding task (SRET). Exp 1 involved a single instance of the SRET presented before the perception and memory task blocks. Here, participants were asked to self-endorse 42 words (20 pleasant or positive affect adjectives, 20 unpleasant or negative affect adjectives and 2 catchwords; words from ref. 44; Supplementary Fig. 15). Words were shown on the screen (in random order) for 1 s, and participants were asked, 'Does this word describe you?' They then selected one of two options, 'Yes' or 'No.' After responding to all the words, participants were asked further questions about their Prolific ID, gender, and how many online studies they had completed in the last 24 h and last month. This was followed by a surprise memory test where they were asked to recall as many words as possible from the self-referential encoding task phase (data not analysed here).

In Exp 2, participants were administered the SRET at two-time points—before and after the performance feedback intervention. Positive and negative affect words from Exp 1 were split into two sets of 10 positive, 10 negative and 1 catchwords, with one set presented at each time point. Words were split based on how well they predicted individual global self-performance estimates on perception and memory tasks during the baseline trials of Exp 1, such that both sets of words had roughly similar relationships with global confidence (Supplementary Fig. 15). With this design we aimed to control for any non-specific differences in the sensitivity of the two-word sets to changes in global confidence. For the SRET in Exp 2, we sought a continuous answer to the question 'How much does this word describe you?' Participants responded using a slider with 5 equidistant markers, 'Not at all,' 'A little', 'Somewhat', 'A good amount' and 'Very much.' Each word was presented for 1.5 s and responses were self-paced.

## Procedure

The entire study was programmed using the Phaser 3 game framework for JavaScript to provide a gamified look and feel and hosted on the online platform Pavlovia. Participants from Prolific were given the study link where they were first required to read and agree to a Study Information and Consent Form before proceeding. Each participant then proceeded through multiple self-paced phases of the study. Participants were paid £7.50/h for completing the study, or on a pro-rated basis if they chose to leave the study at any point. All study procedures were approved by the UCL Research Ethics Committee; approval number 21029/001.

The first phase comprised the self-referential encoding task. This included instructions about how to perform the task along with four practice words, the 42 test words (21 in Exp 2), and the surprise memory test (only in Exp 1). The next phase comprised the Fruitville game. This commenced with extensive step-by-step instructions on how to perform one or both tasks (depending on the participant's group) along with a description of the game scenario and 5 training trials. Participants were given the option to repeat the instructions and training trials as many times as they wished. In Exp 2, instructions for each task were followed by a prospective global SPE where participants were asked to estimate how many trials they thought they would respond to correctly out of a 100 on that task as an estimate of their prospective global SPE. Instructions (and in Exp 2, the prospective global SPE) were followed by 50 practice trials of that particular task, which also served the purpose of initiating the staircase. From the practice trials, we used the mean value of the last 5 staircase reversals for each task as the starting difficulty level of the first block of that task for the main part of the study, which involved participants performing six task blocks. Then, to ensure stable performance despite potential learning effects, the staircase procedure was continued through the six blocks where we again used the mean of the last 5 reversals (or fewer, if there were fewer reversals) from each block as the starting level for subsequent blocks of respective tasks. Figure 2a and Supplementary Fig. 1 show sequences of the blocks for different groups in Exp 1 and Exp 2 respectively. In Exp 2, the second block of SRET occurred after block 3 of the perception/memory task.

In the final phase, participants completed several mental health questionnaires. For Exp 1 this included the PHQ-9 for depression[45], GAD-7 for general anxiety[46] and the 3-item mini-SPIN for social phobia[47]. For Exp 2, this included a set of 71 questions[48] designed to allow efficient determination of scores along three transdiagnostic mental health axes[49]—Anxious-Depression (A-D), Compulsivity and Intrusive Thought (CIT), and Social Withdrawal (SW).

Across the entire study, there were three 'catch' questions designed to detect participants who may not have been paying sufficient attention. Two catch questions were included during the self-referential encoding task, and one was included within the mental health questionnaires.

## Statistical analyses

Model-free data analysis and figure generation were performed in R[50]. One set of analyses tested effects of performance feedback interventions on subsequent global SPEs, accuracy and local confidence. For these analyses, we baseline-corrected global SPEs, confidence and accuracy by subtracting baseline block SPEs, mean local confidence and mean accuracy respectively (we use variables with the subscript $bc$ for baseline corrected values below). This transformation effectively normalised the bounded SPE, accuracy and confidence data and allowed the use of linear regression models for analyses (these transformed variables are still bounded between [−1, 1] but the normalisation zero-centred their distributions with no extreme values near the bounds). As these analyses were performed at a within-participant level, we used linear mixed regression models (LMM) implemented by the *lmer* function in the *lmerTest* package (version 3.1-3). Models were

tested for assumptions of homoscedasticity and normality of residuals. Relevant random intercepts and slopes were used for all models, which were sequentially reduced if they did not explain sufficient variance in the data, as indicated by non-converging model fits—though before excluding random slopes we ensured that relevant main or interaction effects regressed upon non-independent data points (e.g. where trials or blocks were repeated) were still significant when those slopes were retained in the (non-converging) model.

For testing differences in SPE on feedback blocks with Feedback type (positive, negative), we included 3- and 2-way interactions with Task (perception, memory) and Feedback order (positive first, negative first) along with main effects of Accuracy and Staircase level, as follows:

$$SPE_{bc} \sim Feedback\_type * Task * Feedback\_order$$
$$+ Accuracy_{bc} + Staircase\_level_{bc} + (1|Participant) + (1|Group)$$

For control analyses, to test if accuracy and staircase level were modulated by Feedback type we used the following LMMs (a similar model was used for Staircase level):

$$Accuracy_{bc} \sim Feedback\_type * Task + (1|Participant) + (1|Group)$$

For testing, if confidence and accuracy predicted global SPEs, we used all 6 task blocks with the following LMM:

$$SPE_{bc} \sim Local\_confidence_{bc} + Accuracy_{bc}$$
$$+ (1 + Local\_confidence_{bc} + Accuracy_{bc}|Participant)$$
$$+ (1|Task) + (1|Group)$$

For measuring baseline associations of mental health scores with confidence (both local confidence and global SPEs) and accuracy, we used confidence values from the baseline blocks (mean value for local confidence) and entered these into the following LMM:

$$Confidence \sim MH\_score + Age + Gender + Accuracy + Staircase\_level$$
$$+ RT\_Zscored + (1 + MH\_score|Task)$$

Analyses were performed using linear mixed regression models while ensuring that assumptions of linear models were satisfied. For entry into linear models, we z-scored confidence and accuracy values to transform them to a continuous scale. For Exp 1, separate regression models for PHQ and GAD and were used in place of MH_score, while for Exp 2, we entered all transdiagnostic dimensions simultaneously such that MH_score = AD + CIT + SW.

For a model-free test of distortions in learning from local confidence, we asked if the linear relationship between local confidence and global SPEs differed as a function of anxious-depression scores. Simulations showed that a blunting of this relationship was particularly pronounced at intermediate-to-high levels of local confidence (i.e. z-scored confidence > 0; Supplementary Fig. 8a). We tested this distortion as a 2-way interaction between SPE and MH_score in predicting (within-individual) z-scored local confidence using the following linear mixed model:

$$Local\_confidence\_Zscored \sim SPE * MH\_score + Accuracy$$
$$+ (1 + SPE * MH\_score|Trial\_number)$$
$$+ (1|Block\_number) + (1|Participant)$$

Note that here we predicted local confidence from SPE and not vice-versa as there were a larger number of data points for local confidence to aid in statistical analyses. The validity of this effect was confirmed in simulation.

For a model-free test of distortion in learning from feedback, we asked if the reduction in global SPEs was greater following feedback vs. no feedback blocks, which we estimated separately for positive and negative feedback (against no feedback):

$$SPE\_Zscored \sim Feedback\_type * MH\_score$$
$$+ (1 + Feedback\_type * MH\_score|Participant)$$
$$+ (1 + Feedback\_type * MH\_score|Block\_number)$$

To test for a lack of feedback distortion (i.e. evidence in favour of the null hypothesis), we calculated Bayes Factors using the *BayesFactor* package (version 0.9.12-4.7) in R for the Feedback_type by MH_score interaction. This was done by comparing the above regression model with one where the interaction term was removed.

To test if confidence (local and global) or mental health scores predicted self-endorsement of valenced (positive/negative) words, we used the following generalised linear regression model with a binomial link function:

$$Word\_endorsement \sim MHQ\_score[/Confidence] * Word\_valence$$
$$+ (1|Word\_number)$$

For Exp 1, Word_endorsement included all presented words, while for Exp 2, endorsements were only used from the (pre-feedback) baseline measurement.

To test if feedback impacted self-endorsement of positive and negative words, we used the following regression model:

$$Word\_endorsement\_diff \sim Feedback\_type * Word\_valence$$
$$+ (1|Word\_set) + (1|Participant) + (1|Task)$$

Here Word_endorsement_diff was computed as the difference in mean word endorsements for positive and negative words after compared to before feedback. There were two sets of words (Word_set) presented to participants before and after the intervention block, with order counterbalanced across participants.

For the LMMs, $p$ values were calculated from likelihood ratio tests ($\chi^2$ distribution with one degree of freedom) by comparing the likelihood of the model with the relevant effect term to a reduced model from which that term was removed. Any 2- or 3-way interactions also included main effects of all terms in those interactions. Any 2- or 3-way interactions that were not significant at an alpha = 0.05 were excluded from models and models were re-evaluated without those interactions (by first reducing 3-way and then 2-way interactions). All tests were two-tailed.

For between-participant analyses of relationships between local/global confidence on baseline blocks, mental health scores and SRET self-endorsements, we used the *lm* function in the *stats* package (version 4.1.0) in R. For mediation analyses, we used the *mediation* package (version 4.5.0) in R. Marginal effects obtained from regression models were plotted using the *plot_model* function of the *sjPlot* package (version 2.8.11) in R. Cohen's d was calculated using the *effectsize* (version 0.8.3) toolbox in R.

To create plots of behavioural data in Fig. 4a, b, we used the following procedure. First, we regressed the interaction between Anxious-Depression scores and z-scored local confidence (ConfZ) upon global SPEs separately for high local confidence (ConfZ > 0) and low local confidence (ConfZ ≤ 0) trials while regressing out random effects of Participant, Block number, and Trial number. Next, we divided ConfZ values within each participant into four quantiles (i.e. each containing equal probability mass) and calculated the centre local confidence value for each quantile by taking the mean of ConfZ within each quantile across participants. Finally, we used the regression model to predict global SPEs and their 95% confidence intervals at these four centre ConfZ values (using the *emmeans* function in R). This process of standardizing and discretizing the x-axis (local confidence) across participants aided visualization and facilitated the construction of appropriate

confidence intervals for overlaying the data and model fits. Note that discretized local confidence bins were only used for visualisation purposes; all statistics were obtained from analyses of continuous confidence values. Similarly, for plots in Fig. 4c, d, we obtained SPE values from a model where the interaction of Feedback type and Anxious-Depression scores was regressed upon global SPE scores with random effects of Participant and Block number regressed out.

## Computational models

We modelled the dynamics of global SPEs by building on a model developed by Rouault et al.[8]. This model maintains Beta distributions over expected success for each task as a proxy for global self-performance estimates. A Beta distribution is characterised by $a$ and $b$ parameters, with higher values of $a$ tending to 'pull' the distribution towards 1 (higher SPE) and higher values of $b$ pulling it towards 0 (lower SPE). Higher values of both $a$ and $b$ result in higher precision, and more certainty around a particular SPE. In Rouault et al.'s[8] model, for trials with explicit feedback, Beta parameters are updated trial-by-trial as in Eqs. (1) and (2):

$$a_{t+1} = \begin{cases} a_t + 1 \ if \ correct \\ a_t \ if \ incorrect \end{cases} \tag{1}$$

$$b_{t+1} = \begin{cases} b_t \ if \ correct \\ b_t + 1 \ if \ incorrect \end{cases} \tag{2}$$

When feedback is provided on all trials, this algorithm naturally leads to mean of the Beta distribution (the mean SPE) converging on one's true underlying probability correct, with increasing precision as more data (trials) are acquired. In contrast, when participants do not get feedback, the best information they have about their performance is their local confidence estimate. Rouault et al. proposed local confidence can be used as a proxy of the probability of a correct response on a given trial. Thus, in the absence of feedback, the $a$ and $b$ parameters are updated by reported confidence (normalised to 0–1) as in Eqs. (3) and (4):

$$a_{t+1} = a_t + conf_t \tag{3}$$

$$b_{t+1} = b_t + (1 - conf_t) \tag{4}$$

In the present study, we were interested in a potential asymmetry in how global SPEs are updated by positive vs. negative feedback, and/or high vs. low confidence, and whether such asymmetries relate to anxious-depression symptoms. Thus, we modified the above model by introducing parameters that controlled the asymmetry in updating global SPEs from 1) positive and negative feedback ($\Delta LR_f$), and 2) high and low confidence ($\Delta LR_c$), as follows.

In the presence of feedback (Eqs. (5) and (6)):

$$a_{t+1} = \begin{cases} a_t + (1 + \Delta LR_f) \ if \ correct \\ a_t \ if \ incorrect \end{cases} \tag{5}$$

$$b_{t+1} = \begin{cases} b_t \ if \ correct \\ b_t + (1 - \Delta LR_f) \ if \ incorrect \end{cases} \tag{6}$$

In the absence of feedback (Eqs. (7) and (8)):

$$a_{t+1} = a_t + conf_t * (1 + \Delta LR_c) \tag{7}$$

$$b_{t+1} = b_t + (1 - conf_t) * (1 - \Delta LR_c) \tag{8}$$

In these equations, the $\Delta LR$ parameters control a boost in learning rate on correct/high confidence trials relative to a boost in learning rate on incorrect/low confidence trials. Thus, if participants learn equally from both trial types, $\Delta LR$ should be zero. These asymmetry parameters could be similar or different for the two tasks (perception and memory). We compared different versions of the model in terms of goodness of fit (using Deviation Information Criteria; DIC): 1) $\Delta LR = 0$; global SPEs are updated equally from positive vs. negative feedback, and high vs. low local confidence, 2) $\Delta LR \neq 0$; the same distortions are applied for the two tasks; the same distortions are applied to both feedback and local confidence (1 parameter), 3) $\Delta LR \neq 0$; the same distortions are applied for two tasks; distortions are allowed to differ between feedback and local confidence (2 parameters), 3) $\Delta LR \neq 0$; distortions are allowed to differ for the two tasks; the same distortions are applied for feedback and local confidence (2 parameters), and 4): $\Delta LR \neq 0$; distortions are allowed to differ across tasks; distortions are allowed to differ between feedback and local confidence (4 parameters).

For model fitting, the initial values of $a$ and $b$ were determined by allowing free parameters for the mean ($\mu_0$) and variance ($v_0$) of the Beta distributions for the two tasks. Beta parameters $a$ and $b$ are defined by mean and variance as in Eqs. (9) and (10):

$$a_0 = \frac{\mu_0}{v_0} * (\mu_0 - {\mu_0}^2 - v_0) \tag{9}$$

$$b_0 = \frac{(1 - \mu_0)}{v_0} * (\mu_0 - {\mu_0}^2 - v_0) \tag{10}$$

The participant's reported global SPE at the end of a given block $B$ is used to fit Beta parameters on the final trial $n$ of each block as in Eq. (11):

$$SPE_B \sim Beta(a_n^B, b_n^B) \tag{11}$$

The best fitting model from those listed above served as the 'no distortion' model used to investigate the influences of anxious-depression symptoms on learning (D0; see Supplementary Information §6), which we then extended to test for the presence of several potential cognitive distortions impacting global confidence formation in people with high anxious-depression symptoms. When forming global confidence, we considered that high anxious-depression symptoms could lead to D1) higher sensitivity to negative vs. positive feedback trials, D2) higher sensitivity to low vs. high local confidence, or D3) biases in reporting lower SPEs at the end of the block without differing in learning rates. We adjudicated between these potential mechanisms by regressing individual mental health scores ($MH_i$) upon learning rate asymmetry parameters $\Delta LR_f$ (for D1), $\Delta LR_c$ (D2), or global SPE values (D3), and estimated the corresponding regression slopes $\beta_f$, $\beta_c$ and $\beta_a$ respectively, as in Eqs. (12), (13) and (14):

$$\Delta LR_f = \Delta LR_{f0} + \beta_f * MH_i \tag{12}$$

$$\Delta LR_c = \Delta LR_{c0} + \beta_c * MH_i \tag{13}$$

$$SPE_B = \mu_n^B + \beta_a * MH_i \tag{14}$$

Here, $\mu_n$ denotes the Beta mean computed after updating the last trial as in Eq. (15):

$$\mu_n^B = \frac{a_n^B}{a_n^B + b_n^B} \tag{15}$$

The regression models deviate from those noted in the preregistration document in two ways: 1) at the time of preregistration we

did not consider the response bias model D3 as an alternative to our key hypotheses regarding feedback and confidence learning distortions; and 2) in our preregistered methods, we only considered the regression slope (β) terms, and erroneously omitted the intercepts terms for Models D1 and D2 (i.e. $\Delta LR_{f0}$ and $\Delta LR_{c0}$). Simulations showed that the absence of an intercept term can introduce artificially significant effects that load on the slope term. Indeed, after including the intercept terms, we no longer observed $\beta_f < 0$ for Exp 1 as stated in preregistered Hypothesis 2. However, consistent with the same preregistered hypothesis, we still observed $\beta_c < 0$.

As anxious-depression symptoms are known to have a significant relationship with baseline local confidence[2], we sought to ensure this relationship did not confound interpretation of biases in global SPE formation by z-scoring local confidence within each participant prior to analysis.

Models were fit using MCMC sampling (3 chains of 2000 samples with 1000 burn-in samples) implemented by the JAGS toolbox (version 3.4.1) in MATLAB (Mathworks Inc.; 2022b). All free parameters were inferred at the group level. The three regression slope parameters− $\beta_f$, $\beta_c$ and $\beta_a$ – were fully recoverable in simulation (Supplementary Fig. 16). We note that recovery of the confidence distortion regression parameter was more precise compared to the feedback distortion regression parameter across a range of intercept values of confidence and feedback asymmetry. This is likely due to the difference in number of trials used for estimating these two distortions. It is thus possible that our model may be insensitive to small $\beta_f$ values. However, simulations showed that these small $\beta_f$ magnitudes were unable to explain observed relationships between global SPEs and anxious-depression scores. All models were run using uninformative priors.

**Reporting summary**
Further information on research design is available in the Nature Portfolio Reporting Summary linked to this article.

## Data availability
The aggregated data for reproducing all results and figures used in this study are publicly available at Zenodo[51] (https://doi.org/10.5281/zenodo.14244645) and GitHub (https://github.com/sucharitk/confidence-distortion-AD/). Aggregated data contains all information available in raw data for the included study participants. The raw data, which also comprises excluded participants but with a more complex data structure, will be made available upon request to the first author (S.K.).

## Code availability
All code for stimulus presentation, data analysis and figure generation is publicly available at Zenodo[51] (https://doi.org/10.5281/zenodo.14244645) and GitHub (https://github.com/sucharitk/confidence-distortion-AD/).

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

## Acknowledgements

S.K. was supported by a grant from Koa Health. S.M.F. is a CIFAR Fellow in the Brain, Mind and Consciousness Program, and funded by a Wellcome/Royal Society Sir Henry Dale Fellowship (206648/Z/17/Z) and UK Research and Innovation (UKRI) under the UK government's Horizon Europe funding guarantee [selected as ERC Consolidator, grant number 101043666]. QJMH acknowledges support by the UCLH NIHR BRC and grant funding from the Wellcome Trust and Carigest S.A. R.J.D. is supported by the Max Planck Society. The Max Planck-UCL Centre for Computational Psychiatry and Ageing Research is a joint initiative supported by University College London and the Max Planck Society. We thank Henrik Singmann for valuable suggestions regarding the use of mixed models. Fruit images used in the stimuli were obtained with permission from Robert Brooks (https://www.gamedeveloperstudio.com/). Emojis used in the stimuli were obtained from (https://github.com/twitter/twemoji) published under a Creative Commons license. For the purpose of Open Access, the authors have applied a CC-BY public copyright license to any author-accepted manuscript version arising from this submission.

## Author contributions

S.K. and S.M.F. designed the study. S.K. developed the models and tasks, collected data, and performed analyses. S.K. and S.M.F. wrote the paper with revisions from Q.J.M.H. and R.J.D. S.M.F., Q.J.M.H., and R.J.D. obtained funding for the research, and S.M.F. provided overall supervision of the study.

## Competing interests

S.K. was funded by Koa Health. The funder did not participate in conceptualization, design, data collection, analysis, or preparation of the manuscript. However, the funder was informed of the concept and design prior to data collection. They were also provided an earlier draft of this manuscript before it was submitted for publication. Q.J.M.H. has obtained fees and options for consultancies for Aya Technologies and Alto Neuroscience. The remaining authors declare no competing interests.
