## [Peer review File · Nature Communications]

REVIEWER COMMENTS

Reviewer #1 (Remarks to the Author):

This study is well-written, timely, and intriguing. It delves into how metacognition operates both at a local level (confidence) across two distinct tasks and globally, with longer-term self-performance estimates (SPEs) influenced by the impact of feedback valence. These dynamics are explored through two online experiments conducted within a sample of English-speaking individuals from the general population. Furthermore, the study examines the moderating influence of symptoms related to anxiety and depression on the effect of feedback valence on both local and global confidence.

The comprehensive inclusion of information within the paper is praiseworthy, and the opportunity to delve into the supplementary results and figures is genuinely gratifying. However, there are instances where crucial information that would enhance comprehension is currently missing and should be incorporated. Additionally, certain conceptual issues are evident, and some conclusions drawn within the abstract and discussion seem to lack support from the provided evidence. I will outline my comments in accordance with the order of sections in the paper.

Abstract:

The claims put forth in the abstract are overly ambitious and not entirely in sync with the actual data. Consequently, the title of the manuscript also appears misaligned with its content and should be reconsidered.

In the abstract, lines 21-22: "Individuals with anxiety and depression exhibit chronic metacognitive biases such as underconfidence."

Given that the study explores the relationship between anxiety and depression scores within a general population sample, a more accurate statement would be to describe it as follows: "Individuals experiencing symptoms of anxiety and depression have shown to exhibit metacognitive biases," omitting the term "chronic."

In the abstract, line 32: "... underconfidence in anxious-depression rooted in .." would be more accurate if changed to "underconfidence associated with anxious and depressed symptoms is associated with...."

The claim made in the abstract: "Together, our results reveal a mechanistic basis for chronic underconfidence in anxious depression rooted in distorted interactions between local and global metacognition, while elucidating a method to restore confidence through targeted feedback," exceeds the support provided by the findings.

The paper employs a computational model to examine potential associations between asymmetrical reactions to positive/negative feedback and low/high confidence and symptoms of anxiety and depression. The results indicate that higher scores on symptoms of anxiety and depression correlate with an increased sensitivity to instances of low local confidence when forming global self-performance estimates. Furthermore, the effects observed in self-endorsement suggest no discernible difference in the alteration of self-endorsements between positive and negative words following positive feedback, whereas negative words received greater endorsement compared to positive words following negative feedback. Consequently, this implies that the provision of positive feedback did not yield a positive impact on self-endorsement. The claim that targeted feedback training could serve as a viable method for addressing symptoms of anxiety and depression thus lacks robust foundation.

In the introduction:

The introduction effectively references related and prior research while leading towards the study's objectives. However, to enhance its clarity and coherence, it is essential to integrate a framework detailing the relationship between low confidence and feedback in relation to the formation of global confidence. As the authors possess a model for testing, it would be prudent to elucidate this framework and then distinctly outline the current gaps in evidence, the formulated hypotheses, and the planned testing methods.

Introduction line 73: "Such biases could predispose depressed individuals to overweigh individual instances of negative feedback and/or low local confidence when forming global SPEs."

Indeed, the mentioned studies suggest heightened sensitivity to negative feedback among individuals with depression. However, the connection between this sensitivity and the weighting of low local confidence when forming global self-performance estimates requires further elucidation.

Furthermore, crafting a comprehensive framework would aid readers in grasping the rationale behind statements such as those in lines 77-78 of the introduction: "It is therefore also important to establish whether asymmetries in global confidence formation transfer across different tasks and/or influence more distal self-beliefs."

Introduction line 37: "A significant component of human mental activity involves how we think about ourselves." The word "However" in this sentence seems redundant and can be omitted.

Introduction line 62: To enhance the persuasiveness of the presented evidence, it would be more compelling to initiate by referencing the plethora of studies demonstrating intervention effects on local confidence. This could then culminate with the smaller subset of studies that delve into the impact of feedback on extended time scales, particularly encompassing global confidence.

Considering the literature, it's plausible that mood manipulation could influence confidence. Hence, there's a need to consider how the identified feedback effects might be influenced by mood manipulation. After all, receiving praise or being informed of errors elicits emotional responses as well.

Additionally, the preregistration notes that "after completing the three phases, participants will be asked a few debriefing questions, including whether they perceived any bias from the auditor in a) giving feedback more frequently when they were correct compared to incorrect, or b) giving feedback more frequently when they were incorrect compared to correct. Furthermore, they will be asked if feedback on correct and incorrect trials influenced their emotional state positively or negatively (or not)." Integrating this data into the manuscript is essential as it not only provides insight into participants' perception of feedback authenticity and trustworthiness but also sheds light on potential mood effects resulting from feedback. Furthermore, examining the associations between anxiety and depression scores and these debriefing responses would be insightful.

In terms of exclusion criteria, it would be valuable to detail the catch questions employed for exclusion and present the numerical breakdown for each exclusion reason.

Results:

Figure 2B provides valuable insights, yet it should incorporate task accuracy per domain and symptom dimensions. This data is crucial since negative feedback can carry greater significance when performance is excellent, and vice versa for subpar performance. Additionally, metacognitive sensitivity at an individual level might impact the feedback effect, given that high metacognitive

sensitivity could reduce the information value of feedback or lead to confusion, unlike individuals with lower sensitivity.

Kindly provide histogram plots for all questionnaires, encompassing the total range of scores and any clinical cutoffs. Furthermore, offer more detailed descriptions of the data concerning the three symptom dimensions and their reliability.

It would be illuminating to observe data from individuals meeting clinical cutoff scores compared to those who do not. If similar outcomes are replicated, this would enhance the face validity of the results.

Considering the literature, it's surprising that the authors did not explore an interactive bias in asymmetric learning from both feedback AND confidence. A model wherein individuals with high scores in anxiety and depression are more sensitive to negative feedback and incorporate low confidence extensively in their self-performance estimates seems like a plausible avenue for investigation. While the authors eventually test this hypothesis, the lack of direct modeling despite relevant literature is unexpected.

Moreover, upon further reflection, it appears unjust to disregard the possibility of confidence bias during feedback trials. It's reasonable to assume that a signal of confidence emerges with any decision, even when not explicitly prompted. While I admit my understanding of the computational model might be limited, the approach of estimating "learning rates" separately for feedback and non-feedback trials (pertaining to local confidence measures) seems insufficient. Furthermore, using these isolated estimates for pos/neg feedback learning and low/high confidence learning appears incongruent with the methodology outlined in Rouault's work from 2019 in Nature Communications.

Could you kindly provide information about the number of trials that constitute the variables of interest? Specifically, how many trials were included for both negative and positive feedback trials? From my understanding, there were around 10 feedback trials per block, is that accurate? Moreover, it's apparent that the number of feedback trials is notably lower than that of the local confidence questions, totalling 30 trials. This discrepancy raises concerns about the available data points for estimating feedback learning in comparison to confidence learning.

Additionally, it's worth considering the potential implications of the unbalanced trial distribution on the model fit. The utilization of 4 parameters (2 for each task and 2 for learning, encompassing feedback and confidence) to achieve the best fit might have been influenced by the substantial variations in data variance, partly stemming from the uneven trial distribution between feedback

and confidence blocks. Despite this, the discussion does not currently address this potential caveat, which could be a substantial limitation warranting consideration.

What happens if model 3 (domain-specific; feedback and confidence same) is used for the analyses concerning the relation with the transdiagnostic symptom domains?

Could you elaborate on the convention used for plotting the data in Figures 4A-4D?

The unanticipated upward shift in block SPEs among individuals with higher anxiety and depression scores is intriguing. Supplementary Figure 4 also underscores this trend across all subjects. This shift could be due to a sense of relief as the task nears completion, possibly indicating a positive mood effect.

Regarding the influence of feedback on self-endorsement, the tests reveal no distinction in the change between positive and negative words following positive feedback. Conversely, negative words receive greater endorsement after negative feedback. While the abstract and discussion are optimistic about positive feedback's impact on self-endorsement, the data here do not support such optimism.

Discussion:

As previously noted, it's important to acknowledge that certain suggestions lack adequate support from the current dataset and should not be presented as conclusive findings.

Moreover, the current discussion would benefit from a comprehensive assessment of the study's potential limitations. Addressing these limitations would provide a more balanced perspective on the study's implications and contribute to a more robust understanding of its scope and implications.

In conclusion, the submitted manuscript presents an interesting exploration of metacognition and feedback's influence on self-performance estimates and its relation with psychiatric symptoms. Nevertheless, there are several aspects that warrant careful consideration. The expansive conclusions drawn from the current data, coupled with the lack of a well-defined conceptual framework and certain methodological intricacies, raise authentic concerns. Addressing these aspects is pivotal before the manuscript can truly make a noteworthy contribution to the field.

Reviewer #2 (Remarks to the Author):

Review of manuscript NCOMMS-23-30273 titled “How underconfidence is maintained in anxiety and depression”.

In this manuscript the authors present two online behavioural experiments where participants were asked to make decisions in gamified versions of perceptual and memory tasks, with different feedback manipulations, measuring local and global confidence estimates, as well as questionnaires and a self-referential encoding task. This is rich and interesting data, and the critical feedback manipulation is well motivated by the literature. The manuscript promises (in the title and abstract) to show how underconfidence is maintained in anxiety and depression, revealing a “mechanistic basis for chronic underconfidence”. This wasn’t so clearly presented in the manuscript. The manuscript lacks clarity and key details that make it difficult to give an endorsement of scientific rigor. For this reason, many of my comments are clarification comments that I hope will aid the authors in generating a more transparent and comprehensible presentation of their findings.

I present below my recommendations for improving the manuscript in what is my standard review structure.

1. Do the conclusions follow from the data?

It is difficult to evaluate the findings because of the lack of clarity and missing details, which I will go into more detail below.

From what I can tell, the argument for a mechanism for maintaining underconfidence comes from the model comparison. The clearest statement for me is at the end of the discussion: “In summary, we identify a mechanistic source of metacognitive distortions present in individuals with high anxious-depression symptoms. Specifically, high AD individuals exhibit greater sensitivity to instances of low local confidence when forming global self-performance estimates, potentially explaining the maintenance of a chronic underconfidence in this population.” I can’t find any standard analyses showing this effect from the data, it is also unclear what the effect is – is it a simple underconfidence bias or also an interaction with feedback condition? It would be helpful to

visually present the data that support this claim. The model comparison adjudicates between 3 possible explanations of why lower global confidence is associated with higher AD, but there is no tangible presentation of how well these possible models explained the data, making it difficult for the reader to accept that the winning model is a good model. It would be good to see a direct comparison of the best fitting model predictions and the data. How does this compare to a simple model that just adds response bias to report lower confidence in AD?

2. Is the analysis conducted appropriately?

As mentioned above, this is difficult to assess. Some clarity could be added by stating more explicitly what was the aim of each test, and perhaps organising the results section into smaller more digestible chunks. Giving plain, acronym free explanations of critical results would also be helpful.

More visuals would also help, plots of raw data: average confidence and accuracy (or difficulty, depending on whether there was a staircase or not) over blocks, according to feedback manipulation, and so forth. Showing how AD score predicts average confidence in a scatter, or a median split. For the major tests, a visual of the data would help orient readers to understand what the test means.

Details of the tests would also help the reader understand what the test means. In the first paragraph of the results (after introducing the experiment), I had assumed the t-statistics corresponded to t-tests, especially because of the wording “Metrics of first-order performance did not differ between positive and negative feedback blocks”, normally for a linear model the precise wording would be “positive vs negative feedback block did not predict first-order performance metrics”. At this point I also wondered how the degrees of freedom in experiment 2 differed for accuracy and difficulty level, but they are the same for experiment 1. I also wondered why the tests didn’t match the title of the paragraph, which calls for a significant interaction effect and therefore an ANOVA would be a good choice.

I think these t-statistics actually come from linear mixed effect models, which are the only statistics described in the methods. For each model, before reporting the outcome, the model itself needs to be clearly described. What was being predicted? How did you make sure data were normally distributed (accuracy, SPEs, endorsement scores, are bounded at 0 and 1)? What were the predictors? Were these normalised (z-scored)? What correction was applied for multiple comparisons? How does one get 20,920 degrees of freedom out of ~250 participants? Some readers are fine with Satterthwaite, but all readers would be happy if with a comparison of the

likelihoods of the model with and without your variable (likelihood ratio test). Maybe this is helpful: https://lmackerman.com/notebooks/prose_statistics.nb.html

The methods only discuss linear mixed effects models, but these are inappropriate for proportions and measures bounded between 0 and 1. For proportions, like accuracy, a generalised linear model with a binomial link function is appropriate.

The statement about pre-registration followed by an OSF link implied an OSF pre-registration. It may be more transparent to state that this is actually a custom file uploaded, not a formal OSF pre-registration. There are some details about deviations from the pre-registration in the manuscript. I was surprised to notice that the pre-registration includes an analysis of Experiment 1 that differs from what is presented in the manuscript. Notably, the number of participants. Was there a mistake? Were the exclusion criteria changed after the pre-registration? Were participants added later? The reason should be noted in the manuscript in case some readers infer that there was p-hacking.

The null model, D0 uses local confidence to update global confidence, line 632. Local confidence is lower in high AD, Line 167. Why does D0 predict no difference in SPE between high/low AD (Figure 3B)?

There should probably be some statistics to support the claims on lines 155-158, or leave it all to the supplementary.

3. Do the introduction and methods give sufficient information for understanding the experiment?

What is “difficulty level” (line 132)? The methods state “we used the mean value of the last 5 reversals of staircase level as the starting level of task difficulty used for the main part of the study” so it should be the same in both feedback conditions. There is also a supplementary analysis on “difficulty reached”. Was the staircase continued throughout, contrary to what is stated in the methods? If the staircase was continued, why would there be any effects on mean accuracy?

In the introduction, I experienced some difficulty following what was the purpose or the main story being told. Perhaps this could be made more specific to the hypotheses tested in this experiment? It would also help with the flow of the results section if the introduction more closely followed the structure of the specific hypotheses tested in this experiment.

4. Does the discussion sufficiently outline the findings?

This was difficult to assess because of the lack of clarity of the results section.

Minor:

“Replicate” may be used a little too liberally. Normally “replicate” means the same method and results were found. At least the same results. At line 136: “Replicating previous findings^{9,26}, mean local confidence (metacognitive bias) significantly predicted global SPEs” reference 9 refers to a study that found local confidence sensitivity predicted global confidence sensitivity (not bias), and reference 26 refers to a study that did not even measure local confidence. Experiment 2 is called a “replication sample”, but there are several differences in the methods of Exp 2 and Exp 1. It may be better to call it something like a ‘partial replication’ – some of the methods were replicated, other aspects were not.

Figure 2A mentions the ‘baseline’ in the legend, but the y-axes are misleading (at first impression it looks like participants had negative ‘actual performance’).

Reviewer #3 (Remarks to the Author):

The authors conducted two experiments (one exploratory and one preregistered) and used computational methods to investigate the sources of metacognitive biases and their relationships with anxiety and depression (AD) symptoms. Specifically, they carefully manipulated feedback valence and found a domain-general effect of feedback valence on confidence. They found that individuals with stronger AD symptoms tended to put heavier weights on low local confidence responses in the process of computing their global self-performance estimates (SPEs).

The study asks an important question concerning the bias in self evaluation in relation to AD symptoms. It has implications across basic sciences and clinical areas. The methods are appropriate and solid. I am most impressed by the rigor in the designs of the behavioral tests and the computational methods. The model-comparison approach is creative. In general, the study is a

great example of how to combine behavioral and computational methods to address a question that interests researchers and clinicians across fields.

I only have one major comment, plus a few minor comments/suggestions.

MAJOR:

The manuscript is drawing conclusions about individuals with “anxiety and depression”. I suppose the authors were referring to the clinical conditions of anxiety and depression. But it seems that the samples were taken from a general population instead of a clinical one. It is of course not necessary for the study to have a sample clinically diagnosed with AD conditions in order to draw such conclusions, but individuals with clinically-diagnosed AD could demonstrate qualitatively different patterns of behavior (and perhaps different computational processes as well).

Perhaps it will be helpful to mention this as a possible limitation in the Discussion section. Another more transparent and informative way to address this point is to report (perhaps in supplementary info) some descriptive statistics of scores related to the AD symptoms in both Exp 1 and Exp 2. An even more helpful piece of information is to provide the clinical cutoff (if any) for each of those AD symptom scales, so that the readers will have a better idea about where the samples stand in the spectrum of AD symptoms.

Minor comments/suggestions:

Behavioral tasks

1. In Figure 2A, consider adding a panel for the local confidence responses as well. Currently, it feels like some of the basic measures (like the local confidence responses and the scores on the AD symptom scales) are missing in the manuscript, but there are a lot of graphs and analyses for the “secondary” outcomes (e.g., fitted parameters and the symptom-bias betas).
2. Have the authors considered analyzing individuals’ metacognitive sensitivity? The data seem to be rich enough for that to be computed at both local (e.g., the SDT approach like meta-d’ approach or type-2 AUROC, etc.) and global level (e.g., correlation of SPE and actual performance)? This might further lengthen the manuscript (especially the supp materials) but it will be interesting to see how that may be related to the metacognitive biases identified and the AD symptoms.
3. Consider reporting effect sizes as well (e.g., Cohen’s d or Hedges’ g), especially for the key effects (e.g., effects of feedback valence). This will be helpful for future meta analysis studies.

Computational models / Model comparison

1. It may be helpful to show the distributions of the symptom-bias betas for all models fitted. It will make it transparent to readers about which parameters may be more stable across models (somewhat related to point 3 below).
2. Why are results from D4 (instead of those from D6, which is the best model) plotted in Figure 4F? The beta-confidence and beta-feedback are indeed interesting, but so is beta-additive, especially if D6 has been shown to be the best model in both Experiments.
3. How would a D7 model that includes all bias components (feedback + confidence + additive) compare with others? In fact, doing so may allow an exhaustive factorial comparison of the bias components as possible factors, which may be a more systematic analysis.
4. Consider plotting Fig 4E on separate scales of DIC for Exp 1 and Exp 2. Currently the differences across the dark bars (Exp 2) are not obvious with the huge DIC values of Exp 1.

Response to Reviewers: **How underconfidence is maintained in the face of reality**

We thank both the Editor and the three Reviewers for their helpful and constructive comments. We have addressed each of their comments carefully, and believe the paper to be much stronger as a result. We provide a point-by-point response to each comment below. The Reviewers' comments are in standard font, our responses are in **green**, and excerpts from the revised manuscript are in **blue**.

Reviewer #1 (Remarks to the Author):

This study is well-written, timely, and intriguing. It delves into how metacognition operates both at a local level (confidence) across two distinct tasks and globally, with longer-term self-performance estimates (SPEs) influenced by the impact of feedback valence. These dynamics are explored through two online experiments conducted within a sample of English-speaking individuals from the general population. Furthermore, the study examines the moderating influence of symptoms related to anxiety and depression on the effect of feedback valence on both local and global confidence.

The comprehensive inclusion of information within the paper is praiseworthy, and the opportunity to delve into the supplementary results and figures is genuinely gratifying. However, there are instances where crucial information that would enhance comprehension is currently missing and should be incorporated. Additionally, certain conceptual issues are evident, and some conclusions drawn within the abstract and discussion seem to lack support from the provided evidence. I will outline my comments in accordance with the order of sections in the paper.

Abstract:

The claims put forth in the abstract are overly ambitious and not entirely in sync with the actual data. Consequently, the title of the manuscript also appears misaligned with its content and should be reconsidered.

Thank you. We have now removed any overly ambitious claims from the abstract (see below) and modified the title to read "How underconfidence is maintained in the face of reality".

In the abstract, lines 21-22: "Individuals with anxiety and depression exhibit chronic metacognitive biases such as underconfidence."

Given that the study explores the relationship between anxiety and depression scores within a general population sample, a more accurate statement would be to describe it as follows: "Individuals experiencing symptoms of anxiety and depression have shown to exhibit metacognitive biases," omitting the term "chronic."
In the abstract, line 32: "... underconfidence in anxious-depression rooted in .." would be more accurate if changed to "underconfidence associated with anxious and

depressed symptoms is associated with...."

Thank you. Our modified Abstract now takes these suggestions into account (see below). We refer to "persistent" underconfidence in relation to symptoms of anxiety and depression in the Abstract, as this is informed by past clinical literature (e.g., Fox et al., *eLife*, 2023; Agnoli et al., *Depression & Anxiety*, 2023).

The claim made in the abstract: "Together, our results reveal a mechanistic basis for chronic underconfidence in anxious depression rooted in distorted interactions between local and global metacognition, while elucidating a method to restore confidence through targeted feedback," exceeds the support provided by the findings.

The paper employs a computational model to examine potential associations between asymmetrical reactions to positive/negative feedback and low/high confidence and symptoms of anxiety and depression. The results indicate that higher scores on symptoms of anxiety and depression correlate with an increased sensitivity to instances of low local confidence when forming global self-performance estimates. Furthermore, the effects observed in self-endorsement suggest no discernible difference in the alteration of self-endorsements between positive and negative words following positive feedback, whereas negative words received greater endorsement compared to positive words following negative feedback. Consequently, this implies that the provision of positive feedback did not yield a positive impact on self-endorsement. The claim that targeted feedback training could serve as a viable method for addressing symptoms of anxiety and depression thus lacks robust foundation.

Thank you – we agree that this claim requires further nuance, and have now removed it from the abstract. The revised abstract now reads as follows:

"Individuals experiencing symptoms of anxiety and depression have been shown to exhibit persistent underconfidence. The origin of such metacognitive biases presents a puzzle, given that individuals should be able to learn appropriate levels of confidence from observing their own performance. In two large general population samples (N=230 and N=278), we measured both "local" confidence in individual task instances and "global" confidence as longer run self-performance estimates while manipulating external feedback. Global confidence was sensitive to both local confidence and feedback valence – more frequent positive (negative) feedback increased (respectively decreased) global confidence, with asymmetries in feedback also leading to shifts in affective self-beliefs. Notably, however, global confidence exhibited reduced sensitivity to instances of higher local confidence in individuals with greater transdiagnostic anxious-depression symptomatology, despite sensitivity to feedback valence remaining intact. Our finding of blunted sensitivity to increases in local confidence offers a mechanistic basis for how persistent underconfidence is maintained in the face of reality."

In the introduction:

The introduction effectively references related and prior research while leading towards the study's objectives. However, to enhance its clarity and coherence, it is essential to integrate a framework detailing the relationship between low confidence and feedback in relation to the formation of global confidence. As the authors possess a model for testing, it would be prudent to elucidate this framework and then distinctly outline the current gaps in evidence, the formulated hypotheses, and the planned testing methods.

Introduction line 73: "Such biases could predispose depressed individuals to overweigh individual instances of negative feedback and/or low local confidence when forming global SPEs."

Indeed, the mentioned studies suggest heightened sensitivity to negative feedback among individuals with depression. However, the connection between this sensitivity and the weighting of low local confidence when forming global self-performance estimates requires further elucidation.

Furthermore, crafting a comprehensive framework would aid readers in grasping the rationale behind statements such as those in lines 77-78 of the introduction: "It is therefore also important to establish whether asymmetries in global confidence formation transfer across different tasks and/or influence more distal self-beliefs."

Thank you for pointing out the need for more clearly communicating our theoretical framework in the Introduction. We have now made several changes throughout the paper, including a new Figure 1 (reproduced below as Figure R1) in which we elucidate the model and its predictions. We have significantly modified paragraph 2 of the Introduction in line with your recommendation below to review the literature on factors influencing local and global confidence starting on page 3, line 52. This new passage culminates in an outline of the standard model in which local confidence and feedback are combined to form global confidence. We supplement this narrative description with Figures 1a–1c which demonstrate the core features of the model.

"One promising route to understanding the source of these metacognitive biases is to unpack how confidence is formed, particularly at a global level. Several factors have been shown to influence confidence formation over and above objective performance. For example, induction of negative vs. positive affect or low vs. high reward expectations can decrease vs. increase local confidence, respectively^{10–12}. Manipulating beliefs about performance on an upcoming task, either through feedback or expected task difficulty, affects local confidence on subsequent instances of a similar task¹³. Finally, local confidence is higher on trials immediately following positive compared to negative feedback¹⁴. With respect to global confidence, false performance feedback in an ambiguous task has been found to influence people's global SPEs¹⁵. Similarly, providing true performance feedback, compared to a no-feedback condition, can increase global SPEs, despite objective task performance remaining unaffected^{7,16}. At the same time, local confidence remains a robust predictor of global SPEs in the absence of feedback^{7,16,17}. Together these findings have informed a computational model in which global confidence is formed by probabilistically combining instances of local

confidence with feedback to update a prior over expected performance⁷ (Figures 1a–1c).”

We have also added text to paragraph 3 to clearly describe our key hypotheses about how different learning distortions are related to anxious-depression symptoms (page 4, line 68). This narrative description is accompanied by Figure 1d which highlights the different mechanistic sources of distortions in global SPEs we seek to test.

“Despite this progress in understanding the computations underpinning confidence formation, whether and how these influences differ in people with anxious-depression symptoms remains unexplored. One attractive hypothesis is that underconfidence may be grounded in a tendency for individuals with anxious-depression symptoms to incorporate less positive and more negative information when making future predictions, particularly about themselves^{18–22}. Recent work has identified a self-related negativity bias in individuals with anxious-depression symptoms in relation to learning about expectations of future adverse life events^{21,23} and when updating performance expectations from feedback²². Such biases could predispose individuals with anxious-depression symptoms to distort individual instances of feedback and/or local confidence when forming global SPEs. Specifically, when forming a global sense of confidence, individuals with anxious-depression symptoms may be more sensitive to negative (compared to positive) external feedback, low (compared to high) local confidence, or a combination of the two (Figure 1d).”

Finally, in a final paragraph of the Introduction, we now clearly state the key hypotheses tested in this study (page 4, line 87). This is supplemented by new Figures 1e–1f which depict the different qualitative predictions made by different sources of distortion within the model.

“In the present study, we sought to determine how computations underpinning global confidence formation are altered in individuals with anxious-depression symptoms. We systematically manipulated performance feedback to impact global confidence. Then, using computational modelling, we distinguished between several mechanisms of how anxious-depression symptoms distort the incorporation of local confidence and performance feedback into global SPEs. Specifically, such distortions could manifest as altered sensitivity to local confidence, altered sensitivity to feedback valence, a general negative response bias, or a combination of these factors. Each of these three mechanisms makes qualitatively distinct predictions for the patterns of behaviour we expect to observe in the experiments (see Figures 1e–1g for schematic illustration).”

Figure R1. a) Outline of a canonical task in which individual trials are followed by local confidence reports and occasional (veridical) performance feedback. After completing the block, participants provide a global self-performance estimate (SPE) for that block. **(b-c)** Schematic illustration of different influences on global confidence within computational accounts of confidence formation. **b)** Global SPEs are expected to increase proportionally with local confidence. **c)** More frequent feedback on correct (respectively incorrect) trials is expected to increase (decrease) global SPEs. **d)** Three candidate computational mechanisms for how the formation of global SPEs differs as a function of anxious-depression symptoms. **(e-g)** Visualisation of how the base patterns of global confidence formation depicted in panels **(b)** and **(c)** are expected to differ as a function of the different mechanisms described in **(d)**. **e)** Prediction for model D1, distortion in confidence learning – people with higher anxious-depression symptom scores (blue lines) show blunted influence of higher local confidence on global SPEs compared to lower anxious-depression symptom scores (gold lines). **f)** Prediction for model D2, distortion in feedback learning – compared to the absence of feedback, higher anxious-depression symptoms are associated with a greater sensitivity to incorrect vs. correct feedback, and vice-versa for lower anxious-depression symptoms. **g)** Prediction for D3, negative response bias – under a simple response bias model, we expect an overall reduction in global SPEs when comparing high vs. low anxious-depression scores in the absence of the nonlinearities depicted in **(e)** and **(f)**.

Introduction line 37: "A significant component of human mental activity involves how we think about ourselves." The word "However" in this sentence seems redundant and can be omitted.

Thank you. We feel on balance that 'however' is useful here, because we are contrasting "how we think about ourselves" in this line with "how humans perceive and represent their environment" from the previous line.

Introduction line 62: To enhance the persuasiveness of the presented evidence, it would be more compelling to initiate by referencing the plethora of studies demonstrating intervention effects on local confidence. This could then culminate with the smaller subset of studies that delve into the impact of feedback on extended time scales, particularly encompassing global confidence. Considering the literature, it's plausible that mood manipulation could influence confidence. Hence, there's a need to consider how the identified feedback effects might be influenced by mood manipulation. After all, receiving praise or being informed of errors elicits emotional responses as well.

Thank you for these suggestions. We have addressed this comment above where we outline the extensive changes now made to the Introduction to situate our work within the context of studies demonstrating effects of interventions on local confidence.

We have now also added a line to the Discussion to consider how mood changes may mediate the impact of feedback on global confidence (page 17, line 306):

"Our participants reported that feedback induced valence-specific mood changes (Supplementary Material); considering the relationship between mood and confidence, it is possible that feedback's impact on global confidence may have been mediated by mood shifts¹¹."

Additionally, the preregistration notes that "after completing the three phases, participants will be asked a few debriefing questions, including whether they perceived any bias from the auditor in a) giving feedback more frequently when they were correct compared to incorrect, or b) giving feedback more frequently when they were incorrect compared to correct. Furthermore, they will be asked if feedback on correct and incorrect trials influenced their emotional state positively or negatively (or not)." Integrating this data into the manuscript is essential as it not only provides insight into participants' perception of feedback authenticity and trustworthiness but also sheds light on potential mood effects resulting from feedback. Furthermore, examining the associations between anxiety and depression scores and these debriefing responses would be insightful.

Thank you for noting this omission from our side. We have now included this information in Supplementary Results under the subheading "Participants' awareness of study manipulations" along with plots in Supplementary Figure 23 (reproduced as Figure R2 below). We find an interesting effect where in both experiments, participants who report feeling worse after negative feedback (compared to those who do not) have higher anxiety scores. There was however no

Figure R2. a, d) The number of participants who reported noticing that one of the blocks had biased positive feedback (more feedback on correct trials) and negative feedback (more feedback on incorrect trials) in Exp 1 and Exp 2. **b, e)** Number of participants who reported different changes in how they felt after feedback on correct and incorrect trials in Exp 1 and Exp 2. **c, g)** Anxiety scores in Exp 1 and AD axis scores of participants who reported feeling worse after incorrect trials compared to those who reported not feeling worse.

difference in symptoms for people who reported awareness of the positive / negative feedback manipulation compared to those who did not.

In terms of exclusion criteria, it would be valuable to detail the catch questions employed for exclusion and present the numerical breakdown for each exclusion reason.

We have now described the catch questions in the Participants subheading in Supplementary Methods (page 2). We also now provide full details of the exclusion procedure, specifying the number of participants that were excluded with each criterion as follows:

“Participants were excluded from all analyses if they missed one of the three “catch” questions. Two catch questions were administered during the self-referential encoding task (SRET). Here participants were shown the words ‘human and ‘keyboard’ randomly placed between the other 20 positive and negative adjectives; participants were excluded if they did not self-endorse these words maximally (‘Yes’ in Exp 1 and $>.875$ on the slider in Exp 2) and minimally (‘No’ in Exp 1 and $<.125$ on the slider in Exp 2) respectively. Additionally, embedded within mental health questionnaires, participants were asked “I take astronaut missions to space” and were excluded if they did not select the option “Never.” In all, 14 participants were excluded in Exp 1 and 71 participants in Exp 2 for missing at least one catch question.

For analyses involving local confidence and global self-performance estimates, we also excluded participants, 1) whose performance was outside the interval (.60 .85] on any one of the 6 task blocks (Exp 1: 32 participants, Exp 2: 99 participants), 2) who did not exhibit sufficient variability in trial-by-trial confidence ratings defined as having $<.05$ SD across trials for each task (on a continuous confidence scale of 0—1; Exp 1: 9 participants, Exp 2: 12 participants), and 3) who did not have stable behavioural staircases in the perception/memory tasks as assessed visually (Exp 1: 19 participants, Exp 2: 0 participants). Finally, one participant from Exp 2 was excluded because their questionnaire data did not get saved on our database (possibly due to an internet issue at their end).”

Results:

Figure 2B provides valuable insights, yet it should incorporate task accuracy per domain and symptom dimensions. This data is crucial since negative feedback can carry greater significance when performance is excellent, and vice versa for subpar performance. Additionally, metacognitive sensitivity at an individual level might impact the feedback effect, given that high metacognitive sensitivity could reduce the information value of feedback or lead to confusion, unlike individuals with lower sensitivity.

Thank you for these suggestions. We have now made the following changes to address these points:

- a) We now include plots of accuracy per task domain, and for consistency also split global self-performance estimates by task domain. Following the suggestion of another reviewer, we have now also included a plot of local confidence (per domain).
- b) We have included a plot of regression estimates relating task accuracy to the three symptom dimensions.

These changes are shown in Figure R3 below, which is a reproduction of Figure 3 in the revised manuscript.

In addition, we have conducted the analyses you suggested with regards to metacognitive sensitivity. We calculated M_{ratio} (meta- d'/d') for each participant and task during baseline blocks and tested if these baseline estimates of metacognitive ability moderated the influence of feedback on global SPEs (and if such a moderation effect is task-specific) using the following regression models:

$SPE \sim Feedback_type * Task * M_{ratio} + (1|Participant)$

$SPE \sim Feedback_type * M_{ratio} + (1|Participant)$

There were no significant 3- or 2-way interactions with M_{ratio} in either Exp 1 or Exp 2 (all $p > .38$), suggesting that baseline metacognitive ability did not moderate the impact of feedback on SPEs. We now report these findings in Supplementary Material.

Figure R3. a) Global confidence measured as self-performance estimates (global SPEs), **b)** performance (mean accuracy), and **c)** mean local confidence on intervention blocks with more frequent positive (green) or negative feedback (red) plotted separately for the two tasks in Exp 1. For global SPEs and local confidence, baseline values were subtracted out for each individual. Each dot is an individual participant. Error bars show 95% bootstrapped confidence intervals. **d)** Regression coefficients depicting the relationship in Exp 2 between transdiagnostic symptom axes and mean local confidence, global SPEs measured retrospectively, global SPEs measured prospectively, and accuracy. **** $p < .0001$, * $p < .05$.

Kindly provide histogram plots for all questionnaires, encompassing the total range of scores and any clinical cutoffs. Furthermore, offer more detailed descriptions of the data concerning the three symptom dimensions and their reliability.

Thank you for suggesting we include this information. We have now added Supplementary Figure 22 (reproduced below as Figure R4) depicting histograms for PHQ-9 and GAD-7 with clinical cutoffs from Exp 1.

In Exp 2, we used a questionnaire battery that included subsets of questions from different standard clinical questionnaires optimised for extracting transdiagnostic symptom dimensions (Hopkins et al, *PsyArxiv*, 2022). Because full depression and anxiety questionnaires were not collected, we could not evaluate the Exp 2 sample in terms of standardised scores and clinical cutoffs.

We have included a subsection in the Supplementary Material titled “Mental health scales and clinical cutoffs” where we outline these points.

It would be illuminating to observe data from individuals meeting clinical cutoff scores compared to those who do not. If similar outcomes are replicated, this would enhance the face validity of the results.

Thank you for this suggestion. In Supplementary Figure 10 (reproduced here as Figure R5), we now depict our key effect – moderation of the linear relationship between local and global confidence at mid-to-high local confidence levels by GAD-7 and PHQ-9 scores – by separating participants into two groups based on standardised clinical cutoffs for minimal risk and moderate-to-severe risk. We also statistically tested if participants with moderate-to-severe risk exhibited the blunted sensitivity of global confidence to local confidence compared to those with minimal risk separately for general anxiety (GAD-7) and depression (PHQ-9) symptom scores. We observed a statistically significant effect of such blunted sensitivity in participants with moderate-to-severe general anxiety scores ($\chi^2 = 5.77, p = .016$), and a trend for this effect in participants with moderate-to-severe depression scores ($\chi^2 = 3.35, p = .067$).

Considering the literature, it's surprising that the authors did not explore an interactive bias in asymmetric learning from both feedback AND confidence. A model wherein individuals with high scores in anxiety and depression are more sensitive to negative feedback and incorporate low confidence extensively in their self-performance estimates seems like a plausible avenue for investigation. While the authors eventually test this hypothesis, the lack of direct modeling despite relevant literature is unexpected.

Thank you for raising this important point. As you highlight, we did fit such a feedback+confidence distortion model to our data which was presented in the Results section of the original submission. However, in light of past work, we agree that it is also useful to mention the possibility of asymmetric learning from both feedback and confidence upfront in the Introduction (and not just when describing the analyses). We have now modified the Introduction text to reflect this, as follows (page 4, line 77):

“Specifically, when forming a global sense of confidence, individuals with anxious-depression symptoms may be more sensitive to negative (compared to positive) external feedback, low (compared to high) local confidence, or a combination of the two (Figure 1d).”

We also highlight in the Introduction how our novel plots and model-agnostic regression analyses allow us to test for the presence of a simultaneous feedback + confidence distortion (page 4, line 92):

“Specifically, such distortions could manifest as altered sensitivity to local confidence, altered sensitivity to feedback valence, a general negative

response bias, or a combination of these factors. Each of these three mechanisms makes qualitatively distinct predictions for the patterns of behaviour we expect to observe in our experiments (see Figures 1e–1g for schematic illustration).”

Note that, as before, these new analyses confirm the presence of distortions in the integration of confidence but not feedback when forming global SPEs.

Moreover, upon further reflection, it appears unjust to disregard the possibility of confidence bias during feedback trials. It's reasonable to assume that a signal of confidence emerges with any decision, even when not explicitly prompted. While I admit my understanding of the computational model might be limited, the approach of estimating "learning rates" separately for feedback and non-feedback trials (pertaining to local confidence measures) seems insufficient. Furthermore, using these isolated estimates for pos/neg feedback learning and low/high confidence learning appears incongruent with the methodology outlined in Rouault's work from 2019 in Nature Communications.

Thank you for raising these points. We agree that confidence is likely to be formed regardless of whether it is explicitly prompted. An interesting question, though, is whether such unprompted confidence interacts with (or even overrides) the external feedback. In Rouault et al's (2019) original model, on trials without feedback, the model assumes that the best estimate the agent has of their performance is their confidence, and thus the reported confidence (the probability that one is correct) is used to probabilistically update one's SPE. In contrast, on trials with (veridical) feedback the model instead posits that only feedback (and not reported confidence) is used to update SPEs – as subjects are assumed to be certain about being correct or incorrect based on this information. This is the model we extended for use in our study.

However, we agree that participants might not conform to these assumptions, and that a feeling of confidence they have on trials where they subsequently also receive veridical performance feedback may contribute to the formation of SPEs. To explore this possibility, we modified our original set of models so that on feedback trials both the reported confidence and feedback were used to update SPEs, in line with the reviewer's suggestion. Our original model updated the a and b parameters of the beta distribution over SPEs on feedback trials as follows:

$$a_{t+1} = \begin{cases} a_t + (1 + \Delta LR_f) & \text{if correct} \\ a_t & \text{if incorrect} \end{cases}$$
$$b_{t+1} = \begin{cases} b_t & \text{if correct} \\ b_t + (1 - \Delta LR_f) & \text{if incorrect} \end{cases}$$

In a new model variant, we now modify these equations to also include learning from reported confidence, as follows:

$$a_{t+1} = \begin{cases} a_t + (1 + \Delta LR_f) + conf_t * (1 + \Delta LR_c) & \text{if correct} \\ a_t & \text{if incorrect} \end{cases}$$

$$b_{t+1} = \begin{cases} b_t & \text{if correct} \\ b_t + (1 - \Delta LR_f) + (1 - conf_t) * (1 - \Delta LR_c) & \text{if incorrect} \end{cases}$$

When doing this we find that our original models (where confidence is not incorporated into SPEs on feedback trials) provide better fits to both our datasets in terms of DIC scores – hinting that the original assumptions of Rouault et al., where feedback overrides confidence, provide a more parsimonious fit to the data. The following table shows DIC values for the two experiments for both our “original” models (without confidence influencing learning on feedback trials), and “modified” models, where confidence also exerts an influence on feedback trials.

	Exp 1		Exp 2	
	Original model	Modified model	Original model	Modified model
No distortion	6692	6840	4665	4748
Confidence distortion only	6232	6371	4646	4728
Feedback distortion only	6610	6731	4667	4750
Confidence + feedback distortion	6233	6374	4641	4724
Confidence distortion + additive bias	6210	6353	4625	4706

Thus, based on these model comparisons, adding an additional influence of confidence on feedback trials does not help in explaining subjects’ learning. Due to the exploratory nature of this analysis and for the sake of brevity, we have not included it in the manuscript, but would be happy to revisit this decision should the reviewer or editor wish us to do so.

We interpret the second part of your comment about our model being incongruent with Rouault et al (2019) as referring to the estimation of asymmetric learning rates *separately* for feedback and confidence, despite the two being considered along a continuum in the original model. Our reason for estimating the learning rates separately for external performance feedback and confidence asymmetries had to do with the distinct qualitative nature of these two processes that we were sought to distinguish in the current study. In the original Rouault et al. model a local confidence rating of 1 (i.e., maximum confidence) on a trial without feedback is treated equivalently to the receipt of external “correct” feedback when forming SPEs. Here our goal was to allow for the possibility that feedback and local confidence differentially impact SPEs. Our distinct learning rates allowed us to do this, by testing if learning of longer-run self-related negative biases was more (or less) pronounced

when driven by local confidence compared to feedback. We have added the following line in the Introduction to make this point explicit (page 4, line 77):

“Specifically, when forming a global sense of confidence, individuals with anxious-depression symptoms may be more sensitive to negative (compared to positive) external feedback, low (compared to high) local confidence, or a combination of the two (Figure 1d).”

Could you kindly provide information about the number of trials that constitute the variables of interest? Specifically, how many trials were included for both negative and positive feedback trials? From my understanding, there were around 10 feedback trials per block, is that accurate? Moreover, it's apparent that the number of feedback trials is notably lower than that of the local confidence questions, totalling 30 trials. This discrepancy raises concerns about the available data points for estimating feedback learning in comparison to confidence learning.

Additionally, it's worth considering the potential implications of the unbalanced trial distribution on the model fit. The utilization of 4 parameters (2 for each task and 2 for learning, encompassing feedback and confidence) to achieve the best fit might have been influenced by the substantial variations in data variance, partly stemming from the uneven trial distribution between feedback and confidence blocks. Despite this, the discussion does not currently address this potential caveat, which could be a substantial limitation warranting consideration.

We thank you for noting this point. In total, participants performed 240 trials in Exp 1 and 200 trials in Exp 2. Of these, 80 were confidence-only baseline trials, and 80 (40 in Exp 2) were confidence-only test trials (designed to probe how our intervention transfers to subsequent blocks without feedback). In our critical intervention blocks, each participant received 20 feedback trials (~10 trials of positive feedback and ~10 trials of negative feedback), with the remaining 60 trials being confidence-only trials. We note that in the intervention block we could only provide a limited number of feedback trials to prevent the feedback manipulation from becoming apparent to the participants. In addition, because feedback was under experimental control, estimating the impact of our feedback manipulation on SPEs was expected to be more straightforward than estimating the impact of confidence fluctuations on SPEs.

We appreciate your concern that a difference in the number of confidence and feedback trials may have affected estimation of learning rates for feedback, and therefore our ability to assess the relationship between feedback learning and individual differences. However, several features of our data convince us that our inferences on feedback learning are robust. First, both positive and negative feedback interventions had a strong and systematic impact on global SPEs with a large effect size, despite being driven by a small number of critical trials (in both experiments, we observed Cohen's $d = 1.77$). Second, both regression analyses and computational model fits were performed hierarchically, such that data were pooled over subjects when estimating learning parameters, allowing our analyses to utilise 4600 and 5560 (positive and negative) feedback trials in Exp 1 and Exp 2 respectively. Third, our novel Bayes Factor analysis in the manuscript shows at least moderate (and in some cases strong) evidence for a lack of effect of feedback in both datasets. Fourth, and most importantly, the parameter recovery plots depicted in Supplementary Figure 10 were obtained from a scenario in which we

simultaneously introduced variation in both ground-truth feedback and confidence distortions. This figure shows that our model can successfully recover both learning asymmetry parameters independently, despite the difference in trial numbers contributing to these fits. We note however that, likely due to this trial number difference, the recovery of the confidence asymmetry parameter was more precise than the feedback asymmetry parameter across a range of intercept values of confidence and feedback asymmetry.

We now mention all these points in a new paragraph of the Discussion which explores limitations of our study, as follows (page 19, line 365):

“One limitation of the present study is that in our design the number of trials for determining asymmetric feedback learning were substantially fewer than confidence-only trials used to estimate local-confidence-based distortions. More confidence-only than feedback trials were required by design because while feedback was experimentally controlled, sufficient variation in the trial-by-trial confidence was not. Nonetheless, several reasons lead us to believe that the lack of a feedback effect was real. First, both positive and negative feedback interventions had a strong and systematic impact on global SPEs, despite being driven by a small number of critical trials. Second, both regression analyses and computational model fits were performed hierarchically, such that data were pooled over subjects when estimating learning parameters, leading to 4600 and 5560 (positive and negative) feedback trials in Exp 1 and Exp 2 respectively. Third, model-neutral Bayes Factor analysis showed at least moderate (in some cases strong) evidence for the lack of an effect of feedback in both datasets. Fourth, and most importantly, simultaneously introduced variation in both ground-truth feedback and confidence distortion regression parameters were recoverable from our model in simulation. Another limitation is that we used a general population sample from an online participant recruitment platform. While the sample contained a substantial number of individuals who surpassed clinical cutoffs on standard anxious-depression scales, further work is required to investigate global confidence distortions in individuals clinically diagnosed with anxiety and depression or in general samples recruited through platforms other than the one used in this study.”

What happens if model 3 (domain-specific; feedback and confidence same) is used for the analyses concerning the relation with the transdiagnostic symptom domains?

Thank you for this suggestion. We have now fitted such a model (domain-specific; feedback and confidence same) when predicting anxious-depression symptom scores in both our datasets. In both experiments, we find that such a model provides worse fits based on DIC scores than the set of models reported in the manuscript (domain-specific + feedback and confidence different learning rates). The DIC scores for such a model are 6314 for Exp 1 and 4712 for Exp 2 (compare that with values shown in ‘Original model’ columns in the Table above). For the sake of brevity, we do not plan to include this exploratory analysis in the manuscript but would be happy to revisit this decision if the reviewer or editor prefers.

Could you elaborate on the convention used for plotting the data in Figures 4A-4D?

We have now substantially modified the plots in Figure 4 and have tried to ensure that the figure caption effectively communicates what is depicted in the plots.

The unanticipated upward shift in block SPEs among individuals with higher anxiety and depression scores is intriguing. Supplementary Figure 4 also underscores this trend across all subjects. This shift could be due to a sense of relief as the task nears completion, possibly indicating a positive mood effect.

We apologise for any confusion introduced by this plot due to a lack of clarity in captions. The (previous) Supplementary Figure 4 (now Supplementary Figure 16) depicted the trial-by-trial transfer effect of feedback upon test blocks. This is different from the plots showing elevated baseline-subtracted feedback-block SPEs for individuals with high anxiety and depression scores in our previous Figure 4. As mentioned above, we have now substantially modified Figure 4, and hope that the caption and manuscript text more clearly describe the plots.

Regarding the influence of feedback on self-endorsement, the tests reveal no distinction in the change between positive and negative words following positive feedback. Conversely, negative words receive greater endorsement after negative feedback. While the abstract and discussion are optimistic about positive feedback's impact on self-endorsement, the data here do not support such optimism.

Thank you. As we mentioned above, we have now removed this optimism from the abstract and discussion.

Discussion:

As previously noted, it's important to acknowledge that certain suggestions lack adequate support from the current dataset and should not be presented as conclusive findings.

Moreover, the current discussion would benefit from a comprehensive assessment of the study's potential limitations. Addressing these limitations would provide a more balanced perspective on the study's implications and contribute to a more robust understanding of its scope and implications.

Thank you. We agree it is important to outline the limitations of the study. As mentioned above, we have now added a new paragraph to the Discussion on this point (quoted in response to your earlier point).

In conclusion, the submitted manuscript presents an interesting exploration of metacognition and feedback's influence on self-performance estimates and its relation with psychiatric symptoms. Nevertheless, there are several aspects that warrant careful consideration. The expansive conclusions drawn from the current data, coupled with the lack of a well-defined conceptual framework and certain methodological intricacies, raise authentic concerns. Addressing these aspects is pivotal before the manuscript can truly make a noteworthy contribution to the field.

We are grateful to the reviewer for this very thoughtful and constructive review.

Reviewer #2 (Remarks to the Author):

Review of manuscript NCOMMS-23-30273 titled “How underconfidence is maintained in anxiety and depression”.

In this manuscript the authors present two online behavioural experiments where participants were asked to make decisions in gamified versions of perceptual and memory tasks, with different feedback manipulations, measuring local and global confidence estimates, as well as questionnaires and a self-referential encoding task. This is rich and interesting data, and the critical feedback manipulation is well motivated by the literature. The manuscript promises (in the title and abstract) to show how underconfidence is maintained in anxiety and depression, revealing a “mechanistic basis for chronic underconfidence”. This wasn’t so clearly presented in the manuscript. The manuscript lacks clarity and key details that make it difficult to give an endorsement of scientific rigor. For this reason, many of my comments are clarification comments that I hope will aid the authors in generating a more transparent and comprehensible presentation of their findings.

I present below my recommendations for improving the manuscript in what is my standard review structure.

1. Do the conclusions follow from the data?

It is difficult to evaluate the findings because of the lack of clarity and missing details, which I will go into more detail below.

From what I can tell, the argument for a mechanism for maintaining underconfidence comes from the model comparison. The clearest statement for me is at the end of the discussion: “In summary, we identify a mechanistic source of metacognitive distortions present in individuals with high anxious-depression symptoms. Specifically, high AD individuals exhibit greater sensitivity to instances of low local confidence when forming global self-performance estimates, potentially explaining the maintenance of a chronic underconfidence in this population.” I can’t find any standard analyses showing this effect from the data, it is also unclear what the effect is – is it a simple underconfidence bias or also an interaction with feedback condition? It would be helpful to visually present the data that support this claim. The model comparison adjudicates between 3 possible explanations of why lower global confidence is associated with higher AD, but there is no tangible presentation of how well these possible models explained the data, making it difficult for the reader to accept that the winning model is a good model. It would be good to see a direct comparison of the best fitting model predictions and the data. How does this compare to a simple model that just adds response bias to report lower confidence in AD?

We thank the reviewer for their comprehensive assessment of our paper, and for raising a number of important points. We agree that the data should be visualised

more clearly to convey the key effects. We now present the data in a way that clearly depicts the different relationships between local confidence, local feedback, global confidence and symptom status in a model-neutral manner. This can be appreciated schematically in Figure R1 (a reproduction of Figure panels 1d—1g in the main manuscript), and in the formatting of the output of the computational model in Supplementary Figure 8. These new visualisations also demonstrate why distortions in local confidence and feedback learning are qualitatively distinct from simple underconfidence biases (which are instead captured by our response-bias model). As we describe in more detail below, unlike additive biases, learning distortions manifest as specific nonlinearities in the relationships between global SPEs with local confidence and/or feedback.

Specifically, a distortion in local confidence learning can be visualised in a plot of the relationship between different levels of local confidence (from high to low) and global SPEs, separately for low and high anxious-depression symptoms (Figure R1e). This distortion manifests as heightened sensitivity (steeper slope) relating global SPEs to higher local confidence in people with lower anxious-depression symptoms. The same slope is reduced for people with higher anxious-depression symptoms, reflecting a blunted sensitivity to higher (compared to lower) local confidence. On the other hand, a feedback learning distortion can be visualised in a plot of the relationship between feedback (positive, negative, none) and global SPEs, separately for low and high anxious-depression symptoms (Figure R1f). This distortion in contrast manifests as a differential impact of positive (vs. negative) feedback on global SPEs in people with low (vs. high) anxious-depression symptoms. Finally, a simple response bias can be visualised by combining the previous two plots, resulting in an overall reduction in SPEs, but in the absence of the above two nonlinearities (Figure R1g).

Figure R1. a) Outline of a canonical task in which individual trials are followed by local confidence reports and occasional (veridical) performance feedback. After completing the block, participants provide a global self-performance estimate (SPE) for that block. **(b-c)** Schematic illustration of different influences on global confidence within computational accounts of confidence formation. **(b)** Global SPEs are expected to increase proportionally with local confidence. **(c)** More frequent feedback on correct (respectively incorrect) trials is expected to increase (decrease) global SPEs. **(d)** Three candidate computational mechanisms for how the formation of global SPEs differs as a function of anxious-depression symptoms. **(e-g)** Visualisation of how the base patterns of global confidence formation depicted in panels **(b)** and **(c)** are expected to differ as a function of the different mechanisms described in **(d)**. **(e)** Prediction for model D1, distortion in confidence learning – people with higher anxious-depression symptom scores (blue lines) show blunted influence of higher local confidence on global SPEs compared to lower anxious-depression symptom scores (gold lines). **(f)** Prediction for model D2, distortion in feedback learning – compared to the absence of feedback, higher anxious-depression symptoms are associated with a greater sensitivity to incorrect vs. correct feedback, and vice-versa for lower anxious-depression symptoms. **(g)** Prediction for D3, negative response bias – under a simple response bias model, we expect an overall reduction in global SPEs when comparing high vs. low anxious-depression scores in the absence of the nonlinearities depicted in **(e)** and **(f)**.

These ways of depicting data elicit unique qualitative predictions for the 3 types of distortions, which can in turn be evaluated using model-neutral regression analyses to infer different potential mechanisms. Critically, our data in both experiments qualitatively as well as quantitatively supports the presence of a confidence (but not feedback) learning distortion in anxious-depression, which is now depicted in the new plots in Figure R6 (a reproduction of Figure panels 4a–4d from the main paper). The presence of a confidence learning distortion can be statistically evaluated as a difference in the slope of the regression line between local confidence and global SPEs, evaluated at low and high anxious-depression scores, particularly middle-to-high values of local confidence. Here the hypothesis was that the slope would be less positive for high anxious-depression scores at higher values of local confidence (i.e., z-scored confidence > 0). Critically, we observe this difference (i.e., a 2-way interaction) in both the datasets in the expected direction (Figure R6a and R6b), revealing a qualitative match between model predictions and data. Conversely, we do not find a significant effect of feedback distortion in either dataset (Figure R6c and R6d). This can be statistically tested by estimating the 2-way interactions between *Feedback type* (feedback vs. no feedback) and anxious-depression scores when predicting SPEs, separately for positive and negative vs. no feedback. These new analyses confirm our original interpretations obtained with our model-based analyses, while also providing a clearer visualisation of the data to depict the qualitative patterns we expect under different models.

Figure R6. Data from Exp 1 (left column panels A, C, E) and Exp 2 (right column panels B, D, F). **A & B)** Global SPEs plotted against local confidence (z-scored per individual and split into 4 tiles) separately for individuals with low (gold) and high (blue) anxious-depression symptom scores (obtained using the GAD-7 questionnaire in Exp 1 and transdiagnostically in Exp 2). **C & D)** Global SPEs plotted for blocks with more frequent negative feedback, no feedback, more frequent positive separately for individuals with low and high anxious-depression scores.

2. Is the analysis conducted appropriately?

As mentioned above, this is difficult to assess. Some clarity could be added by stating more explicitly what was the aim of each test, and perhaps organising the results section into smaller more digestible chunks. Giving plain, acronym free explanations of critical results would also be helpful.

Thank you for this push towards greater clarity. We have now divided results relating to mental health into separate subsections that report a) relationships with baseline confidence and performance and b) relationships with learning. We now spell out the respective distortion each time we refer to models D1–D6 for ease of reading. Another potential issue was that we were using AD to refer both to anxious-depression generally and the anxious-depression transdiagnostic axis measured in Exp 2. Instead, we now refer to the former as “anxious-depression” and the latter as Anxious-Depression axis. With respect to each test, we have now ensured that it is preceded by a mention of the effect being tested.

More visuals would also help, plots of raw data: average confidence and accuracy (or difficulty, depending on whether there was a staircase or not) over blocks, according to feedback manipulation, and so forth. Showing how AD score predicts average confidence in a scatter, or a median split. For the major tests, a visual of the data would help orient readers to understand what the test means.

We thank the reviewer for these suggestions. We have now provided additional plots of raw data as requested. Supplementary Figures 3, 4 and 6 depict global self-performance estimates, performance and mean local confidence respectively for each of the eight groups in the two experiments across the six task blocks. In these plots, the feedback manipulation has been colour-coded for ease of reference. SPE data show visible increases / decreases with our positive / negative feedback manipulations, respectively. As expected, these differences are not observed for performance.

In Supplementary Figure 7, we now also show scatter plots of anxious-depression scores (GAD & PHQ in Exp 1 and the AD transdiagnostic axis in Exp 2) against mean local confidence and global SPEs (and prospective SPEs for Exp 2). Because of the potential dependence of confidence on a variety of factors (including age, gender and slight variations in performance), we follow previous work (Rouault et al, 2019; Hoven et al, 2023) in reporting regressions rather than correlations, and plot values of z-scored global SPEs and local confidence after partialing out these influences.

Details of the tests would also help the reader understand what the test means. In the first paragraph of the results (after introducing the experiment), I had assumed the t-statistics corresponded to t-tests, especially because of the wording “Metrics of first-order performance did not differ between positive and negative feedback blocks”, normally for a linear model the precise wording would be “positive vs negative feedback block did not predict first-order performance metrics”. At this point I also wondered how the degrees of freedom in experiment 2 differed for accuracy and difficulty level, but they are the same for experiment 1. I also wondered why the

tests didn't match the title of the paragraph, which calls for a significant interaction effect and therefore an ANOVA would be a good choice.

I think these t-statistics actually come from linear mixed effect models, which are the only statistics described in the methods. For each model, before reporting the outcome, the model itself needs to be clearly described. What was being predicted? How did you make sure data were normally distributed (accuracy, SPEs, endorsement scores, are bounded at 0 and 1)? What were the predictors? Were these normalised (z-scored)? What correction was applied for multiple comparisons? How does one get 20,920 degrees of freedom out of ~250 participants? Some readers are fine with Satterthwaite, but all readers would be happy if with a comparison of the likelihoods of the model with and without your variable (likelihood ratio test). Maybe this is helpful: https://lmackerman.com/notebooks/prose_statistics.nb.html

The methods only discuss linear mixed effects models, but these are inappropriate for proportions and measures bounded between 0 and 1. For proportions, like accuracy, a generalised linear model with a binomial link function is appropriate.

Thank you for pointing out the lack of clarity about the tests and the very helpful suggestions for reporting the statistics. You are correct that the t-statistics we presented were from linear(-ised) mixed models. Our reason for using regressions instead of t-tests and ANOVAs was that confidence is known to depend on several continuous variables that were not of direct interest in the present study (specifically, within-subject factors of accuracy, difficulty/staircase level, reaction times, and between-subject factors of age, and gender). We thus sought to regress out these influences from our statistical models when assessing the target effects of interest (such as the effect of feedback on global and local confidence). We regret that these regression models and the reasoning for using them was not specified in our original submission, creating understandable confusion.

We have now carefully outlined each statistical test we use when within our Results in the Methods section (page 28), as follows:

“For testing differences in SPE on feedback blocks with *Feedback type* (positive, negative), we included 3- and 2-way interactions with *Task* (perception, memory) and *Feedback order* (positive first, negative first) along with main effects of *Accuracy* and *Staircase level*, as follows:

$$SPE_{bc} \sim Feedback_type * Task * Feedback_order + Accuracy_{bc} + Staircase_level_{bc} + (1|Participant) + (1|Group)$$

For control analyses, to test if accuracy and staircase were modulated by *Feedback type* we used the following LMMs (a similar model was used for *Staircase level*):

$$Accuracy_{bc} \sim Feedback_type * Task + (1|Participant) + (1|Group)$$

For testing if confidence and accuracy predicted global SPEs, we used all 6 task blocks with the following LMM:

$$SPE_{bc} \sim Local_confidence_{bc} + Accuracy_{bc} \\ + (1 + Local_confidence_{bc} + Accuracy_{bc} | Participant) \\ + (1 | Task) + (1 | Group)$$

For measuring baseline associations of mental health scores with confidence (local confidence and global SPEs) and accuracy, we used confidence values from the baseline blocks (mean value for local confidence) and used the following LMM:

$$Confidence \sim MH_score + Age + Gender + Accuracy + Staircase_level \\ + RT_Zscored + (1 + MH_score | Task)$$

Note that despite confidence and accuracy being bounded variables, here we used linear mixed regressions instead of logistic regressions as the latter violated the homoscedasticity and normality of the residuals while the former did not. For use with linear models, we z-scored confidence and accuracy values to transform them to a continuous scale. For Exp 1, separate regression models for PHQ, GAD and mini-SPIN were used as *MH_score*, while for Exp 2, we used transdiagnostic dimensions with $MH_score = AD + CIT + SW$.

For a model-free test of distortion in learning from local confidence, we tested if the linear relationship between local confidence and global SPEs differed with anxious-depression scores. Simulations showed that the distortion in this relationship was particularly pronounced at intermediate-to-high levels of local confidence (i.e., z-scored confidence > 0; Supplementary Figure 8A). We tested this distortion as a 2-way interaction between *SPE* and *MH_score* in predicting (within individual) z-scored local confidence using the following linear mixed model:

$$Local_confidence_Zscored \sim SPE * MH_score + Accuracy + (1 + SPE \\ * MH_score | Trial_number) + (1 | Block_number) \\ + (1 | Participant)$$

Note that here we predicted local confidence from SPE and not vice-a-versa as there were a larger number of data points for local confidence to aid in statistical analyses. The validity of this effect was confirmed in simulation.

For a model-free test of distortion in learning from feedback, we tested if the presence of feedback heightened the reduction of global SPEs compared to no feedback, which we tested separately for positive and negative feedback (against no feedback):

$$SPE_Zscored \sim Feedback_type * MH_score \\ + (1 + Feedback_type * MH_score | Participant) + (1 \\ + Feedback_type * MH_score | Block_number)$$

To test for the lack of feedback distortion (i.e., test in favour of null hypothesis), we calculated Bayes Factor using the function from the *BayesFactor* package in R for the *Feedback_type* by *MH_score* interaction. This was done by comparing the above regression model with one where the interaction was removed.

To test if confidence (local and global) or mental health scores predicted self-endorsement of valenced (positive/negative) words, we used the following generalised linear regression model with a binomial link:

$$\text{Word_endorsement} \sim \text{MHQ_score}[/\text{Confidence}] * \text{Word_valence} \\ + (1|\text{Word_number})$$

For Exp 1, *Word_endorsement* included all presented words, while for Exp 2, endorsements were only used from the (pre-feedback) baseline measurement.

To test if feedback impacted self-endorsement of positive and negative words, we used the following regression model:

$$\text{Word_endorsement_diff} \sim \text{Feedback_type} * \text{Word_valence} + (1|\text{Word_set}) \\ + (1|\text{Participant}) + (1|\text{Task})$$

Here *Word_endorsement_diff* was computed as the difference in mean word endorsements for positive and negative words after and before feedback.

There were two sets of words (*Word_set*) presented to participants before and after the feedback block whose order was counterbalanced across participants.”

In the Results section, we now also update the language we use when describing regression models and clarify both the predictors and dependent variables.

Linear models do not necessarily require data to be normally distributed, but in certain cases non-normal data can impact other important assumptions such as the homoscedasticity of residuals. We have now checked each model for

Figure R7. Histogram from Exp 2 (qualitatively similar patterns were observed from Exp 1) of **A)** Local confidence as reported by participants on individual trials across all blocks and all participants. **B)** Local confidence from which baseline block values are subtracted. **C)** Local confidence z-scored after averaging across trials within blocks. **D)** z-scored global SPEs.

homoscedasticity and normality of residuals. To ensure that we are running the correct statistical analyses for bounded variables, we have also now modified our analyses in several ways.

First, whenever we test for within-subject variations in local confidence, global confidence, or accuracy, we now use the baseline-subtracted values. This transforms our data from a $[0, 1]$ scale to a $[-1, 1]$ scale, and importantly leads to our data being approximately normally distributed around zero, rather than clustering at the scale bounds. For example, Figure R7 shows that while our individual-trial confidence data are highly skewed (Figure R7A), they become normally distributed following baseline subtraction (Figure R7B).

Second, when testing for between-subject effects (e.g., mental health scores) on local confidence, we previously used a regression model in which trial-by-trial confidence data were entered as the dependent variable. Because this data is highly skewed as shown in Figure R7A, we now instead employ z-scored local confidence averaged over (baseline) blocks, resulting in non-skewed data appropriate for entry into with linear mixed models (Figure R7C). Similarly, we now also z-score SPE data onto a continuous scale for use within regression models (Figure R7D).

The reason we obtained >20,000 degrees of freedom for the mental health regressions predicting local confidence (and >15,000 degrees of freedom for the feedback-to-confidence transfer analysis) was due to employing trial-by-trial confidence data as mentioned above in the regression models (while also not exhaustively modelling random slopes). This issue is rectified in our new analyses where we now use mean blockwise local confidence data while also modelling the relevant random slopes, ensuring degrees of freedom are comparable to the number of participants in the analysis. As a result, there is a slight change in the results presented in Supplementary Materials from the previous submission where we test for a domain-general transfer of feedback to test-block local confidence, though the updated results are still qualitatively similar to the ones reported earlier. Specifically, in Exp 1, we no longer observe a domain-general effect of the perception task intervention on subsequent memory task confidence if we consider the average local confidence over all 40 trials of the test block. However, we do observe such a transfer effect if we only consider average local confidence over the first half of trials of the test block (this limited duration of domain-general transfer effects was also the reason for choosing 20 trials for the test block in Exp 2). The transfer effect was still observed in Exp 2 after reanalysis.

As per your suggestion, we have also now changed the way we compute p-values – instead of Satterthwaite, we now use likelihood ratio tests. Reassuringly, the difference in p-values between the two procedures was negligible.

The statement about pre-registration followed by an OSF link implied an OSF pre-registration. It may be more transparent to state that this is actually a custom file uploaded, not a formal OSF pre-registration. There are some details about deviations from the pre-registration in the manuscript. I was surprised to notice that the pre-registration includes an analysis of Experiment 1 that differs from what is presented in the manuscript. Notably, the number of participants. Was there a mistake? Were the exclusion criteria changed after the pre-registration? Were participants added later? The reason should be noted in the manuscript in case some readers infer that there was p-hacking.

Thank you for noticing this oversight. First, as per your suggestion, we have clarified stating that our preregistration was in the form of a publicly available document at page 6, line 99 as follows:

“These questions were addressed first in an exploration sample (Exp 1, N = 230), and then in a validation sample (Exp 2, N = 278) with hypotheses and analysis plans preregistered prior to data collection (preregistration document uploaded to osf.io/7xfqw).”

Second, the reason for the deviation in sample size from preregistration is because we modified the exclusion criterion with regards to task performance. In the preregistration we used a closed interval (.60 .85) to restrict any subjects whose performance on the perception and memory tasks was outside this interval in even a single task block. This criterion was devised based on experimental parameters of Exp 1. However, because in Exp 2 we had only 20 trials (instead of 40 trials) on the test blocks, this criterion ended up being overly conservative (see the paragraph below for the specific reason). Therefore, we modified the performance exclusion criterion to a part open interval (.60 .85] for analysing Exp 2, and to be consistent

with analyses across the two studies we also used this slightly modified criterion for analysing Exp 1 in the manuscript. We now clarify this reason for the deviation in sample size from preregistration under the *Exclusion* subheading in Supplementary Methods (Page 3), as follows:

“Note that the sample size in Exp 1 deviated slightly from our preregistered analysis. In the preregistered analysis we excluded participants whose performance was outside the closed interval (.60 .85), whereas for the reported analyses here we used a part open interval at the upper end (i.e., we included participants whose performance was exactly .85 on a block). This decision was made prior to data analysis for Exp 2 and was done because unlike Exp 1 where all blocks had 40 trials (and each correct trial would correspond to a .025 increment in accuracy), in Exp 2 the test blocks had only 20 trials (each correct trial corresponding to a .05 increase in accuracy). This would result in a more stringent exclusion criterion in Exp 2 as participants would be excluded from the study for getting only 3 (vs. 6) wrong responses on even one of the two test blocks. Consistent with this idea, we found that the preregistered criterion resulted in a performance-based exclusion of 39% of the participants compared to our expected performance-based exclusion rate of 18% from Exp 1. However, we find that even if we use the more stringent (i.e., preregistered) exclusion criterion, all our key results – namely feedback manipulation of SPE, greater sensitivity of SPEs to low vs. high confidence with higher anxious-depression scores, lack of difference in sensitivity of SPEs to negative vs. positive feedback with higher anxious-depression scores – remain the same in both Exp 1 and 2.”

As we mention in the section of manuscript text reproduced above, all our key findings remain significant even when using the more stringent exclusion criterion specified in the preregistration. Notably, our main finding of a distortion in confidence learning being predicted by anxious-depression symptoms remained unchanged, as shown in the Figure R8 (similar to Figure 4g from the manuscript), but now with the more conservative exclusion criterion.

The null model, D0 uses local confidence to update global confidence, line 632. Local confidence is lower in high AD, Line 167. Why does D0 predict no difference in SPE between high/low AD (Figure 3B)?

This is an astute observation that we are glad to have the opportunity to clarify. The reason D0 does not feature differences in global SPEs as a function of AD symptoms is because our models seek to model the impact of confidence/feedback on global SPEs after baseline-correcting for between-subject differences in local confidence. To do this, we used within-participant z-scored values of local confidence in these analyses and model fits. As a result, any differences in local confidence related to AD scores are eliminated before the local confidence values are input into the model.

There should probably be some statistics to support the claims on lines 155-158, or leave it all to the supplementary.

Thank you for your suggestion. We have now removed these lines from the main Results and only provide details in the Supplementary.

3. Do the introduction and methods give sufficient information for understanding the experiment?

What is “difficulty level” (line 132)? The methods state “we used the mean value of the last 5 reversals of staircase level as the starting level of task difficulty used for the main part of the study” so it should be the same in both feedback conditions. There is also a supplementary analysis on “difficulty reached”. Was the staircase continued throughout, contrary to what is stated in the methods? If the staircase was continued, why would there be any effects on mean accuracy?

Thank you for pointing out the lack of clarity about the staircase procedure. We now clarify in the Results that by difficulty level we mean staircase level. In the Methods, we now clarify that the staircase was continued throughout the six blocks, as follows (page 26, line 545):

“Then, to ensure stable performance despite potential learning effects, the staircase procedure was continued through the six blocks where we again used the mean of the last 5 reversals (or fewer, if there were fewer reversals) from each block as the starting level for subsequent block of respective tasks.”

You are right that continuing the staircase should ideally ensure stable accuracy across blocks. However, it is possible that accuracy may vary in certain scenarios (for example, if the feedback manipulation impacts performance and there is an insufficient number of trials for the staircase to plateau). Hence, we still analysed any potential differences in accuracy in our Results to confirm that our staircase worked as expected.

In the introduction, I experienced some difficulty following what was the purpose or the main story being told. Perhaps this could be made more specific to the hypotheses tested in this experiment? It would also help with the flow of the results

section if the introduction more closely followed the structure of the specific hypotheses tested in this experiment.

Thank you for raising this point. Based on your and other reviewers' comments we have now substantially revised the Introduction to provide a clearer narrative and overview of our hypotheses. Notably, in the second paragraph we now sequentially build up the concept of global confidence formation, and highlight the factors that influence it, including local confidence and feedback, as follows (page 3, line 52):

“One promising route to understanding the source of these metacognitive biases is to unpack how confidence is formed, particularly at a global level. Several factors have been shown to influence confidence formation over and above objective performance. For example, induction of negative vs. positive affect or low vs. high reward expectations can decrease vs. increase local confidence, respectively^{10–12}. Manipulating beliefs about performance on an upcoming task, either through feedback or expected task difficulty, affects local confidence on subsequent instances of a similar task¹³. Finally, local confidence is higher on trials immediately following positive compared to negative feedback¹⁴. With respect to global confidence, false performance feedback in an ambiguous task has been found to influence people's global SPEs¹⁵. Similarly, providing true performance feedback, compared to a no-feedback condition, can increase global SPEs, despite objective task performance remaining unaffected^{7,16}. At the same time, local confidence remains a robust predictor of global SPEs in the absence of feedback^{7,16,17}. Together these findings have informed a computational model in which global confidence is formed by probabilistically combining instances of local confidence with feedback to update a prior over expected performance⁷ (Figures 1a–1c).”

In the third paragraph, we connect the existing confidence literature to our subsequently stated hypotheses (page 4, line 70):

“One attractive hypothesis is that underconfidence may be grounded in a tendency to incorporate more negative and less positive information when making future predictions, particularly about oneself^{18–22}. Recent work has identified a self-related negativity bias in individuals with anxious-depression symptoms in relation to learning about expectations of future adverse life events^{21,23} and when updating performance expectations from feedback²². Such biases could predispose individuals with anxious-depression symptoms to distort individual instances of feedback and/or local confidence when forming global SPEs. Specifically, when forming a global sense of confidence, individuals with anxious-depression symptoms may be more sensitive to negative (compared to positive) external feedback, low (compared to high) local confidence, or a combination of the two (Figure 1d).”

Finally, in the fourth paragraph we now more explicitly mention our hypotheses, as follows (page 4, line 87):

“In the present study, we sought to determine how computations underpinning global confidence formation are altered in individuals with anxious-depression

symptoms. We systematically manipulated performance feedback to influence global confidence. Then, using computational modelling, we distinguished between several mechanisms of how anxious-depression symptoms distort the impact of local confidence and performance feedback on global confidence. Specifically, such distortions could manifest as altered sensitivity to local confidence, altered sensitivity to feedback valence, a general negative response bias, or a combination of these factors. Each of these three mechanisms makes qualitatively distinct predictions for the patterns of behaviour we expect to observe in our experiments (see Figures 1e–1g for schematic illustration).”

4. Does the discussion sufficiently outline the findings?

This was difficult to assess because of the lack of clarity of the results section.

We hope that there is now sufficient clarity to allow a straightforward reading of the Discussion section following the new changes to the Results (and Methods) sections in the revised submission.

Minor:

“Replicate” may be used a little too liberally. Normally “replicate” means the same method and results were found. At least the same results. At line 136: “Replicating previous findings^{9,26}, mean local confidence (metacognitive bias) significantly predicted global SPEs” reference 9 refers to a study that found local confidence sensitivity predicted global confidence sensitivity (not bias), and reference 26 refers to a study that did not even measure local confidence. Experiment 2 is called a “replication sample”, but there are several differences in the methods of Exp 2 and Exp 1. It may be better to call it something like a ‘partial replication’ – some of the methods were replicated, other aspects were not.

Thank you for pointing this out. It was a mistake on our part to cite Lee et al in relation to this finding; we have now removed the citation from this line (while retaining it in relation to other relevant text). You are correct that Exp 2 exhibited slight methodological differences from Exp 1, and so was a conceptual replication rather than direct replication. Our reason for labelling Exp 2 a replication sample was to highlight this attempt to replicate key findings, despite these methodological differences. However, we do see the concern that this label could be interpreted as signalling to the reader that the methods were identical. We have now changed this label to a “validation sample” on lines 100 and 399. At another location, we have changed “Exp 2 was a replication of Exp 1” to “Exp 2 attempted to replicate the key findings of Exp 1 with minor differences to the design.” Note however that we still retain the word replication in relation to the key results that we replicated from past studies.

Figure 2A mentions the ‘baseline’ in the legend, but the y-axes are misleading (at first impression it looks like participants had negative ‘actual performance’).

Thank you. We have now mentioned ‘baseline-corrected’ on the y-axis in relevant plots in Figure 3 (which was previously Figure 2).

Reviewer #3 (Remarks to the Author):

The authors conducted two experiments (one exploratory and one preregistered) and used computational methods to investigate the sources of metacognitive biases and their relationships with anxiety and depression (AD) symptoms. Specifically, they carefully manipulated feedback valence and found a domain-general effect of feedback valence on confidence. They found that individuals with stronger AD symptoms tended to put heavier weights on low local confidence responses in the process of computing their global self-performance estimates (SPEs).

The study asks an important question concerning the bias in self evaluation in relation to AD symptoms. It has implications across basic sciences and clinical areas. The methods are appropriate and solid. I am most impressed by the rigor in the designs of the behavioral tests and the computational methods. The model-comparison approach is creative. In general, the study is a great example of how to combine behavioral and computational methods to address a question that interests researchers and clinicians across fields.

I only have one major comment, plus a few minor comments/suggestions.

MAJOR:

The manuscript is drawing conclusions about individuals with “anxiety and depression”. I suppose the authors were referring to the clinical conditions of anxiety and depression. But it seems that the samples were taken from a general population instead of a clinical one. It is of course not necessary for the study to have a sample clinically diagnosed with AD conditions in order to draw such conclusions, but individuals with clinically-diagnosed AD could demonstrate qualitatively different patterns of behavior (and perhaps different computational processes as well). Perhaps it will be helpful to mention this as a possible limitation in the Discussion section. Another more transparent and informative way to address this point is to report (perhaps in supplementary info) some descriptive statistics of scores related to the AD symptoms in both Exp 1 and Exp 2. An even more helpful piece of information is to provide the clinical cutoff (if any) for each of those AD symptom scales, so that the readers will have a better idea about where the samples stand in the spectrum of AD symptoms.

Thank you for raising this important point regarding potential differences between cohorts with clinically-diagnosed anxiety and depression and our general population sample. To address this point, we have made the following changes:

- 1) We now avoid referring to our participants as individuals with anxiety and depression, instead referring to individuals exhibiting symptoms of anxiety and depression (there were only a few instances in which this needed to be corrected, most notably in the title and the abstract).
- 2) We have now included this as a potential limitation of our work in the Discussion, and highlight the need for future studies in clinical populations, as follows:

“Another limitation is that we used a general population sample from an online participant recruitment platform. While the sample contained a substantial number of individuals who surpassed clinical cutoffs on standard anxious-depression scales, further work is required to test if our findings also generalise to individuals with clinical diagnoses of anxiety and/or depression³⁹.”

- 3) We now also include more information regarding how our key results behave in relation to clinical cutoffs in a new subsection of Supplementary Results titled “Mental health scales and clinical cutoffs,” as follows:

“In Exp 1, we used standardised clinical scales for depression (PHQ-9) and general anxiety (GAD-7). Supplementary Figure 22 shows histograms of scores on the PHQ and GAD separate by clinical cutoffs for these scores. Median scores for both PHQ and GAD were 4 (black vertical line in Supplementary Figure 22) indicating that nearly half the participants had greater than minimal severity for depression and anxiety. According to Kroenke and Spitzer⁷, for the PHQ-9, the single recommended cut-off point is ≥ 10 with a sensitivity of 88% for major depression with a specificity of 88%. In Exp 1, 24.3% of the participants had PHQ scores ≥ 10 . Similarly, for the GAD-7, the single recommended cutoff is again ≥ 10 (sensitivity 89%, specificity, 82%) – 20% of the Exp 1 participants had GAD scores ≥ 10 .

In Exp 2, we used a questionnaire battery that included subsets of questions from different standard clinical questionnaires to extract transdiagnostic symptom dimensions⁴. Because the full depression and anxiety questionnaires were not collected, we could not evaluate our sample in terms of standard clinical cutoffs.”

Minor comments/suggestions:

Behavioral tasks

1. In Figure 2A, consider adding a panel for the local confidence responses as well. Currently, it feels like some of the basic measures (like the local confidence responses and the scores on the AD symptom scales) are missing in the manuscript, but there are a lot of graphs and analyses for the “secondary” outcomes (e.g., fitted parameters and the symptom-bias betas).

Thank you for this suggestion. We have now added several new figures that ensure all the basic measures are clearly described and plotted. First, as per your suggestion, we have added a panel for local confidence as part of Figure 2. We have also added plots of our raw data comprising global SPEs, local confidence, and accuracy across the six blocks and eight groups (Supplementary Figures 3, 4 and 6), together with histograms and clinical cutoffs for the anxiety and depression symptom scales from Exp 1 (Supplementary Figure 22; in Exp 2, as mentioned above, mental health scores are not clinically normed). We also now include scatter plots depicting baseline relationships between local and global confidence and mental health scores (Supplementary Figure 7). Finally, we have now modified Figure 4 to depict the data in a manner that should reflect confidence and feedback learning distortions.

2. Have the authors considered analyzing individuals’ metacognitive sensitivity? The

data seem to be rich enough for that to be computed at both local (e.g., the SDT approach like meta-d' approach or type-2 AUROC, etc.) and global level (e.g., correlation of SPE and actual performance)? This might further lengthen the manuscript (especially the supp materials) but it will be interesting to see how that may be related to the metacognitive biases identified and the AD symptoms.

Thank you for this suggestion. We have now included these analyses in the revision of the paper. As expected, and consistent with previous literature (Rouault et al, 2018; Benwell et al, 2022; Hoven et al, 2023), we did not find reliable relationships between metacognitive efficiency as measured by M-ratio (meta-d'/d') and mental health scores. This analysis is provided in the new subsection "Metacognitive efficiency not related to mental health symptoms" in Supplementary Results, as follows:

"We explored if metacognitive efficiency, as measured by M-ratio (meta-d'/d'), was related to mental health scores. M-ratio was calculated for the baseline blocks separately for each subject and task. We then used the following regression model to assess if mental health scores (MHS) predicted M-ratio:

$$M_ratio \sim MHS + Age + Gender + (1|Task)$$

In Exp 1, neither PHQ ($\chi^2 = .07$, $p = .79$) nor GAD ($\chi^2 = .63$, $p = .43$) scores predicted M-ratio. Similarly, in Exp 2, none of AD ($\chi^2 = 2.60$, $p = .11$), CIT ($\chi^2 = 3.18$, $p = .08$) or SW ($\chi^2 = 2.34$, $p = .13$) axes predicted M-ratio. These results are consistent with previous literature, which shows a lack of relationship between metacognitive sensitivity and mental health symptoms^{5,8}."

We also agree with the reviewer that devising a measure of metacognitive efficiency at the global level would be very interesting indeed. However, we were unable to pursue this analysis here due to a limited number of datapoints for calculating within-subject correlations between SPE and performance.

3. Consider reporting effect sizes as well (e.g., Cohen's d or Hedges' g), especially for the key effects (e.g., effects of feedback valence). This will be helpful for future meta analysis studies.

Thank you for this suggestion. We have now provided Cohen's d for the effect of feedback valence on global SPEs.

Computational models / Model comparison

1. It may be helpful to show the distributions of the symptom-bias betas for all models fitted. It will make it transparent to readers about which parameters may be more stable across models (somewhat related to point 3 below).

Thank you. Based on this suggestion, we have now added Supplementary Figure 12 that shows symptom-bias beta distributions for all the fitted models.

2. Why are results from D4 (instead of those from D6, which is the best model) plotted in Figure 4F? The beta-confidence and beta-feedback are indeed interesting, but so is beta-additive, especially if D6 has been shown to be the best model in both Experiments.

We agree, and had previously deliberated over this point before making the figures. Our reason for presenting model D4 instead of D6 (which we now label D5 in the revision) is that, unlike for the confidence and feedback learning asymmetries, it is hard to interpret how a small positive response bias β contributes to underconfidence. While we agree that an additive bias (which we have now renamed response bias in the revision) may have been an important explanatory factor if the β was significantly negative, a positive response bias β is unable to account for the global underconfidence effect. Additionally, and as we reported in the manuscript, the difference in SPEs between anxious-depression symptoms explained by the response bias β is negligible. Therefore, we decided to depict the estimates for the confidence+feedback distortion model in a main manuscript figure, and estimates for the confidence+bias model in Supplementary Figure 11.

3. How would a D7 model that includes all bias components (feedback + confidence + additive) compare with others? In fact, doing so may allow an exhaustive factorial comparison of the bias components as possible factors, which may be a more systematic analysis.

Thank you for this suggestion. Our reason for not including all 3 bias components in the model was that the extra free parameter has a detrimental impact on recovery of the regression parameters. This is illustrated in Figure R9 below where the upper panel shows the simulated feedback asymmetry regression slope (β_f) and the lower panel the confidence asymmetry regression slope (β_c). On the x-axis we show the simulated values of the parameters and on the y-axis the values recovered from fitting the model. In blue we illustrate parameter recovery when the model is fitted with two regression parameters (feedback and confidence asymmetry), showing good recovery. In contrast, in red we show parameter recovery when the model fitted with all three regression parameters (i.e., feedback and confidence asymmetries along with additive bias). Dots represent a range of values of the regression intercept and slope parameters. As can be seen in these plots, the recovered values are substantially noisier for the 3-parameter than the 2-parameter model, and also negatively biased.

Figure R9. Parameter recovery of regression slope parameters linking symptoms to feedback distortion (above) and confidence distortion (below) with model recovery performed with either 2 free parameters (blue) corresponding to feedback distortion and confidence distortion betas or 3 free parameters (red) corresponding to feedback distortion, confidence distortion and response bias betas. Simulated values on x-axis and model recovered values on y-axis.

For completeness, however, we have now fitted the model with all 3 bias components as per your suggestion, and reassuringly we find that our findings do not change. We again find the β -confidence parameter to be negative, the β -feedback parameter to be not different from zero and the β -response bias parameter to be slightly positive. Due to the weaker parameter recovery for this model, we prefer not to include this model in the manuscript.

4. Consider plotting Fig 4E on separate scales of DIC for Exp 1 and Exp 2. Currently the differences across the dark bars (Exp 2) are not obvious with the huge DIC values of Exp 1.

Thank you for this suggestion. We have now added DIC scores in the two experiments as two separate panels with different axes.

REVIEWER COMMENTS

Reviewer #1 (Remarks to the Author):

I am expressing my sincere appreciation for the thoroughness and seriousness with which the authors addressed all of my questions and concerns regarding the paper. Their dedication to making substantial textual revisions, creating supporting figures, and providing crucial statistical details has undoubtedly strengthened the paper.

I believe these changes have not only addressed concerns but have also enhanced the paper's overall appeal to a broader audience. The revisions make it a more compelling read and significantly improve its accessibility. I expect that it is now positioned for a positive and impactful reception within the academic community.

Reviewer #2 (Remarks to the Author):

Review of manuscript NCOMMS-23-30273A titled “How underconfidence is maintained in the face of reality” for Nature Communications

The authors have responded well to the last round of reviews, and the manuscript is much improved. There are still a few points which remain unaddressed, which are important for the scientific integrity of the manuscript.

1. Critical claims of the manuscript rely on the model comparison, but the manuscript does not present evidence that A) the winning model provides a good description of the data and B) that the model comparison analysis guarantees the conclusions. A) can be addressed by adding the model predictions to Figure 4a,b,c,d – simply add markers of a different colour describing the simulated global SPE from the winning model fit to the data. This way the reader can see that the model is a good description of the data (not simply the best of some models that are not a good description of the data). B) can be addressed by adding a model recovery analysis. The authors can simulate data from each model, fit each model, and show that the winning model is guaranteed to win when the ground truth is that model, and the winning model would not win when the ground truth is a different model (i.e. that the model comparison result is unlikely to result from a type-I or type-II error). This technique is recommended for modelling analyses and outlined more fully in: Wilson, R. C., & Collins, A. G. (2019). Ten simple rules for the computational modeling of behavioral data. *Elife*,

8, e49547. (The suggestion A relates to the section “Validate (at least) the winning model”; the suggestion B relates to the section “Can you arbitrate between different models?” in particular, Box 5. The authors may also want to adjust their current parameter recovery analysis from Supplementary Figure 16 in line with the suggestions in box 4 and 6).

2. The new Figures 1 e-g and 4 a-d nicely demonstrate the behavioural effects predicted by the models. However, I cannot work out what “confidence (tiled)” means. Are these points from different participants? Or selected blocks within participants?

3. Some statements in the manuscript are not supported, or overstated. Prominently:

a. Line 314: “An asymmetric learner provided good fits to our data across both experiments” It provided the best fit across selected models, but the analysis in suggestion 1A is required to suggest it is a ‘good fit’.

b. Line 337: “In both experiments, we indeed found that an impact of feedback on confidence persisted beyond intervention blocks to also affect confidence on “test” blocks without feedback”. This refers to a supplementary analysis where, for both experiments, the effect was a “trend” ($p = 0.057$ for Exp 1; $p = 0.07$ for Exp 2), with significant effects only for the perception task. The authors should qualify this claim only refers to feedback on confidence in the perception task.

c. Line 341: “Strikingly, this effect of feedback on test block confidence was observed not only when the two tasks were the same, but also when they differed”. This refers to the same analysis in the supplementary materials, where the authors show some evidence for a small perception-to-memory transfer only if they select particular trials, but no evidence for memory-to-perception transfer, and in fact little evidence for memory-to-memory transfer. The authors should tone down this claim, it is not a “striking” effect but a small effect that may not generalise to the kind of real-world scenarios they refer to in this paragraph.

4. The results section of the main manuscript should report the statistics underlying claims of evidence for an effect: The claims at Lines 168 and 178. It would also be helpful to number the sections of the supplementary materials and refer specifically to those, so that the interested reader can easily find the analyses that the authors refer to.

5. Readers may read the manuscript in the order presented (results before methods) so it is important to state what the test was in the results section. Somewhere before line 140 state that linear regression models were used for the analyses, or at least that the statistics in the following line refer to a linear regression analysis.

6. The statement at line 100 “with hypotheses and analysis plans pre-registered prior to data collection” implies the analysis plan was followed. It would be important to mention this was not the case (at first mention of the pre-registration), it looks like most of the presented analyses in this manuscript are different from what was pre-registered (from the exclusion criteria to the R-package used for analysis and the very metrics analysed). The main manuscript should also state explicitly that the exclusion criteria for experiment 1 were changed based on the data from experiment 2.

Minor:

Line 603: “Note that despite confidence and accuracy being bounded variables, here we used linear mixed regressions instead of logistic regressions as the latter violated the homoscedasticity and normality of the residuals while the former did not. For use with linear models, we z-scored confidence and accuracy values to transform them to a continuous scale.” If data were z-scored to a continuous scale then the logistic regression would not be appropriate.

I appreciate the change of title, but the use of the term “reality” does seem a bit strange given the experiment puts participants in the fictional realm of “Fruitville”. Maybe “... in the face of accurate positive feedback” would be more specific, or just “...in the face of feedback”?

Reviewer #4 (Remarks to the Author):

Summary

I was asked to supply an additional review for this manuscript; thus, my summary and comments come after having reviewed (1) the manuscript, (2) the reviewer comments, and (3) the response to the reviewers. I wrote my review before reading the reviewer comments; where my review was informed by their comments, I note it explicitly below.

In this paper, the authors systematically manipulated performance feedback to influence “global confidence” (i.e., long-run self-performance estimates (SPE)). Game-ified tasks were conducted online, and included both a perceptual task and a memory task. The perceptual task required participants to decide which of two types of berries (raspberries or blackberries) were more numerous. The memory task required participants to decide which fruit was present in a box of fruits that was recently opened. The key manipulation involved the extent to which participants received veridical positive or negative feedback; some blocks contained more positive than

negative feedback, and vice versa. Baseline blocks were followed by intervention and test blocks; feedback type was a within-subjects variable, and order effects were manipulated across individuals, resulting in eight groups of participants.

Computational modeling was employed to determine how anxiety and depression symptoms alter local confidence/feedback's influence on global confidence. As shown in figure 1d-g, either a confidence learning distortion, feedback learning distortion, or a response bias could account for how global SPEs differs as a function of anxious-depression symptoms. The work was conducted in an exploratory manner in Experiment 1, and that validated in a preregistered Experiment 2. Results showed that individuals with greater anxious-depression symptoms exhibited reduced sensitivity to increases in local confidence, and a confidence distortion plus response-bias model provided the best fit to data in both Experiments 1 and 2.

Overall, I found the paper quite interesting, as it addresses an important and timely question. However, I do have some concerns about how the modeling results are portrayed, as well as whether the anxiety-depression scores may have been influenced by the experimental procedures. I note my major and minor criticisms below.

Major Concerns

- In Supplementary Figure 9, it is interesting that the confidence learning distortion (D1) and Response Bias (D3) models make opposite predictions (fig. S9a and S9c) about the slope between the low and high anxiety-depression symptom lines for confidence (tiled) 3-4. Your behavioral data in Figure 4a shows a significant difference between these two lines, supporting the confidence learning distortion account, and refuting the response bias account. Yet the model that contains BOTH the confidence and response value appears to win, according to DIC values. I assume this is because of the nonsignificant values shown in Figure 4c and 4d of the effect of feedback type on global SPE. Supplementary Figure 9e doesn't say anything about significance, unlike 9f and 9g. Thus: can you please clarify what is predicted (in terms of significance values) in Supplementary Figure 9e? And can you please comment in the Discussion section on how your "winning" model actually contains diametrically opposing predictions as to what the behavioral data in Experiment 1 and Experiment 2 should look like in Figure 4a and 4b? This issue became clear to me after reading the comments from Reviewer 2 on this topic. In Figure R6 that you sent to this reviewer, you are plotting the data that shows support for the confidence learning distortion in R6a and Rb, and support for the response bias effect in R6c and R6d, but you are glossing over the issues I referenced above, and considering the diametrically opposing predictions in Figure S9a and S9c, it warrants further commentary.

- Not a single model predicts a flat slope between confidence values 1 and 2 for the "low" anxiety-depression group, and yet this is what the data in Experiment 1 shows in Experiment 4a. What do you make of this? Is it simply that participants struggle to distinguish between confidence values 1 and 2 in this task? If so, why do you think you don't see that same trend in Experiment 2?

- p.9 – It appears you have discrepant results between Experiment 1 and Experiment 2 regarding whether feedback type influences local confidence. Any speculation as to why there is a difference across the two experiments?
- p.11 – You claim that “across both experiments, we replicated previous findings showing that heightened anxiety and depression symptoms are associated with lower average local confidence (metacognitive bias) and lower global SPEs (Figure 3d; Supplementary Figure 8). However, when I look at supplementary figure 8, panel A does not reveal a significant correlation between the GAD-7 score and the perceptual task local confidence, and panel D does not reveal a significant correlation between the PHQ-9 score and the memory task global SPE. This would warrant some commentary to scale back your claim in the paper, yes?
- Can you explain why you use the GAD-7 score in Exp 1 and “transdiagnostic” scores in Experiment 2, before you introduce the results in the manuscript?
- You mention on p.26 that in the final phase of your experiment, participants completed the mental health questionnaires. Isn’t it possible that putting these questionnaires at the end influenced how participants would respond? For example, since your data in Figure 5 shows that the feedback type influenced self-endorsement of words of different valence types, why do you think that assessments about mental health would be robust to the different types of feedback provided during the task? I understand the issue in play: if you put these things first, it can induce demand characteristics shaping how participants perform the task. But when you put them last, don’t you also need to analyze whether having positive or negative feedback in the last block before taking these assessments influenced participants’ scores?

Minor Edits

- I was a bit confused by Model D2 in Figure 1f. The way model D1 is described in the caption is that people with higher anxious-depression symptom scores (blue lines) show blunted influence of HIGHER local confidence on global SPEs compared to lower anxious-depression symptom scores (gold lines). The reduced slope of the blue solid line makes this apparent. In Model D2, the idea is that higher anxious-depression symptoms are associated with a greater sensitivity to incorrect vs. correct feedback. The low global SPE in the “Incorrect” column on the x-axis makes that prediction apparent, but I wasn’t sure how you determined what the right baseline expectation for “Correct” feedback would be under this model. The schematic makes it look like the global SPE for “Correct” feedback is quite comparable to the “None” feedback condition, but couldn’t model D2 also predict a “normal” amount of adjustment to the Global SPE in the “Correct” feedback category, while maintaining greater amount of sensitivity to the incorrect feedback? The way you depict it now, it almost appears that there is both INCREASED sensitivity to incorrect feedback, AND DECREASED sensitivity to correct feedback in the D2 model. In this way, it seems like there is potential for a “D2a” and “D2b” model, where the magnitude of sensitivity to the “Correct” feedback is either normal, or blunted (the way you have it now), respectively. Do you agree? P.11 describes versions of your three original models that combine predictions from one another, but I’m wondering if an additional model that includes this element is warranted.

- I was a bit confused by the first two “until response” panels in the perceptual trials on p.8. Why do the labels for the confidence ratings only appear in the SECOND “until response” panel? Does this mean that those labels appear AFTER the participant first moves the slider? Or are they giving two responses? (One without labels, and one with labels?) It’s not clear based on the current caption. The same criticism applies to the memory trial part of Figure 2b, and needs to be clarified as well. p.23 provides a hint as to what was going on: was this when participants were allowed to switch their choice? If so, this should probably be briefly referenced in the caption.
- On the perception task trials, it looks like one color is light, while another color is dark. I didn’t see any reference in the manuscript about trying to control for luminance across the two choice options. My understanding from Doby Rahnev’s work using a similar “choose the more numerous color” task (e.g., Haddara and Rahnev, Psych Science, 2022) is that this is a factor that may influence confidence judgments. If you didn’t control for luminance, why do you think this is a factor that is inconsequential for your research?
- Can you clarify in figure 4a and 4b whether the p-values labeling the difference between the gold and blue lines are for differences in SLOPE between the two lines, or some other value? Based on the description in the manuscript on p.12, I’m assuming it’s slope, but I was hoping to see it labeled in the caption, too.
- Did participants earn money for completing your task? I didn’t see any listing of how much they earned for completing it in the participants section of your manuscript. Also, what was the IRB approval information?

Reviewer #4 (Remarks on code availability):

I briefly reviewed the code, but I did not have time to run it on my machine, to verify their results. The documentation seems reasonable, and includes a brief "ReadMe" file, as well as links to versions in both MATLAB and R.

Response to Reviewers: **How underconfidence is maintained in the face of reality**

We thank the Editor and the three Reviewers for their helpful and constructive comments. We are pleased that Reviewer 1 had no further concerns, and we appreciate the time taken by Reviewer 2, and new Reviewer 4, to provide constructive feedback. This has been very useful in strengthening the final version of the paper.

We provide a point-by-point response to each comment below. The Reviewers' comments are in standard font, our responses are in **green**, and excerpts from the revised manuscript are in **blue**.

Reviewer #1 (Remarks to the Author):

I am expressing my sincere appreciation for the thoroughness and seriousness with which the authors addressed all of my questions and concerns regarding the paper. Their dedication to making substantial textual revisions, creating supporting figures, and providing crucial statistical details has undoubtedly strengthened the paper.

I believe these changes have not only addressed concerns but have also enhanced the paper's overall appeal to a broader audience. The revisions make it a more compelling read and significantly improve its accessibility. I expect that it is now positioned for a positive and impactful reception within the academic community.

We are delighted to hear this, and thank the Reviewer for highly beneficial feedback.

Reviewer #2 (Remarks to the Author):

Review of manuscript NCOMMS-23-30273A titled "How underconfidence is maintained in the face of reality" for Nature Communications

The authors have responded well to the last round of reviews, and the manuscript is much improved. There are still a few points which remain unaddressed, which are important for the scientific integrity of the manuscript.

We are glad to hear this and thank the Reviewer for continued constructive feedback. We address the remainder of their concerns below.

1. Critical claims of the manuscript rely on the model comparison, but the manuscript does not present evidence that A) the winning model provides a good description of the data and B) that the model comparison analysis guarantees the conclusions. A) can be addressed by adding the model predictions to Figure 4a,b,c,d – simply add markers of a different colour describing the simulated global SPE from the winning model fit to the data. This way the reader can see that the model is a good description of the data (not simply the best of some models that are not a good description of the data). B) can be addressed by adding a model recovery analysis.

The authors can simulate data from each model, fit each model, and show that the winning model is guaranteed to win when the ground truth is that model, and the winning model would not win when the ground truth is a different model (i.e. that the model comparison result is unlikely to result from a type-I or type-II error). This technique is recommended for modelling analyses and outlined more fully in: Wilson, R. C., & Collins, A. G. (2019). Ten simple rules for the computational modeling of behavioral data. *Elife*, 8, e49547. (The suggestion A relates to the section “Validate (at least) the winning model”; the suggestion B relates to the section “Can you arbitrate between different models?” in particular, Box 5. The authors may also want to adjust their current parameter recovery analysis from Supplementary Figure 16 in line with the suggestions in box 4 and 6).

Thank you for these constructive suggestions. We have now comprehensively addressed both points as follows:

- A) We now overlay model fits and data in Figures 4a-4d (reproduced below as Figure R1). These new plots show that the model fits are a close match to the observed data within the margin of experimental error.

While constructing these plots, we noted that the error bars (confidence intervals) shown in the previous version (calculated across all trials for all subjects for each confidence tile) were overly precise and not reflective of the true variance of the data. The issue here is that while confidence (x-axis) was measured for each trial, global SPEs (y-axis) were only measured once per block. However, in the plots, each block’s global SPEs is replicated for all trials within the block – leading to an inappropriate reduction in plotted error bars. We have now resolved this issue by plotting estimates and confidence intervals from a mixed regression model where we first regress out random effects of participants, blocks, and trials (for consistency, we have also similarly replaced raw data in Supplementary Figure 9, and Supplementary Figure 10 with regression estimates).

Additionally, in the Methods section we now added a paragraph explaining how we obtain these plots (page 31, line 670):

“For plots of behavioural data in Figures 4a–4b, we obtained global SPE values, and their 95% confidence intervals, after regressing out random effects of participant, block number and trial number (separately for lower and higher values of local confidence corresponding to the lower and upper two confidence tiles shown in the figure respectively). For depicting the relationship between feedback and global SPEs in Figures 4c–4d, we regressed out random effects of participant and block number.”

Figure R1. Data from Exp 1 (left column panels a, c, e) and Exp 2 (right column panels b, d, f). Observed data are plotted as filled circles; model fits are plotted as unfilled triangles. **a & b**) Global SPEs plotted against local confidence (z-scored and split into 4 tiles per individual) separately for individuals belonging to the lowest (gold) and highest (blue) tertiles of anxious-depression symptom scores (obtained using the GAD-7 questionnaire in Exp 1 and transdiagnostically in Exp 2). Depicted statistical comparisons are for slopes of higher confidence values between higher and lower anxious-depression scores (* $p < .05$; **** $p < .0005$). **c & d**) Global SPEs plotted for blocks with more frequent negative feedback, no feedback, and more frequent positive feedback, separately for individuals with low and high anxious-depression scores. Global SPEs and 95% confidence intervals in panels a—d are obtained from linear regression models after regressing out random effects of participant, block number, and trial number.

As per the suggestion of the referee, we also performed a model recovery analysis, following the guidelines in Wilson & Collins (2019). Consequently, we add two new panels to Supplementary Figure 16, depicting the probabilities of the fitted model given the simulated model and vice-versa. We reproduce these plots here as Figure R2. When comparing an ability to recover the ground-truth simulated model [$p(\text{fit model} | \text{simulated model})$, left-hand panel], we find that both the feedback and confidence distortion models are recovered well (85% and 100% of the time, respectively), whereas the response bias model is prone to misidentification (being identified 69% of the time). However as Wilson & Collins (2019) point out, we typically are more interested in the inverse inference: given a particular model fits the data best, what is the probability that it generated the data [$p(\text{simulated model} | \text{fit model})$, righthand panel]. Here we see that all models perform reasonably well, with an approximately 85% recovery rate.

With regard to Wilson & Collins' recommendation that parameter recovery analysis should seek to match the parameter values from the observed data, our observed confidence distortion values ranged between (-.04 0) (after adjusting Exp 2 units to those of Exp 1, which is what we used to simulate our

Figure R2. c) Confusion matrix of model recovery depicting the probabilities of the best fitting distortion model based on deviation information criterion (DIC) given which of the three distortions was simulated. d) Inversion matrix of model recovery depicting the probabilities for which of the three distortions was simulated given a best-fitting distortion.

data). We conducted parameter recovery within this range, as well as within a similar range for positive values to aid future potential use of the model. For feedback distortion and response bias, we simulated values that would lead to similar magnitudes of difference in global SPEs between higher and lower anxious-depression scores. This approach is designed to show that our model can recover these distortions if they were responsible for producing differences in the observed global SPEs.

2. The new Figures 1 e-g and 4 a-d nicely demonstrate the behavioural effects predicted by the models. However, I cannot work out what “confidence (tiled)” means. Are these points from different participants? Or selected blocks within participants?

Thank you for prompting a clarification. Local confidence was divided into 4 individually-specific quantiles (i.e., each with equal probability mass) for each participant. We have now slightly changed the wording of Figure 4’s caption to “z-scored and split into 4 quantiles per individual” to clarify this point. We also confirm that plots looked qualitatively very similar if confidence data were quantiled across participants, or per block within participants.

3. Some statements in the manuscript are not supported, or overstated. Prominently:
 a. Line 314: “An asymmetric learner provided good fits to our data across both experiments” It provided the best fit across selected models, but the analysis in suggestion 1A is required to suggest it is a ‘good fit’.

Thank you for raising this point. As described above, we now show that the model fits qualitatively match the observed data (within the range of error). Nonetheless, as the term ‘good fit’ is a qualitative judgment and as it is not essential to this discussion point, we have removed it accordingly.

b. Line 337: “In both experiments, we indeed found that an impact of feedback on confidence persisted beyond intervention blocks to also affect confidence on “test” blocks without feedback”. This refers to a supplementary analysis where, for both experiments, the effect was a “trend” ($p = 0.057$ for Exp 1; $p = 0.07$ for Exp 2), with significant effects only for the perception task. The authors should qualify this claim only refers to feedback on confidence in the perception task.

c. Line 341: “Strikingly, this effect of feedback on test block confidence was observed not only when the two tasks were the same, but also when they differed”. This refers to the same analysis in the supplementary materials, where the authors show some evidence for a small perception-to-memory transfer only if they select particular trials, but no evidence for memory-to-perception transfer, and in fact little evidence for memory-to-memory transfer. The authors should tone down this claim, it is not a “striking” effect but a small effect that may not generalise to the kind of real-world scenarios they refer to in this paragraph.

Thank you for this suggestion – we agree that we should exercise greater caution on this point. We have now modified the text as follows (line 341):

“In both experiments, we found that an impact of feedback on confidence can persist beyond intervention blocks to also affect confidence on “test” blocks without feedback: during test blocks, local confidence was higher when the previous intervention blocks had more frequent positive compared to negative feedback. Interestingly, this effect of feedback on test block confidence was observed specifically when feedback was delivered on a perception task and confidence assayed on a memory task (and not vice-versa), and most prominently during the early phase of a subsequent test block. These results indicate that receiving feedback on some domains may have broader, domain-general, effects on people’s confidence while remaining more restricted in other domains.”

4. The results section of the main manuscript should report the statistics underlying claims of evidence for an effect: The claims at Lines 168 and 178. It would also be helpful to number the sections of the supplementary materials and refer specifically to those, so that the interested reader can easily find the analyses that the authors refer to.

Thank you. We have now added section numbers in Supplementary Materials and are now referring to the specific section numbers in the main manuscript. We have also added statistics supporting these effects.

Line 170 (previously, line 168) now reads,

“In both experiments, we found evidence that the feedback manipulation delivered within an intervention block continued to impact local confidence estimates in the early phase of a subsequent test block, even when the test block involved a distinct task domain (perception-to-memory transfer: Exp1 – $t(226) = 2.19$, $p = .030$; Exp2 – $t(275) = 3.12$, $p = .002$; Supplementary Figure 7; see Supplementary Material §4 for a full analysis).”

And line 181 (previously, line 178) has been changed to,

“We replicated previous findings showing that heightened anxiety symptoms are associated with lower average local confidence (metacognitive bias; Exp 1 – GAD-7: $\chi^2 = 9.76$, $p = .0018$; Exp 2 – Anxious-Depression axis: $\chi^2 = 29.63$, $p < .0001$)² and lower global SPEs (Exp 1: $\chi^2 = 24.11$, $p < .0001$; Exp 2: $\chi^2 = 20.26$, $p < .0001$; Figure 3d; Supplementary Figure 8; see Supplementary Material §7 for details).”

5. Readers may read the manuscript in the order presented (results before methods) so it is important to state what the test was in the results section. Somewhere before line 140 state that linear regression models were used for the analyses, or at least that the statistics in the following line refer to a linear regression analysis.

Thank you. We have now added a line prior to reporting the results as follows:

“For statistical analyses, we report results from linear (mixed) regression models (see Methods for details).”

6. The statement at line 100 “with hypotheses and analysis plans pre-registered prior to data collection” implies the analysis plan was followed. It would be important to mention this was not the case (at first mention of the pre-registration), it looks like most of the presented analyses in this manuscript are different from what was pre-registered (from the exclusion criteria to the R-package used for analysis and the very metrics analysed). The main manuscript should also state explicitly that the exclusion criteria for experiment 1 were changed based on the data from experiment 2.

Thank you. We now state the following at line 103,

“Due to small changes in the exclusion criteria, the sample size for Exp 1 deviated from the one reported in our preregistration. All deviations from the preregistered plan, are detailed in Methods and Supplementary Methods.”

Minor:

Line 603: “Note that despite confidence and accuracy being bounded variables, here we used linear mixed regressions instead of logistic regressions as the latter violated the homoscedasticity and normality of the residuals while the former did not. For use with linear models, we z-scored confidence and accuracy values to transform them to a continuous scale.” If data were z-scored to a continuous scale then the logistic regression would not be appropriate.

We understand the confusion. We intended to say that we first compared linear and logistic regressions without z-scoring confidence and accuracy and subsequently used linear models with z-scored values. We have now modified these lines (line 609) to read

“Analyses were performed using linear mixed regression models while ensuring that assumptions of linear models were satisfied. For entry into linear models, we z-scored confidence and accuracy values to transform them to a continuous scale.”

I appreciate the change of title, but the use of the term “reality” does seem a bit strange given the experiment puts participants in the fictional realm of “Fruitville”. Maybe “... in the face of accurate positive feedback” would be more specific, or just “...in the face of feedback”?

We thank the Reviewer for prompting further consideration of the title here. We intend “reality” here to stand in for external, objective information about the world or oneself – rather than indicating something about the ecological validity of the paradigm. This is similar to how this term has been used in narrative introductions of previous work (e.g., in neuroeconomics; Sharot et al. (2011) *Nat Neuro*). In this sense we consider its use here an appropriate characterisation of the study, and would like to keep it as-is. But we would happily revisit this issue if the Reviewer or Editor consider this crucial.

Reviewer #4 (Remarks to the Author):

Summary

I was asked to supply an additional review for this manuscript; thus, my summary and comments come after having reviewed (1) the manuscript, (2) the reviewer comments, and (3) the response to the reviewers. I wrote my review before reading the reviewer comments; where my review was informed by their comments, I note it explicitly below.

In this paper, the authors systematically manipulated performance feedback to influence “global confidence” (i.e., long-run self-performance estimates (SPE)). Game-ified tasks were conducted online, and included both a perceptual task and a memory task. The perceptual task required participants to decide which of two types of berries (raspberries or blackberries) were more numerous. The memory task required participants to decide which fruit was present in a box of fruits that was recently opened. The key manipulation involved the extent to which participants received veridical positive or negative feedback; some blocks contained more positive than negative feedback, and vice versa. Baseline blocks were followed by intervention and test blocks; feedback type was a within-subjects variable, and order effects were manipulated across individuals, resulting in eight groups of participants.

Computational modeling was employed to determine how anxiety and depression symptoms alter local confidence/feedback’s influence on global confidence. As shown in figure 1d-g, either a confidence learning distortion, feedback learning distortion, or a response bias could account for how global SPEs differs as a function of anxious-depression symptoms. The work was conducted in an exploratory manner in Experiment 1, and that validated in a preregistered Experiment 2. Results showed that individuals with greater anxious-depression symptoms exhibited reduced

sensitivity to increases in local confidence, and a confidence distortion plus response-bias model provided the best fit to data in both Experiments 1 and 2.

Overall, I found the paper quite interesting, as it addresses an important and timely question. However, I do have some concerns about how the modeling results are portrayed, as well as whether the anxiety-depression scores may have been influenced by the experimental procedures. I note my major and minor criticisms below.

We thank the Reviewer for their very helpful review, and for these constructive comments which we have addressed in detail below.

Major Concerns

- In Supplementary Figure 9, it is interesting that the confidence learning distortion (D1) and Response Bias (D3) models make opposite predictions (fig. S9a and S9c) about the slope between the low and high anxiety-depression symptom lines for confidence (tiled) 3-4. Your behavioral data in Figure 4a shows a significant difference between these two lines, supporting the confidence learning distortion account, and refuting the response bias account. Yet the model that contains BOTH the confidence and response value appears to win, according to DIC values. I assume this is because of the nonsignificant values shown in Figure 4c and 4d of the effect of feedback type on global SPE. Supplementary Figure 9e doesn't say anything about significance, unlike 9f and 9g. Thus: can you please clarify what is predicted (in terms of significance values) in Supplementary Figure 9e?

We thank the Reviewer for prompting further consideration of these nuances of the model behaviour. The reviewer rightly recognises that Figure 9e (reproduced below as Figure R3e) shows a significant interaction in which the effect of symptoms is greater in the no feedback condition, compared to both the positive and negative feedback conditions when simulating a confidence distortion. However, the direction of this interaction is in the opposite direction to that predicted under a feedback distortion account in Figure 9f. In other words, unlike feedback distortion where the difference between lower and higher anxious-depression scores *increases* with both positive and negative feedback compared to no feedback, in the case of confidence distortion the difference between anxious-depression scores slightly *decreases* with both types of feedback. In our data, while we don't see such significant interactions in either experiment, we do find a trend towards the scenario expected under the confidence distortion account for both types of feedback in Exp 2, and for positive feedback in Exp 1.

We suspect that the reason that we don't see the subtle interaction of the kind observed in Supplementary Figures 9e in our data (Figures 4c and 4d) is that the model operates by incorporating trial-by-trial feedback (and local confidence) into global SPEs, whereas behavioural effects of feedback are examined statistically at a block-level. Thus, the model simulation has the capacity to exhibit more nuanced effects that may be missed by block-level averaging of the kind we use for our behavioural analyses.

Figure R3. Simulated data from different models, organised akin to Figures 1e–1f and Figures 4a–4d, depicting our qualitative statistical predictions. **a, e)** Confidence distortion model where increasing anxious-depression symptoms are associated with a blunted effect of higher local confidence on global SPEs. **b, f)** Feedback distortion model where higher anxious-depression symptoms are associated with global SPEs being more sensitive to negative feedback, and lower symptoms more sensitive to positive feedback. **c, g)** Response bias model where higher anxious-depression symptoms are associated with an overall decrease in global SPEs (but no differences in learning). **d, h)** The null model where global SPEs do not depend on anxious-depression symptoms. Note that the small but significant interaction effects between symptoms and feedback type predicted under the confidence distortion model (e) are in the opposite direction to those predicted under the feedback distortion model (f).

And can you please comment in the Discussion section on how your “winning” model actually contains diametrically opposing predictions as to what the behavioral data in Experiment 1 and Experiment 2 should look like in Figure 4a and 4b? This issue became clear to me after reading the comments from Reviewer 2 on this topic. In Figure R6 that you sent to this reviewer, you are plotting the data that shows support for the confidence learning distortion in R6a and R6b, and support for the response bias effect in R6c and R6d, but you are glossing over the issues I referenced above, and considering the diametrically opposing predictions in Figure S9a and S9c, it warrants further commentary.

Thank you for this observation. You are right – in the winning model, the confidence distortion was negative, as expected (leading to lower SPEs for higher anxious-depression symptoms) and the response bias was counterintuitively positive (leading to higher SPEs for lower anxious-depression symptoms). However, we found through simulations that the fitted values of the response bias parameter were two orders of magnitude smaller than values which would lead to noticeable changes in SPEs. Accordingly, as shown in Supplementary Figure 11c, the fitted response bias parameter in isolation produces a negligible effect on the predicted differences in

SPEs as a function of anxious-depression symptoms. This is now stated with further clarity in the Results (line 249), as follows:

“Finally, and contrary to our expectation, the response bias parameter β_a was significantly positive in both experiments (Exp 1: PHQ, 99% HDI = [1.96e-07 3.86e-05]; Exp 2: Anxious-Depression axis, 99% HDI = [4.46e-05 7.75e-04]; Supplementary Figure 11b), suggesting that, if anything, anxious-depression symptoms are associated with a positive response bias in global SPEs... However, the magnitudes of fitted β_a values were extremely small in both experiments, with values two orders of magnitude smaller than that required to produce observed differences in global SPEs. Accordingly, simulations using similar values revealed a negligible impact on global confidence profiles (Supplementary Figure 11c).”

Because of the negligible impact of response bias on SPEs we decided not to revisit this point in the Discussion.

Also, please note that Figures 4c and 4d (R6c and 6d in the previous rebuttal) were not designed to show evidence for the presence of a response bias but instead show the absence of a feedback distortion. This can be appreciated by the fact that in these plots, higher anxious-depression scores are associated with lower SPEs. But, as mentioned above, the fitted response bias parameter predicted (very small) effects in the opposite direction.

- Not a single model predicts a flat slope between confidence values 1 and 2 for the “low” anxiety-depression group, and yet this is what the data in Experiment 1 shows in Experiment 4a. What do you make of this? Is it simply that participants struggle to distinguish between confidence values 1 and 2 in this task? If so, why do you think you don’t see that same trend in Experiment 2?

Thank you for this insightful comment. As part of our response to a similar point raised by Reviewer 2, we have substantially revised Figures 4a–4b to estimate error bars using a mixed regression model, accounting appropriately for the random effects of participants, blocks, trial numbers in the global SPE data. These new plots more appropriately reflect confidence in the mean SPEs for each condition, and illustrate how it is difficult to say that the model and data are substantially at odds in their predictions of the slope between confidence tiles 1 and 2 in Experiment 1. In other words, we believe any differences in slope between the model predictions and the data of Experiments 1 and 2 are within the range of experimental error.

- p.9 – It appears you have discrepant results between Experiment 1 and Experiment 2 regarding whether feedback type influences local confidence. Any speculation as to why there is a difference across the two experiments?

Thank you. One possibility is that because the specific trials on which feedback was delivered varied randomly between subjects, and in Experiment 2 a larger number of participants may have randomly received more positive than negative feedback during the early half of the block compared to latter half of the block in Experiment 1.

As a result, in Experiment 1, there may have been fewer trials on which the feedback manipulation exerted effects on local confidence. Because this analysis was conducted by averaging local confidence across the block, this potentially explains why we were unable to see any difference in local confidence when comparing feedback valence in Experiment 1.

To ascertain if this explanation explains this discrepancy between experiments, we averaged the feedback difference (positive – negative feedback) across subject as a function of trial number for the two experiments (Figure R4). When doing this, we find this value to be greater in Exp 2 than Exp 1 in the early part of the block, and greater in Exp 1 than Exp 2 in the latter half of the block. This is potentially consistent with subtle differences in the randomised timing of feedback affecting our local confidence measures in the two experiments – but future work will be needed to explicitly manipulate the timing of biased feedback to definitively test for its effects on confidence dynamics.

• p.11 – You claim that “across both experiments, we replicated previous findings showing that heightened anxiety and depression symptoms are associated with lower average local confidence (metacognitive bias) and lower global SPEs (Figure 3d; Supplementary Figure 8). However, when I look at supplementary figure 8, panel A does not reveal a significant correlation between the GAD-7 score and the perceptual task local confidence, and panel D does not reveal a significant correlation between the PHQ-9 score and the memory task global SPE. This would warrant some commentary to scale back your claim in the paper, yes?

Thank you for this astute observation. You are right that in Exp 1 we only observe a relationship of local confidence with anxiety symptoms (GAD-7) but not with depression symptoms (PHQ-9). We have modified the text to now make this clear (line 181):

“We replicated previous findings showing that heightened anxiety symptoms are associated with lower average local confidence (metacognitive bias; Exp 1 – GAD-7: $\chi^2 = 9.76$, $p = .0018$; Exp 2 – Anxious-Depression axis: $\chi^2 = 29.63$, $p < .0001$)² and lower global SPEs (Exp 1: $\chi^2 = 24.11$, $p < .0001$; Exp 2: $\chi^2 =$

20.26, $p < .0001$; Figure 3d; Supplementary Figure 8; see Supplementary Material §7 for details).”

- Can you explain why you use the GAD-7 score in Exp 1 and “transdiagnostic” scores in Experiment 2, before you introduce the results in the manuscript?

We have now modified the statement (line 179) introducing results in relation to mental health symptoms to:

“We next asked how the formation of global confidence estimates related to symptoms of anxiety and depression (measured using standardised questionnaires in Exp 1, and for greater precision measured transdiagnostically in Exp 2).”

- You mention on p.26 that in the final phase of your experiment, participants completed the mental health questionnaires. Isn't it possible that putting these questionnaires at the end influenced how participants would respond? For example, since your data in Figure 5 shows that the feedback type influenced self-endorsement of words of different valence types, why do you think that assessments about mental health would be robust to the different types of feedback provided during the task? I understand the issue in play: if you put these things first, it can induce demand characteristics shaping how participants perform the task. But when you put them last, don't you also need to analyze whether having positive or negative feedback in the last block before taking these assessments influenced participants' scores?

Thank you for raising this issue. When designing the study, we debated whether to put the mental health questionnaires at the beginning or the end. Our decision to put them at the end was based on the consideration that by the end of all task blocks, each participant would have received one negative and positive feedback block in counterbalanced order, thus mitigating any potential effect of feedback on subsequent mental health assessments. Moreover, the very last task block was a test block that did not contain feedback, and thus was matched across participants. At the same time, as you highlight, it's possible that by placing the questionnaires at the end, the impact of demand characteristics might be more severe.

To further evaluate this issue, we performed two-sample t-tests comparing mental health scores for participants who received either positive or negative feedback on the penultimate block. In Exp 1, there was no significant difference between the two groups (GAD-7 scores: $t(228) = .11$, $p = .92$; PHQ-9 scores: $t(278) = .23$; $p = .82$). In Exp 2, again the difference was not significant (Anxious-Depression axis: $t(276) = 1.70$; $p = .09$). We are therefore confident that this design choice does not affect our key conclusions.

Minor Edits

- I was a bit confused by Model D2 in Figure 1f. The way model D1 is described in the caption is that people with higher anxious-depression symptom scores (blue lines) show blunted influence of HIGHER local confidence on global SPEs compared

to lower anxious-depression symptom scores (gold lines). The reduced slope of the blue solid line makes this apparent. In Model D2, the idea is that higher anxious-depression symptoms are associated with a greater sensitivity to incorrect vs. correct feedback. The low global SPE in the “Incorrect” column on the x-axis makes that prediction apparent, but I wasn’t sure how you determined what the right baseline expectation for “Correct” feedback would be under this model. The schematic makes it look like the global SPE for “Correct” feedback is quite comparable to the “None” feedback condition, but couldn’t model D2 also predict a “normal” amount of adjustment to the Global SPE in the “Correct” feedback category, while maintaining greater amount of sensitivity to the incorrect feedback? The way you depict it now, it almost appears that there is both INCREASED sensitivity to incorrect feedback, AND DECREASED sensitivity to correct feedback in the D2 model. In this way, it seems like there is potential for a “D2a” and “D2b” model, where the magnitude of sensitivity to the “Correct” feedback is either normal, or blunted (the way you have it now), respectively. Do you agree? P.11 describes versions of your three original models that combine predictions from one another, but I’m wondering if an additional model that includes this element is warranted.

Thank you for this astute observation. You are correct in that there is a possibility of two different models – one where anxious-depression symptoms are associated solely with less sensitivity to Correct feedback, and another where they are associated with greater sensitivity to Incorrect feedback alone. The model we considered (i.e., incorrect vs. correct feedback) was one where either of these distortions could contribute to the observed effects. We started with this combined model, instead of specifying two separate models where our intention was to model the two distortions independently had we found significant effects in the combined model. Importantly, our model-free analysis in lines 223–232 was conducted separately for Correct and Incorrect feedback, and (in line with the results of the combined model) we did not find an effect of either distortion type.

- I was a bit confused by the first two “until response” panels in the perceptual trials on p.8. Why do the labels for the confidence ratings only appear in the SECOND “until response” panel? Does this mean that those labels appear AFTER the participant first moves the slider? Or are they giving two responses? (One without labels, and one with labels?) It’s not clear based on the current caption. The same criticism applies to the memory trial part of Figure 2b, and needs to be clarified as well. p.23 provides a hint as to what was going on: was this when participants were allowed to switch their choice? If so, this should probably be briefly referenced in the caption.

Thank you. Yes, the confusion here may have been because participants pressed an extra button to proceed after reporting their confidence. Thus, participants had an indefinite amount of time for a) making the choice, b) selecting confidence, and then c) pressing a key after selecting to proceed (following which they could get feedback or blank screen for the next trial blank). We have now modified the caption to make this clearer:

“For the perception task, participants attended the green bush in the middle where 121 stimuli of two different kinds of berries (red raspberries and dark purple blackberries) appeared at non-overlapping random locations for 1000 ms. Participants were asked to respond (without time limit) as to whether

there were more raspberries or blackberries, and then use a slider to report their confidence. After reporting their confidence, participants then pressed a key to proceed. Feedback for correct or incorrect trials appeared with a defined frequency depending on the intervention block (positive or negative).”

- On the perception task trials, it looks like one color is light, while another color is dark. I didn't see any reference in the manuscript about trying to control for luminance across the two choice options. My understanding from Doby Rahnev's work using a similar “choose the more numerous color” task (e.g., Haddara and Rahnev, Psych Science, 2022) is that this is a factor that may influence confidence judgments. If you didn't control for luminance, why do you think this is a factor that is inconsequential for your research?

Thank you for bringing this interesting finding of differences in colours on confidence to our attention. We think it's unlikely that differences between stimuli had an effect on our key dependent variables, for the following reasons. First, target colours were randomised across trials within a block, and consequently any differences in local confidence between colours cannot systematically impact on manipulations of global confidence across blocks. Second, even if there were small colour-induced biases in local confidence, our focus was on how the relationship between local and global confidence interacted with anxious-depression symptoms, a relationship we would not predict would be impacted by stimulus colour.

For completeness, we tested whether the target stimulus colour on a particular trial affected the relationship between local confidence and anxious-depression scores (modelling the interaction term), and as expected, did not find a significant interaction ($p = .21$).

- Can you clarify in figure 4a and 4b whether the p-values labeling the difference between the gold and blue lines are for differences in SLOPE between the two lines, or some other value? Based on the description in the manuscript on p.12, I'm assuming it's slope, but I was hoping to see it labeled in the caption, too.

You are right that the p values are for differences in the slopes. We have now clarified this in the caption for Figure 4 as follows:

“Depicted statistical comparisons are for slopes of higher confidence values between higher and lower anxious-depression scores (* $p < .05$; **** $p < .0005$).”

- Did participants earn money for completing your task? I didn't see any listing of how much they earned for completing it in the participants section of your manuscript. Also, what was the IRB approval information?

Thank you for noticing this omission. We have now added a line (line 537) in the Methods stating,

“Participants were paid £7.50/hour for completing the study, or on a pro-rated basis if they chose to leave the study at any point.”

The ethics committee approval information is provided in the Procedure subsection of the Methods (line 534), as follows:

“Participants from Prolific were given the study link where they were first required to read and agree to a Study Information and Consent Form (both approved by the UCL Research Ethics Committee; approval number 21029/001) before proceeding.”

Reviewer #4 (Remarks on code availability):

I briefly reviewed the code, but I did not have time to run it on my machine, to verify their results. The documentation seems reasonable, and includes a brief "ReadMe" file, as well as links to versions in both MATLAB and R.

REVIEWER COMMENTS

Reviewer #2 (Remarks to the Author):

The authors have mostly addressed the comments from my previous review.

Could the authors please confirm how many participants were simulated in their model recovery analysis (i.e. which experiment 1 or 2 was simulated for model recovery)? Why was model recovery not performed for the winning model?

It is still unclear to me what is plotted in Figure 4a,b. Each participant gives 6 global SPEs, 4 data points are plotted relating local confidence to global SPEs within participants. The authors state that local confidence was z-scored and then quantiled (though z-scoring has no effect on how data are split into quantiles within subjects), but then how is global SPE calculated for each quantile? Is this just sorting average local confidence for the 4 blocks that didn't have feedback? Or some sort of reverse average Global SPE, counting the block global SPE the number of times the local confidence quantile was given in that block? Were all blocks included? The authors state "We also confirm that plots looked qualitatively very similar if confidence data were quantiled across participants, or per block within participants." The latter would result in flat Global SPE if a reverse average Global SPE were calculated (equal probability of each quantile within each block).

The prospective/retrospective global SPE in Figure 3d is not described in the methods. What is described in the methods seems more like a retrospective global SPE.

Reviewer #4 (Remarks to the Author):

I have no further comments for the authors; I am satisfied with their replies to my critiques and comments.

Response to Reviewers: **How underconfidence is maintained in the face of reality**

We are glad to hear that both Reviewers agree that the majority of their comments from the previous round have now been addressed. We are also grateful to Reviewer 2 for their constructive feedback on the model recovery analysis.

We provide a point-by-point response to each comment below. The Reviewers' comments are in standard font, our responses are in **green**, and excerpts from the revised manuscript are in **blue**.

Reviewer #2 (Remarks to the Author):

The authors have mostly addressed the comments from my previous review.

Thank you – we are glad to hear this.

Could the authors please confirm how many participants were simulated in their model recovery analysis (i.e. which experiment 1 or 2 was simulated for model recovery)?

Our simulation procedure for model recovery was as follows. We started with N = 600 participants. For each participant we simulated an experiment for which d' values were drawn from a Gaussian distribution ($\mu = 1.1$, $\sigma = .05$) leading to a range of accuracy values. Then we excluded simulated participants whose accuracy values were outside the interval (.6 .85], to match the exclusion criterion we used on empirical data. This procedure led to ~300 participants per simulated experiment on average – a sample-size similar to Experiment 2. We have now added these details to Supplementary Material §1 under the subheading *Sample size for simulations*.

Why was model recovery not performed for the winning model?

Thank you for prompting further reflection here. We did not perform model recovery on the combined confidence distortion+response bias model due to the very small and negligible impact of the response bias parameter in this combination (as outlined in our previous response to reviewers and revised version of the manuscript, the response bias Beta parameter β_a had extremely small values, 2 to 3 order of magnitude smaller than what would produce a difference in global SPEs through anxious-depression symptoms). In other words, this model is practically indistinguishable from the confidence distortion-only model, which captures the key

experimental results, and which shows good model and parameter recovery. We have now further investigated if a full combinatorial model recovery procedure would be necessary for further interpreting our findings. To this end, we first tested if β_a is recoverable across a range of values comparable to the fitted values in the winning model, even when it is simulated as the only distortion (in the manuscript we had originally reported this analysis for the larger range of values of β_a that would be required to exhibit the observed effect of difference in global SPEs with anxious-depression symptoms and found good recovery, but had not considered recovery across the much smaller parameter range obtained in the fits of the combined distortion+bias model). As shown in Figure R1, within this narrower range, β_a values were not recoverable. Interestingly, here the recovered values were always biased towards being positive, similar to the positive sign of parameter estimates obtained when fitting this model to empirical data.

Figure R1. Recovered vs. simulated values of β_a when simulating and fitting only response bias, showing that β_a was not recoverable in simulation for very small values occupying a range similar to fitted values.

Next, we performed recovery on β_a in the context of the best fitting model – i.e., containing both confidence distortion and response biases in the range of fitted values of response bias. We again found that β_a in this range was not recoverable (Figure R2), and the recovered values were positively biased (we note that β_c is recoverable, as described in the manuscript).

Figure R2. Recovered vs. simulated values of β_a (left) and β_c (right) when simulating and fitting both confidence distortion and response bias. Here, similar to Figure R1, β_a was not recoverable for very small values.

Together these analyses suggest that the model fits of the winning combinatorial model spuriously result in very small positive values in β_a that are not meaningful or useful for explaining subjects' behaviour. To confirm this point, we also simulated data from a model with only a confidence distortion (and not response bias), and fitted both β_a and β_c (akin to the architecture of our best-fitting model). We found that the fitting procedure indeed induced small positive values in β_a – again supporting their spurious nature (Figure R3).

Figure R3. Recovered vs. simulated values of β_a when simulating only confidence distortion and fitting both confidence distortion and response bias. The fitting procedure spuriously induces small positive values into β_a .

The question remains as to why this apparently overfitted distortion+bias model was preferred in the model comparison procedure. As a first step towards understanding this, we compared DIC scores of confidence distortion and confidence distortion+bias models fitted to data generated by only a confidence distortion process. Despite the negligible influence of the bias term, we found that the DIC values were smaller (indicating a better fit) for the confidence distortion+bias model compared to the (correct) confidence distortion-only model (Figure R4). In other words, the DIC scores slightly preferred model complexity over ground truth. Having reviewed the current status of DIC as a model comparison metric, we think this is consistent with a known issue with DIC in favouring more complex / overfitted models (Spiegelhalter et al, 2014, *Journal of the Royal Statistical Society Series B: Statistical Methodology*).

Figure R4. Histogram of the difference in DIC values when fitting 2 parameters (β_a and β_c) compared to fitting only β_c when the true underlying model contained only confidence distortion. The difference values were always negative showing that the 2-parameter model fit the data better even when simulation included a 1-parameter model.

Overall, these explorations suggest that the small positive value of additive bias we observed in both experiments is a spurious outcome of the model fitting procedure. Instead, a more parsimonious model is one that that exhibits only a confidence distortion process, and not an additional, small response bias. (We note that in Exp 2, the confidence plus feedback distortion model was slightly preferred to the confidence distortion-only model, but because the fitted feedback distortion parameter was not significantly different from zero, we concluded that both experiments showed only a confidence distortion. Again, we think that this

slight preference for the confidence+feedback distortion model in DIC scores indicates a bias towards favouring more complex models).

In summary, then, we greatly appreciate this push to relax our reliance on JAGS' DIC scores when evaluating model fit, and instead rely more heavily on model and parameter recovery simulations, and qualitative matches between model and data. We are grateful for the reviewer for encouraging these approaches, which has significantly strengthened the robustness of our modelling procedure here, and has greatly benefited the manuscript.

We have now modified the Results paragraph on model comparison to reflect these points, as follows:

“We next pursued formal model comparisons (Figure 4e and 4f). In both Exps 1 and 2, models that contained a confidence distortion either on its own (D1) or combined with either response bias (D5) or feedback distortion (D4) provided better fits to data than models without a confidence distortion. The model-estimated posteriors of the regression slope parameters relating anxious-depression symptoms to confidence distortions (β_c) and feedback distortions (β_f) from the confidence-plus-feedback distortion model (D4) are depicted in Figures 4g and 4h. In both experiments, the confidence distortion parameter β_c was significantly negative (Exp1: GAD-7, 99% HDI = [-.037 – .028]; Exp 1: PHQ-9, 99% HDI = [-.048 –.037]; Exp 2: Anxious-Depression axis, 99% HDI = [-.101 –.030]) while the feedback distortion parameter β_f did not differ from 0 (GAD-7, 99% HDI = [-.029 .036]; Exp 1: PHQ-9, 99% HDI = [-.034 .035]; Exp 2: 99% HDI = [-.030 .370]). Fitting the confidence distortion plus response bias model (D5) or the confidence distortion only model (D1) to both experiments yielded similar values of β_c (Supplementary Figure 11a; see Supplementary Figure 12 for regression parameter posteriors for all models). Finally, although DIC scores indicated that the confidence distortion plus response bias model (D5) provided a better fit to the data, the magnitudes of fitted β_a values were extremely small in both experiments, with values two-to-three orders of magnitude smaller than those required to produce observed differences in global SPEs (Supplementary Figure 11c). In addition, and contrary to our expectation, these very small values of the response bias parameter β_a were significantly positive in both experiments (Exp 1: PHQ, 99% HDI = [1.96e-07 3.86e-05]; Exp 2: Anxious-Depression axis, 99% HDI = [4.46e-05 7.75e-04]; Supplementary Figure 11b). Further analyses indicated that β_a values of this magnitude cannot be recovered in simulation and are likely to be spurious (we note that a slight bias towards preferring complex models is a known issue with DIC; Spiegelhalter et al., 2014). Taken together, these investigations indicate that the most parsimonious model of the data in both experiments was the confidence distortion-only model (D1). Such a distortion is manifest in individuals with greater anxious-depression symptoms exhibiting a blunted sensitivity to local confidence when forming global SPEs.”

We have also updated Figures 4a–d to display model fits from the confidence distortion-only model.

In light of these investigations, we think there is little additional to be gained in pursuing a full combinatorial model recovery analysis (which is computationally intensive, resting on MCMC sampling within the hierarchical model within each loop of the simulation, leading to several days compute time on a local machine per model).

It is still unclear to me what is plotted in Figure 4a,b. Each participant gives 6 global SPEs, 4 data points are plotted relating local confidence to global SPEs within participants. The authors state that local confidence was z-scored and then quantiled (though z-scoring has no effect on how data are split into quantiles within subjects), but then how is global SPE calculated for each quantile? Is this just sorting average local confidence for the 4 blocks that didn't have feedback? Or some sort of reverse average Global SPE, counting the block global SPE the number of times the local confidence quantile was given in that block? Were all blocks included? The authors state "We also confirm that plots looked qualitatively very similar if confidence data were quantiled across participants, or per block within participants." The latter would result in flat Global SPE if a reverse average Global SPE were calculated (equal probability of each quantile within each block).

We appreciate the Reviewer seeking clarity on this point. We think the confusion here stems from the fact that the data (y-axis) for these plots were obtained using regression models that used continuous (and not quantiled) confidence values, and the data was then superimposed upon confidence quantiles for visualization in Figure 4a,b.

For clarity, we now outline the step-by-step procedure for generating the plots in Methods as follows:

"To create plots of behavioural data in Figures 4a–4b, we used the following procedure. First, we regressed the interaction between Anxious-Depression scores and z-scored local confidence (*ConfZ*) upon global SPEs separately for high local confidence ($ConfZ > 0$) and low local confidence ($ConfZ \leq 0$) trials while regressing out random effects of *Participant*, *Block number*, and *Trial number*. Next, we divided *ConfZ* values within each participant into four quantiles (i.e., each containing equal probability mass) and calculated the centre local confidence value for each quantile by taking the mean of *ConfZ* within each quantile across participants. Finally, we used the regression model to predict global SPEs and their 95% confidence intervals at these four centre *ConfZ* values (using the *emmeans* function in R). This process of standardizing and discretizing the x-axis (local confidence) across subjects aided visualization and facilitated the construction of appropriate confidence intervals for overlaying the data and model fits. Note that discretized local confidence bins were only used for visualisation purposes; all statistics were obtained from analyses of continuous confidence values."

When using such a procedure, the exact way we quantiled the data (within subjects, across subjects, or within blocks within subjects) had little impact on the centre *ConfZ* values, which were obtained by averaging across all subjects. All 6 global SPE blocks were used for these analyses.

The prospective/retrospective global SPE in Figure 3d is not described in the methods. What is described in the methods seems more like a retrospective global SPE.

Thank you for catching this omission on our part. In Experiment 2 (but not Experiment 1), we measured prospective SPEs for each task at a single timepoint after participants received instructions on each task (and before they performed practice trials). To clarify this distinction, we made the following edits to the Methods section:

1. Where we first refer to global SPEs in the Methods for Exp 1 (line 514), we now add the qualifier “retrospectively” in parenthesis, as follows:
“To measure global SPEs, at the end of the block (i.e., retrospectively), participants were shown a slider and asked to indicate how many trials they believed they answered correctly on that block.”
2. In the Methods for Exp 2:
“In Exp 2, instructions for each task were followed by a prospective global SPE where participants were asked to estimate how many trials they thought they would respond to correctly out of a 100 on that task. Instructions (and in Exp 2, the prospective global SPE) were followed by 50 practice trials of that particular task, which also served the purpose of initiating the staircase.”

Reviewer #4 (Remarks to the Author):

I have no further comments for the authors; I am satisfied with their replies to my critiques and comments.

We are glad to hear this, and thank the reviewer for their very helpful review.

REVIEWERS' COMMENTS

Reviewer #2 (Remarks to the Author):

The authors have addressed all of my previous comments. The authors should also update the legend/description of Figure 4 to reflect the analysis described to produce the plots. The legend currently reads: "Figure 4. Data from Exp 1 (left column panels a, c, e) and Exp 2 (right column panels b, d, f). Observed data are plotted as filled circles; model fits are plotted as unfilled triangles. a & b) Global SPEs plotted against local confidence (z-scored and split into 4 quantiles per individual) separately for individuals belonging to the lowest..." But figures A and B do not contain observed data, the 'Global SPEs' are predicted global SPEs from a regression model.

Response to Reviewers: **Distorted learning from local metacognition supports transdiagnostic underconfidence**

We provide a point-by-point response to each comment below. The Reviewers' comments are in standard font, our responses are in **green**, and excerpts from the revised manuscript are in **blue**.

Reviewer #2 (Remarks to the Author):

The authors have addressed all of my previous comments.

We are glad to hear that we have addressed all of Reviewer 2's previous comments.

The authors should also update the legend/description of Figure 4 to reflect the analysis described to produce the plots. The legend currently reads: "Figure 4. Data from Exp 1 (left column panels a, c, e) and Exp 2 (right column panels b, d, f). Observed data are plotted as filled circles; model fits are plotted as unfilled triangles. a & b) Global SPEs plotted against local confidence (z-scored and split into 4 quantiles per individual) separately for individuals belonging to the lowest..." But figures A and B do not contain observed data, the 'Global SPEs' are predicted global SPEs from a regression model.

Thank you. We have now updated the legend to address this point, as follows:

"Results from Exp 1 (left column panels a, c, e; N = 230) and Exp 2 (right column panels b, d, f; N = 287). Mean global self-performance estimates (SPEs) and 95% confidence intervals depicted in panels (a)—(d) are obtained from linear regression models after regressing out random effects of participant, block number, and trial number from observed data (filled circles) and model fits (unfilled triangles). a & b) Global SPEs plotted against local confidence (z-scored and split into 4 quantiles per individual) separately for individuals belonging to the lowest (gold) and highest (blue) tertiles of anxious-depression symptom scores (obtained using the GAD-7 questionnaire in Exp 1 and transdiagnostically in Exp 2). Depicted statistical comparisons are for slopes of higher confidence values between higher and lower anxious-depression scores. c & d) Global SPEs plotted for blocks with more frequent negative feedback, no feedback, and more frequent positive feedback, separately for individuals with low and high anxious-depression scores."